# Least Squares Regression with Markovian Data: Fundamental Limits and Algorithms

**Guy Bresler**
Massachusetts Institute of Technology
Cambridge, USA 02139
guy@mit.edu

**Prateek Jain**
Microsoft Research
Bengaluru, India 560001
prajain@microsoft.com

**Dheeraj Nagaraj**
Massachusetts Institute of Technology
Cambridge, USA 02139
dheeraj@mit.edu

**Praneeth Netrapalli**
Microsoft Research
Bengaluru, India 560001
praneeth@microsoft.com

**Xian Wu**
Stanford University
Stanford, USA 94305
xwu20@stanford.edu

## Abstract

We study the problem of least squares linear regression where the datapoints are *dependent* and are sampled from a Markov chain. We establish sharp information theoretic minimax lower bounds for this problem in terms of $\tau_{\mathrm{mix}}$, the mixing time of the underlying Markov chain, under different noise settings. Our results establish that in general, optimization with Markovian data is *strictly* harder than optimization with independent data and a trivial algorithm (SGD-DD) that works with only one in every $\tilde{\Theta}(\tau_{\mathrm{mix}})$ samples, which are approximately independent, is minimax optimal. In fact, it is strictly better than the popular Stochastic Gradient Descent (SGD) method with constant step-size which is otherwise *minimax optimal* in the regression with *independent* data setting.

Beyond a worst case analysis, we investigate whether structured datasets seen in practice such as Gaussian auto-regressive dynamics can admit more efficient optimization schemes. Surprisingly, even in this specific and natural setting, Stochastic Gradient Descent (SGD) with constant step-size is still no better than SGD-DD. Instead, we propose an algorithm based on experience replay–a popular reinforcement learning technique–that achieves a significantly better error rate. Our improved rate serves as one of the first results where an algorithm outperforms SGD-DD on an interesting Markov chain and also provides one of the first theoretical analyses to support the use of experience replay in practice.

## 1 Introduction

Typical machine learning algorithms and their analyses crucially require the training data to be sampled independently and identically (i.i.d.). However, real-world datapoints collected can be highly dependent on each other. One model for capturing data dependencies which is popular in many applications such as Reinforcement Learning (RL) is to assume that the data is generated by a Markov process. While it is intuitive that the state of the art optimization algorithms with provable guarantees for iid data will not converge as quickly or as efficiently for dependent data, a

solid theoretical foundation that rigorously quantifies tight fundamental limits and upper bounds on popular algorithms in the non-asymptotic regime is sorely lacking. Moreover, popular schemes to break temporal correlations in the input datapoints that have been shown to work well in practice, such as *experience replay*, are also wholly lacking in theoretical analysis. Through the classical problem of linear least squares regression, we first present fundamental limits for Markovian data, followed by an in-depth study of the performance of Stochastic Gradient Descent and variants (ie with experience replay). Our work is comprehensive in its treatment of this important problem and in particular, we offer the first theoretical analysis for experience replay in a structured Markovian setting, an idea that is widely adopted in practice for modern deep RL.

There exists a rich literature in statistics, optimization and control that studies learning/modeling/optimization with Markovian data [1, 2, 3, 4]. However, most of the existing analyses work only in the infinite data regime, ie [5]. [6] provides non-asymptotic analysis of the Mirror Descent method for Markovian data and [7] provides a similar analysis for TD($\lambda$) (Temporal Difference) algorithms which are widely used in RL. However, the provided guarantees are in general suboptimal and seem to at best match the simple data drop technique where most of the data points are dropped. [8] considers constant step size TD($\lambda$) algorithm but their guarantees suffer from a constant bias. [9] establishes a finite-time bound for Q-learning with linear function approximation and constant step size under a restricted sampling distribution. In parallel to our work, [10] considers an accelerated gradient descent algorithm for general optimization problems and establish that for decaying step sizes the rates in the Markovian case incur an additional logarithmic factor of the mixing time compared to the iid case.

Stochastic Gradient Descent (SGD) is the modern workhorse of large scale optimization and often used with dependent data, but our understanding of its performance is also weak. Most of the results, ie [2] are asymptotic and do not hold for finite number of samples. Works like [11, 12] do provide stronger results but can handle only weak dependence among observations rather than the general Markovian structure. The work which is closest to our setting is [6] which presents non-asymptotic analyses for uniformly Lipschitz convex optimization to obtain minimax optimal rates of the order $G\sqrt{\frac{\tau_{\mathrm{mix}}}{T}}$ using decreasing step-sizes. The loss function for linear regression does not satisfy these assumptions. In contrast, we obtain rates of the order $\frac{\tau_{\mathrm{mix}}}{T}$ in the general agnostic noise case and under the assumption of independent noise, we are able to remove the $\tau_{\mathrm{mix}}$ factor. In fact, in general, the existing rates are no better than those obtained by a trivial SGD-Data Drop (SGD-DD) algorithm which reduces the problem approximately to the i.i.d. setting by processing *only one* sample from each batch of $\tau_{\mathrm{mix}}$ training points. These results suggest that optimization with Markov chain data is a strictly harder problem than the i.i.d. data setting, and also SGD might not be the "correct" algorithm for this problem. We refer to [1, 7] for similar analyses of the related TD learning algorithm widely used in reinforcement learning.

To gain a more complete understanding of the fundamental problem of optimization with Markovian data, our work addresses the following two key questions. **Q1**: what are the fundamental limits for learning with Markovian data and how does the performance of SGD compare. **Q2**: can we design algorithms with better error rates than the trivial SGD-DD method that throws out most of the data.

We investigate these questions for the classical problem of linear least squares regression. We establish algorithm independent *information theoretic lower bounds* which show that the minimax error rates are necessarily worse by a factor of $\tau_{\mathrm{mix}}$ compared to the i.i.d. case and surprisingly, these lower bounds are achieved by the SGD-DD method. We also show that SGD is not minimax optimal when observations come with independent noise, and that SGD may suffer from constant bias when the noise correlates with the data.

To study **Q2**, we restrict ourselves to a simple Gaussian Autoregressive (AR) Markov Chain which is popularly used for modeling time series data [13]. Surprisingly, even for this restricted Markov chain, SGD does not perform better than the SGD-DD method in terms of dependence on the mixing time. However, we show that a method similar to experience replay [14, 15, 16], that is popular in reinforcement learning, achieves significant improvement over SGD for this problem. To the best of our knowledge, this represents the first rigorous analysis of the experience replay technique, supporting it's practical usage. Furthermore, for a non-trivial Markov chain, this represents first improvement over performance of SGD-DD.

We elaborate more on our problem setup and contributions in the next section.

| Setting | Algorithm | Lower/upper | Bias | Variance |
|---|---|---|---|---|
| Agnostic | Information theoretic | Lower | $\exp\left(\frac{-T}{\kappa\tau_{\mathrm{mix}}}\right)\|w_0-w^*\|^2$ Theorem 1 | $\frac{\tau_{\mathrm{mix}}\sigma^2 d}{T}$ Theorem 2 |
| | SGD | Lower | Constant    Theorem 4 | — |
| | SGD-DD | Upper | $\exp\left(\frac{-T}{\kappa\tau_{\mathrm{mix}}}\right)\|w_0-w^*\|^2$ Theorem 8 | $\frac{\tau_{\mathrm{mix}}\sigma^2 d}{T}$ Theorem 8 |
| Independent | Information theoretic | Lower | $\exp\left(\frac{-T}{\kappa\tau_{\mathrm{mix}}}\right)\|w_0-w^*\|^2$ Theorem 1 | $\frac{\sigma^2 d}{T}$ [17] |
| | SGD | Lower | — | $\frac{\tau_{\mathrm{mix}}\sigma^2 d}{T}$ Theorem 5 |
| | Parallel SGD | Upper | $\exp\left(\frac{-T}{\kappa\tau_{\mathrm{mix}}}\right)\|w_0-w^*\|^2$ Theorem 9 | $\frac{\sigma^2 d}{T}$ Theorem 9 |
| Gaussian Autoregressive | SGD | Lower | $\exp\left(\frac{-T\log(d)}{\kappa\tau_{\mathrm{mix}}}\right)\|w_0-w^*\|^2$ Theorem 6 | — |
| Dynamics with Independent Noise | SGD-ER (Algorithm 4) | Upper | $\exp\left(\frac{-T\log(d)}{\kappa\sqrt{\tau_{\mathrm{mix}}}}\right)\|w_0-w^*\|^2$ Theorem 7 | $\frac{\sqrt{\tau_{\mathrm{mix}}}\sigma^2 d}{T}$ Theorem 7 |

**Table 1:** See Section 2 for a description of the three settings considered in this paper. We suppress universal constants and log factors in the expressions above. For linear regression with i.i.d. data, tail-averaged SGD with constant stepsize achieves minimax optimal bias and variance rates of $\exp\left(\frac{-T}{\kappa\tau_{\mathrm{mix}}}\right)\|w_0-w^*\|^2$ and $\frac{\sigma^2 d}{T}$ respectively. In contrast, even the minimax rates in the general agnostic Markov chain setting are $\tau_{\mathrm{mix}}$-factor worse, and tail-averaged SGD with constant-step size is not able to achieve these rates as well. We modify and analyze variants of SGD (i.e., SGD-DD and Parallel SGD) that achieve close to minimax error rates. Finally, for the Gaussian Autoregressive Markov chain, SGD still achieves a trivial bias error rate while our proposed experience replay based SGD-ER method can decay the bias significantly faster.

## 1.1 Notation and Markov Chain Preliminaries

In this work, $\|\cdot\|$ denotes the standard $\ell^2$ norm over $\mathbb{R}^d$. Given any random variable $X$, we use $\mathcal{D}(X)$ to denote the distribution of $X$. $\mathrm{TV}(\mu,\nu)$ denotes the total variation distance between the measures $\mu$ and $\nu$. Sometimes, we abuse notation and use $\mathrm{TV}(X,Y)$ as shorthand for $\mathrm{TV}(\mathcal{D}(X),\mathcal{D}(Y))$. We let $\mathrm{KL}(\mu\|\nu)$ denote the KL divergence between measures $\mu$ and $\nu$. Consider a time invariant Markov chain MC with state space $\Omega \subset \mathbb{R}^d$ and transition matrix/kernel $P$. We assume throughout that MC is ergodic with stationary distribution $\pi$. For $x \in \Omega$, by $P^t(x,\cdot)$ we mean $\mathcal{D}(X_{t+1}|X_1 = x)$, where $X_1, X_2, \ldots, X_{t+1} \sim \mathrm{MC}$.

For a given Markov chain MC with transition kernel $P$ we consider the following standard measure of distance from stationarity at time $t$,

$$\mathrm{d}_{\mathrm{mix}}(t) := \sup_{x\in\Omega}\mathrm{TV}(P^t(x,\cdot),\pi).$$

We note that all irreducible aperiodic finite state Markov chains are ergodic and exponentially mixing i.e, $\mathrm{d}_{\mathrm{mix}}(t) \leq Ce^{-ct}$ for some $C, c > 0$. For a finite state ergodic Markov chain MC, the *mixing time* is defined as

$$\tau_{\mathrm{mix}} = \inf\{t : \mathrm{d}_{\mathrm{mix}}(t) \leq 1/4\}.$$

We note the standard result that $\mathrm{d}_{\mathrm{mix}}(t)$ is a decreasing function of $t$ and whenever $t = l\tau_{\mathrm{mix}}$ for some $l \in \mathbb{N}$, we have

$$\mathrm{d}_{\mathrm{mix}}(l\tau_{\mathrm{mix}}) \leq 2^{-l}. \tag{1}$$

See Chapter 4 in [18] for further details.

## 2 Problem Formulation and Main Results

Let $X_1 \rightarrow X_2 \rightarrow \cdots \rightarrow X_T$ be samples from an irreducible Markov chain MC with each $X_t \in \Omega \subset \mathbb{R}^d$. Let $(\mu_x)$ for $x \in \Omega$ be any set of probability measures over $\mathbb{R}$ indexed by elements of $\Omega$ (alternately $\mu(,) : \Omega \times \mathcal{B}(\mathbb{R}) \rightarrow [0,1]$ is a Markov kernel where $\mathcal{B}(\mathbb{R})$ is the set of Borel measurable sets of $\mathbb{R}$). Associated with the Markov chain $(X_t)_{t=1}^T$ is the sequence of observations $(Y_t)_{t=1}^T$ over the same probability space such that $Y_t \sim \mu_{X_t}$. Clearly, $Y_t$ is conditionally independent of $(X_s)_{s\neq t}$ given $X_t$. Given samples $(X_1, Y_1), \cdots, (X_T, Y_T)$, our goal is to estimate a parameter $w^*$ that minimizes the out-of-sample loss, which is the expected loss on a new sample $(X, Y)$ where $X$

is drawn independently from the stationary distribution $\pi$ of MC and $Y \sim \mu_X$:

$$w^* = \arg\min_{\mathbb{R}^d} \mathcal{L}_{MC}(w), \quad \text{where} \quad \mathcal{L}_{MC}(w) := \mathbb{E}_{X \sim \pi}\left[\left(X^\top w - Y\right)^2\right]. \qquad (2)$$

Define $A := \mathbb{E}_{X \sim \pi} X X^\intercal$. Let $\|X_t\| \leq 1$ almost surely and $A = \mathbb{E}_{X \sim \pi} X X^\intercal \succeq \frac{1}{\kappa} I$ for some finite 'condition number' $\kappa \geq 1$, implying unique minimizer $w^*$. Also, let $\upsilon < \infty$ be such that $\mathbb{E}\left[|Y_t|^2 | X_t = x\right] \leq \upsilon$ for every $X_t \in \Omega$. We define the 'noise' or 'error' to be $n_t(X_t, Y_t) := Y_t - \langle X_t, w^* \rangle$ and by abusing notation, we denote $n_t := Y_t - \langle X_t, w^* \rangle$. We also let $\sigma^2 := \mathbb{E}_{X_t \sim \pi}\left[n_t^2\right]$.

## 2.1 Problem Settings

Our main results are in the context of the following problem settings:

- **Agnostic setting**: In this setting, the vectors $X_i$ are stationary (distributed according to $\pi$) and come from a finite state space $\Omega \subseteq \mathbb{R}^d$. We do not make any additional assumptions on the noise. This class of noise is the most general and includes heteroscedastic noise models where the variance of noise $n_t$ depends on $X_t$.
- **Independent noise setting**: In addition to our assumptions in the agnostic setting, in this setting, we assume that $n_t(X)$ is an independent and identically distributed zero mean random variable with variance $\sigma^2$ for all $X \in \Omega$. This noise model is necessarily homoscedastic i.e, the variance of noise $n_t$ does not depend on $X_t$.
- **Experience Replay for the Gaussian Autoregressive Chain**: In this setting, we fix a parameter $\epsilon$ and consider the non-stationary Markov chain $X_t$ that evolves as $X_t = \sqrt{1 - \epsilon^2} X_{t-1} + \epsilon g_t$, where $g_t \sim \frac{1}{\sqrt{d}} \mathcal{N}(0, I)$ is sampled independently for different $t$. The observations $Y_t$ are given by $\langle X_t, w^* \rangle + \xi_t$ for some fixed $w^*$, and $\xi_t$ is an independent mean 0 variance $\sigma^2$ random variable.

## 2.2 Main Results

We are particularly interested in understanding the limits (both upper and lower bounds) of SGD type algorithms, with constant step sizes, for solving (2). These algorithms are, by far, the most widely used methods in practice for two reasons: 1) these methods are memory efficient, and 2) constant step size allows decreasing the error rapidly in the beginning stages and is crucial for good convergence. In general, the error achieved by any SGD type procedure can be decomposed as a sum of two terms: *bias* and *variance* where the bias part depends on step size $\alpha$ and $\|w_1 - w^*\|^2$ and the variance depends on $\sigma^2$, where $w_1$ is the starting iterate of the SGD procedure. Thus,

$$\mathbb{E}\mathcal{L}_{MC}(w_T^{\text{SGD}}) - \mathcal{L}_{MC}(w^*) = \mathcal{L}_{MC}^{\text{bias}}\left(\|w_1 - w^*\|^2, T\right) + \mathcal{L}_{MC}^{\text{variance}}\left(\sigma^2, T\right). \qquad (3)$$

The bias term arises because the algorithm starts at $w_1$ and needs to travel a distance of $\|w_1 - w^*\|$ to the optimum. The variance term arises because the gradients are stochastic and even if we initialize the algorithm at $w^*$, the stochastic gradients are nonzero.

We provide a brief summary of our contributions below; See Table 1 for a comprehensive overview:

- For general least squares regression with Markovian data, we give information theoretic minimax lower bounds under different noise settings that show that any algorithm will suffer from slower convergence rates (by a factor of $\tau_{\text{mix}}$) compared to the i.i.d. setting (Section 3). We then show via algorithms like SGD-DD and parallel SGD that the lower bounds are tight.
- We study lower bounds for SGD specifically and show that SGD converges at a suboptimal rate in the independent noise setting and that SGD with with constant step size and averaging might not even converge to the optimal solution in the agnostic noise setting. (Section 4).
- For Gaussian Autoregressive (AR) dynamics, we show that SGD with experience replay can achieve significantly faster convergence rate (by a factor of $\sqrt{\tau_{\text{mix}}}$) compared to vanilla SGD. This is one of the first analyses of experience replay that validates its effectiveness in practice. (Section 5). Simulations confirm our analysis and indicates that our derived rates are tight.

## 3 Information Theoretic Minimax Lower Bounds for Bias and Variance

We consider the class $\mathcal{Q}$ of all Markov chain linear regression problems $Q$, as described in Section 2, where the following conditions hold:

1. The optimal parameter has norm $\|w^*\| \leq 1$.
2. Markov chain MC is such that $\tau_{\mathrm{mix}} \leq \tau_0 \in \mathbb{N}$.
3. The condition number $\kappa \leq \kappa_0$.
4. Noise sequence from a noise model $\mathcal{N}$ (ex: independent noise, noiseless, agnostic etc.)

We want to lower bound the minimax excess risk:

$$\mathcal{L}(\mathcal{Q}) := \inf_{\mathrm{ALG} \in \mathcal{A}} \sup_{Q \in \mathcal{Q}} \mathbb{E}\left[\mathcal{L}_Q\big(\mathrm{ALG}\left(D_Q(T)\right)\big)\right] - \mathcal{L}_Q(w_Q^*), \tag{4}$$

where for a given $Q \in \mathcal{Q}$, $\mathcal{L}_Q$ is the loss function with optimizer $w_Q^*$, and the class of algorithms $\mathcal{A} := \{\mathrm{ALG} : (\mathbb{R}^d \times \mathbb{R})^T \to \mathbb{R}^d\}$ which take as input the data $D_Q(T) := \{(X_t, Y_t) : 1 \leq t \leq T\}$ and output an estimate $\mathrm{ALG}(D_Q(T))$ for $w_Q^*$.

### 3.1 General Minimax Lower Bound for Bias Decay

Theorem 1 gives the most general minimax lower bound which holds for any algorithm, in any kind of noise setting. In particular, this gives a bound on the 'bias' term in the bias-variance decomposition of SGD algorithm's excess loss (3) by letting noise variance $\sigma^2 \to 0$.

**Theorem 1.** *In the definition of $\mathcal{Q}$, we let $\mathcal{N}$ be any noise model. Then, for any mixing time $\tau_0$ and condition number $\kappa_0 \geq 2$, we have: $\mathcal{L}(\mathcal{Q}) \geq \frac{\kappa_0 - 1}{\kappa_0^2} \left(1 - \frac{C}{\tau_0 \kappa_0}\right)^T$, where $C$ is a universal constant.*

See Appendix C.1 for a complete proof. Note that the bias decay rate is a $\tau_{\mathrm{mix}}$ factor worse than that in the i.i.d. data setting [19]. Furthermore, our result holds for *any noise* model, and for all settings of key parameters $\kappa_0$ and $\tau_0$. This implies that unless the Markov chain itself has specific structure, one cannot hope to design an algorithm with better *bias decay rate* than the trivial SGD-DD method. Section 5 describes a class of Markov chains for which improved rates are indeed possible.

### 3.2 A Tight Minimax Lower Bound for Agnostic Noise Setting

We now present a minimax lower bound in the agnostic setting (Section 2.1). The bound analyzes the variance term $\mathcal{L}_{\mathcal{Q}}^{\mathrm{variance}}\left(\sigma^2\right)$. Again, we incur an additional, unavoidable $\tau_{\mathrm{mix}}$ factor compared to the setting with i.i.d. samples (Table 1).

**Theorem 2.** *For the class of problems $\mathcal{Q}$ with noise model $\mathcal{N}$ of agnostic noise and for the class of algorithms $\mathcal{A}$ defined above, we have $\mathcal{L}(\mathcal{Q}) \geq c_1 \frac{\tau_0 \sigma^2 d}{T}$, where $T$ is the number of observed data points such that $T \geq c_2 d^2 \tau_0 \sigma^2$ and $c_1, c_2$ are universal constants.*

*Furthermore, SGD-DD achieves above mentioned rates up to logarithmic factors (Theorem 8).*

This result combined with Theorem 1, implies that for general MC in agnostic noise setting, both the bias and the variance terms suffer from an additional $\tau_0$ factor. Our proof shows existence of two different MCs whose evolution till time $T$ can be coupled with high probability and hence they give the same sequence of data. But, since the chains are different and the noise is agnostic, the corresponding optimum parameters $w^*$ are different. See Appendix C.2 for a detailed proof.

---

**Algorithm 1** SGD-DD

---

**Require:** $T \in \mathbb{N}$, $(X_1, Y_1), \ldots, (X_T, Y_T) \in \mathbb{R}^d \times \mathbb{R}$, step size $\alpha > 0$, initial point $w_1 \in \mathbb{R}^d$, drop number $K \leq T$
    **for** t in range $[1, T/K]$ **do**
        Set

$$w_{t+1} \leftarrow w_t - \alpha X_{tK}\left(\langle w_t, X_{tK}\rangle - Y_{tK}\right). \tag{5}$$

    **end for**
    **return** $\hat{w} \leftarrow \frac{2K}{T} \sum_{s=T/2K+2}^{T/K+1} w_s$.

---

### 3.3 A Tight Minimax Lower Bound for Independent Noise Setting

We now discuss the variance lower and upper bound for the general Markov Chain based linear regression when the noise is *independent* (Section 2.1).

**Theorem 3.** *For the class of problems $\mathcal{Q}$ with noise model $\mathcal{N}$ of independent noise and for the class of algorithms $\mathcal{A}$ defined above, we have $\mathcal{L}(\mathcal{Q}) \geq \frac{d\sigma^2}{T}$. This bound is tight up to logarithmic factors since 'Parallel SGD' achieves the rates established above. (Theorem 9, Section A.2)*

Note that the lower bound follows directly from the classical iid samples case (Theorem 1 in [20]) apply. For upper bound, we propose and study a Parallel SGD method discussed in detail in Appendix D.4.2. Interestingly, SGD with constant step size, which is minimax optimal for i.i.d samples with independent noise, is not minimax optimal when the samples are Markovian. We establish this fact and others in the next section in our study of SGD.

---

**Algorithm 2** Parallel SGD

---

**Require:** $T \in \mathbb{N}$ , $(X_1, Y_1), \ldots, (X_T, Y_T) \in \mathbb{R}^d \times \mathbb{R}$ , step size $\alpha > 0$, parallelization number $K \leq T$, initial points $w_1^{(i)} \in \mathbb{R}^d$ for $1 \leq i \leq K$.
    **for** $t$ in range $[1, T/K]$ **do**
        **for** $i$ in range $[1, K]$ **do**
            Set $w_{t+1}^{(i)} \leftarrow w_t^{(i)} - \alpha X_{(t-1)K+i} \left( \langle X_{(t-1)K+i}, w_t^{(i)} \rangle - Y_i \right)$
        **end for**
    **end for**
    **return** $\hat{w} \leftarrow \frac{2}{T} \sum_{i=1}^{K} \sum_{t=T/2K+1}^{T/K} w_t^{(i)}$

---

## 4   Sub-Optimality of SGD

In previous section, we presented information theoretic limits on the error rates of *any* method when applied to the general Markovian data, and presented algorithms that achieve these rates. However, in practice, SGD is the most commonly used method for learning problems. So, in this section, we specifically analyze the performance of constant step size SGD on Markovian data. Somewhat surprisingly, SGD shows a sub-optimal rates for both independent and agnostic noise settings.

---

**Algorithm 3** SGD with tail-averaging

---

**Require:** $T \in \mathbb{N}$ , samples $(X_1, Y_1), \ldots, (X_T, Y_T) \in \mathbb{R}^d \times \mathbb{R}$, step size $\alpha > 0$, initial point $w_1 \in \mathbb{R}^d$.
    **for** t in range $[1, T]$ **do**
        Set $w_{t+1} \leftarrow w_t - \alpha(X_t X_t^\mathsf{T} w_t - X_t Y_t)$ .
    **end for**
    **return** $\hat{w} \leftarrow \frac{1}{T - \lfloor T/2 \rfloor} \sum_{t=\lfloor T/2 \rfloor + 1}^{T} w_t$ .

---

**SGD with Constant Step Size is Asymptotically Biased in the Agnostic Noise Setting**    It is well known that when data is iid, the expected iterate of Algorithm 3, $\mathbb{E}[w_t]$ converges to $w^*$ as $t \to \infty$ in any noise setting. However, this does not necessarily hold when the data is Markovian. When the noise in each observation $n_t(X)$ depends on $X$, SGD with constant step size may yield iterates that are biased estimators of the parameter $w^*$ even as $t \to \infty$. In this case, even tail-averaging such as in Algorithm 3, cannot resolve this issue. See Appendix D.1 for the detailed proof. During the presentation at NeurIPS 2020, it was pointed out that a similar result is already known in literature ([4]) for the more general case of Ergodic data.

**Theorem 4.** *There exists a finite Markov chain* MC *with* $\tau_{\text{mix}}, \kappa < C$ $X_0 \sim \pi(\text{MC})$ *and* $X_1 \to X_2 \to \cdots \to X_T \sim$ MC, *SGD (Algorithm 3) run with any constant step size* $\alpha > 0$ *leads to a constant bias, i.e., for every $t$ large enough,*

$$\|\mathbb{E}[w_t] - w^*\| \geq c\alpha,$$

*where $w_t$ is the $t$-th step iterate of the SGD algorithm. (Where $c, C > 0$ are universal constants.)*

**SGD in the Independent Noise Setting is not Minimax Optimal (Appendix D.2)**    Let $\text{SGD}_\alpha$ be the SGD algorithm with step size $\alpha$ and tail averaging (Algorithm 3). For $X_1 \to \ldots \to X_T \sim \text{MC}_0$,

we denote the output of $SGD_\alpha$ corresponding to the data $D_0(T) := (X_t, Y_t)_{t=1}^T$ by $SGD_\alpha(D_0(T))$. We let $w_0^*$ to be the optimal parameter corresponding to the regression problem. We have the following lower bound.

**Theorem 5.** *For every $\tau_0, d \in \mathbb{N}$, there exists a finite state Markov Chain, $\mathrm{MC}_0$ and associated independent noise observation model (see Section 2.1) with points in $\mathbb{R}^d$, mixing time at most $\tau_0$ and $\|w_{\mathrm{MC}_0}^*\| \leq 1$, such that: $\mathbb{E}\mathcal{L}_{\mathrm{MC}_0}(SGD_\alpha(D_0(T))) - \mathcal{L}_{\mathrm{MC}_0}(w_0^*) \geq (1 - o_T(1))\frac{c'\alpha\tau_0\sigma^2 d}{T}$, where $c'$ is a universal constant and $o_T(1) \to 0$ exponentially in $T$ and $\sigma^2 = \mathbb{E}n_t^2$ is the noise variance.*

The above result shows that while SGD with constant step size and tail averaging is minimax optimal in the independent data setting, it's variance rate is $\tau_0$ factor sub-optimal in the setting of Markovian data and independent noise. It is also $\tau_0$ factor worse compared to the rate established in Theorem 3.

## 5 Experience Replay for Gaussian Autoregressive (AR) Dynamics

Previous two sections indicate that for worst case Markov chains, SGD-DD, despite wasting most of the samples, might be the best algorithm for Markovian data. This naturally seems quite pessimistic, as in practice, approaches like experience replay are popular [21]. In this section, we attempt to reconcile this gap by considering a restricted but practical Markov Chain (Gaussian AR chain) that is used routinely for time-series modeling [13] and intuitively seems quite related to the type of samples we can expect in reinforcement learning (RL) problems. Interestingly, even for this specific chain, we show that SGD's rates are no better than the SGD-DD method. On the other hand, an experience replay based SGD method (Algorithm 4) is able to give significantly faster rates, thus supporting it's usage in practice. More details and proofs are found in Section E.

Suppose our sample vectors $X \in \mathbb{R}^d$ are generated from a Markov chain (MC) with the following dynamics:
$$X_1 = G_1, \cdots, X_{t+1} = \sqrt{1 - \epsilon^2}X_t + \epsilon G_{t+1}, \cdots, \tag{6}$$
where $\epsilon$ is fixed and known, and each $G_j$ is independently sampled from $\frac{1}{\sqrt{d}}\mathcal{N}(0, \mathrm{I}_d)$. Each observation $Y_i = X_i^T w^* + \xi_i$, where the noise $\xi_i$ is independently drawn with mean 0 and variance $\sigma^2$. That is, every new sample in this MC is a random perturbation from a fixed distribution of the previous sample, which is intuitively similar to the sample generation process in RL.

The mixing time of this Markov chain is $\tau_{\mathrm{mix}} = \Theta\left(\frac{1}{\epsilon^2}\log(d)\right)$ (Lemma 19, Section E.1). Also, the covariance matrix of the stationary distribution is $\frac{1}{d}\mathrm{I}_d$, so the condition number of this chain is $\kappa = d$.

### 5.1 Lower Bound for SGD with Constant Step Size

We first establish a lower bound on the rate of bias decay for SGD with constant step size for this problem, which will help demonstrate that experience replay is effective in making SGD iterations more efficient.

**Theorem 6** (Lower Bound for SGD with constant step size for Gaussian AR Chain). $\mathcal{L}(w^{\mathrm{bias}}) \geq \Omega\left(\exp\left(\frac{-T\log(d)}{\kappa\tau_{\mathrm{mix}}}\right)\|w_0 - w^*\|^2\right)$. *Recall that $\kappa = d$ for MC in* (6)

The proof for this lemma involves carefully tracking the norm of the error at each iteration $\|w_t - w^*\|$. We show that the expected norm of the error contracts by a factor of at most $\frac{\epsilon^2}{d}$ in each iteration, therefore, we require $T = \Omega(\frac{d}{\epsilon^2})$ samples and iterations to get a $\delta$-approximate $w_T$. Note that the number of samples required here is $\Omega(\frac{d}{\epsilon^2}) = \Omega(\frac{\kappa\tau_{\mathrm{mix}}}{\log(d)})$. See Section E.4 for a detailed proof.

### 5.2 SGD with Experience Replay

We propose that the following interpretation of experience replay applied to SGD, which improves the dependence on $\tau_{\mathrm{mix}}$ on the rate of error decay.

Suppose we have a continuous stream of samples $X_1, X_2, \ldots X_T$ from the Markov Chain. We split the $T$ samples into $\frac{T}{S}$ separate buffers of size $S$ in a sequential manner, ie $X_1, \ldots X_S$ belong to the first buffer. Let $S = B + u$, where $B$ is orders of magnitude larger than $u$. From within each buffer, we drop the first $u$ samples. Then starting from the first buffer, we perform $B$ steps of SGD, where for

each iteration, we sample uniformly at random from within the $[u, B + u]$ samples in the first buffer. Then perform the next $B$ steps of SGD by uniformly drawing samples from within the $[u, B + u]$ samples in the second buffer. We will choose $u$ so that the buffers are are approximately i.i.d..

We run SGD this way for the first $\frac{T}{2S}$ buffers to ensure that the bias of each iterate is small. Then for the last $\frac{T}{2S}$ buffers, we perform SGD in the same way, but we tail average over the last iterate produced using each buffer to give our final estimate $w$. We formally write Algorithm 4.

---

**Algorithm 4** SGD with Experience Replay (SGD-ER)

---

**Require:** $(X_1, Y_1), \ldots (X_T, Y_T) \in \mathbb{R}^d$ sampled using (6), $\eta$: learning rate

$\quad w \sim \mathcal{N}(0, 1)$, $B \leftarrow \frac{1}{\epsilon^7}$, $u \leftarrow \max(\frac{2}{\epsilon^2} \log \frac{300000\pi dB}{\epsilon}, \frac{2}{\epsilon^2} \log \frac{300000\pi d^2 \sigma^6}{\epsilon^2 \delta})$, $S \leftarrow B + u$

$\quad$ **for** each buffer $j \in [0, \frac{T}{S} - 1]$ **do**

$\qquad$ Buffer$_j = [X_{(S \cdot j + 1)}, \ldots, X_{(S \cdot j + S)}]$

$\qquad$ **for** iterate in range[1, $B$] **do**

$\qquad\qquad w = w - \eta(Y_{Sj+i} - \langle X_{Sj+i}, w \rangle) X_{Sj+i}$ where $i \overset{unif}{\sim} [u, S]$

$\qquad$ **end for**

$\qquad$ Store $w_j \leftarrow w$

$\quad$ **end for**

$\quad$ **return** $\frac{2S}{T} \sum\limits_{j = \cdot T/2S + 1}^{\frac{T}{S}} w_j$ (i.e. average over last $T/2S$ buffers)

---

**Theorem 7** (SGD with Experience Replay for Gaussian AR Chain). *For any $\epsilon \le 0.21$, if $B \ge \frac{1}{\epsilon^7}$ and $d = \Omega(B^4 \log(\frac{1}{\beta}))$, with probability at least $1 - \beta$, Algorithm 4 returns $w$ such that $\mathbb{E}[\mathcal{L}(w)] \le O\left(\exp\left(\frac{-T \log(d)}{\kappa \sqrt{\tau_{\mathrm{mix}}}}\right) \|w_0 - w^*\|^2\right) + \tilde{O}\left(\frac{\sigma^2 d \sqrt{\tau_{\mathrm{mix}}}}{T}\right) + \mathcal{L}(w^*)$. Recall that $\kappa = d$ for MC in (6).*

*Proof.* $\mathbb{E}[\mathcal{L}(w)] = \mathbb{E}(\mathcal{L}(w^{\mathrm{bias}})) + \mathbb{E}(\mathcal{L}(w^{\mathrm{var}})) + \mathcal{L}(w^*)$. We give proof sketches for $\mathbb{E}(\mathcal{L}(w^{\mathrm{bias}})) \le O\left(\exp\left(\frac{-T \log(d)}{\kappa \sqrt{\tau_{\mathrm{mix}}}}\right) \|w_0 - w^*\|^2\right)$ and $\mathbb{E}(\mathcal{L}(w^{\mathrm{var}})) \le \tilde{O}\left(\frac{\sigma^2 d \sqrt{\tau_{\mathrm{mix}}}}{T}\right)$. Formal proofs are given in the appendix. $\qquad\square$

**Proof Sketch for Bias Decay (Proof in Section E.2):** Since the samples within the same buffer and across buffers are highly dependent, our algorithm drops the first $u$ samples from each batch. This ensures that across buffers the sampled points are approximately independent. This allows us to break down the problem into analyzing the progress made by one buffer, which we can then multiply $T/S$ times to get the overall bias bound. Lemma 21 formalizes this idea and upper bounds the expected contraction after every $B$ samples from a buffer $j$, by the expected contraction of a parallel process where the first vector in this buffer was sampled i.i.d. from $\mathcal{N}(0, \frac{1}{d}I)$.

The rest of the proof involves solving for expected rate of error decay when taking $B$ steps of SGD using samples generated from a single buffer in this parallel process. We write $H := \frac{1}{B} \sum\limits_{j=u+1}^{S} X_j X_j^T$, where $X_j$ are the vectors in the sampling pool. Lemma 22 establishes that the error in the direction of an eigenvector $v$, with associated eigenvalue $\lambda(v) \ge \frac{1}{B}$ in $H$ contracts by a factor of $\frac{1}{2}$ after $B$ rounds of SGD, while smaller eigenvalues in $H$ in the worst case do not contract. By spherical symmetry of eigenvectors, as long as the fraction of eigenvalues $\ge \frac{1}{B}$ is at least $\frac{\epsilon B}{d}$, and since we draw $B$ samples from each buffer, we see that the loss decays at a rate of $\exp(\frac{-T \epsilon}{d}) = \exp(\frac{-T \log(d)}{\kappa \sqrt{\tau_{\mathrm{mix}}}})$.

So, the key technical argument is to establish that $\frac{\epsilon B}{40\pi}$ of the eigenvalues of $H$ are larger than or equal to $\frac{1}{B}$, the overall proof structure is as follows. The non-zero eigenvalues of $H$ correspond directly to the non-zero eigenvalues of the gram matrix $M = \frac{1}{B} X^T X$, where the columns of $X$ are each $X_j$. We show that the Gram matrix can be written as $C + E$, where $C$ is a circulent matrix and $E$ is a small perturbation which can be effectively bounded when $d$ is large, i.e., $d = O(B^4)$. Using standard results about eigenvalues of $C$ along with Weyl's inequality [22] to handle perturbation $E$, we get a bound on the number of large-enough eigenvalues of $H$. The formal proof is in Section E.2.

**Proof Sketch for Variance Decay (Proof in Section E.3)**

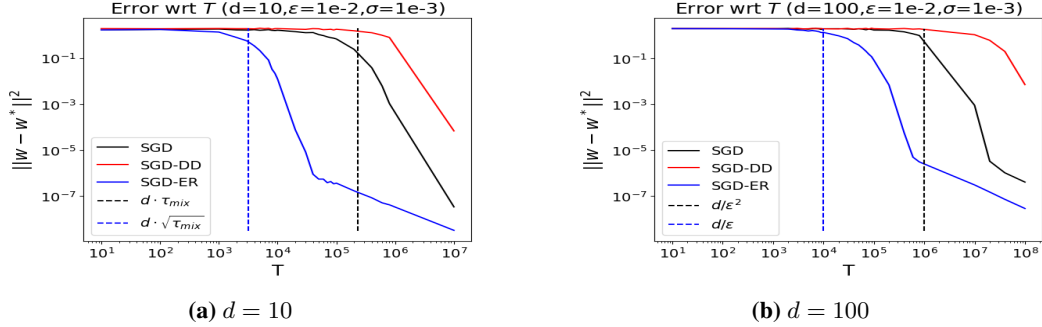

**(a)** $d = 10$            **(b)** $d = 100$

**Figure 1:** Gaussian AR Chain: error incurred by various methods

To analyze the variance, we start with $w_0^{\mathrm{var}} = w^*$, and based on the SGD update dynamics, consider the expected covariance matrix $\mathbb{E}[(w^{\mathrm{var}} - w^*)(w^{\mathrm{var}} - w^*)^T]$, where $w^{\mathrm{var}}$ is the tail-averaged value of the last iterate from every buffer, for the last $\frac{T}{2S}$ buffers. We show that for each iterate in the average, the covariance matrix $\mathbb{E}[(w_t^{\mathrm{var}} - w^*)(w_t^{\mathrm{var}} - w^*)^T] \preceq 3\sigma^2$. Next, we analyze the cross terms $\mathbb{E}[(w_i^{\mathrm{var}} - w^*)(w_j^{\mathrm{var}} - w^*)^T]$, which is approximately equal to $(I - H)^{j-i}\mathbb{E}[(w_i^{\mathrm{var}} - w^*)(w_i^{\mathrm{var}} - w^*)^T]$, when $j > i$ are buffer indices. This approximation is based on perfectly iid buffers, which we later correct by explicitly quantifying the worst case difference in the expected contraction of our SGD process and a parallel SGD process that does use perfectly iid buffers, (see Lemma 20). We use our earlier analysis of the eigenvalues of $H$ to arrive at our final rate.

**Simulations.** We also conducted experiments on data generated using Gaussian AR MC (6). We set $(d, \sigma, \epsilon) = (10, 1e - 3, 0.01)$ and choose the buffer size $B = 1/\epsilon^2$. We report results averaged over 100 runs. Figure 1a compare the estimation error achieved by SGD, SGD-DD, and the proposed SGD-ER method. Note that, as expected by our theorems, the decay regime starts at $d\sqrt{\tau_{\mathrm{mix}}}$ for SGD-ER and $d\tau_{\mathrm{mix}}$ for SGD which is similar to rate of SGD-DD. After about $50,000$ samples, SGD-ER's bias term becomes smaller than the variance term, hence we observe a straight line post that point. Also, according to Theorem 7 the variance at final point should be about $2\sigma^2 d^2/(\epsilon T) \approx 2e - 9$, which matches the empirically observed error. We present results for higher dimensions in the appendix.

We also run the experiment with parameter values $(d, \sigma, \epsilon) = (100, 1e - 3, 0.01)$ and buffer size $B = 1/\epsilon^2$. We report results averaged over 20 runs. Figure 1b compares the estimation error achieved by SGD, SGD-DD, and the proposed SGD-ER method. We obtain results which are similar to the $d = 10$ case.

## 6   Conclusion

In this paper, we obtain the fundamental limits of performance/minimax rates that are achievable in linear least squares regression problem with Markov chain data. Furthermore, we discuss algorithms that achieve these rates (SGD-DD and Parallel SGD). In the general agnostic noise setting, we show that any algorithm suffers by a factor of $\tau_{\mathrm{mix}}$ in both bias and variance, compared to the i.i.d. setting. In the independent noise setting, the minimax rate for variance can be improved to match that of the i.i.d. setting but standard SGD method with constant step size still suffers from a worse rate. Finally, we study a version of the popular technique 'experience replay' used widely for RL in the noiseless Gaussian AR setting and show that it achieves a significant improvement over the vanilla SGD with constant step size. Overall, our results suggest that instead of considering the general class of optimization problems with arbitrary Markov chain data (where things cannot be improved by much), it may be useful to identify and focus on important special cases of Markovian data, where novel algorithms with nontrivial improvements might be possible.

## Broader Impact

We build foundational theoretical groundwork for the fundamental problem of optimization with Markovian data. We think that our work sheds light on the possibilities and impossibilities in this space. For practitioners, our focus on the popular SGD algorithm provides them with a rigorously justified understanding of what SGD can achieve and for specially structured chains, experience replay with SGD can be provably helpful (though not in the general case). We also think that the proof techniques in this paper could impact future research in this space and beyond.

## Acknowledgments and Disclosure of Funding

G.B. and D.N. would like to acknowledge MIT-IBM Watson AI Lab and NSF CAREER award CCF-1940205 which partly funded this research. X.W. is partially supported by a fellowship from the Department of Management Science and Engineering at Stanford University. Part of this work was done while X.W. was an intern at Microsoft Research Bangalore.

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
