[Supplementary Material]

# A  Sharp Upper Bounds via. SGD-type Algorithms

## A.1  SGD with Data Drop for Agnostic Noise Setting

In this section, we modify SGD so that despite having constant step size, the algorithm converges to the optimal solution as $t \to \infty$ even if the noise in each observation $n_t(X)$ can depend on $X$. The modified algorithm is known as SGD with data drop (SGD-DD, Algorithm 1): fix $K \in \mathbb{N}$ and run SGD on samples $X_{Kr}$ for $r \in \mathbb{N}$, and ignore the other samples. Theorem 8 below shows that if $K = \Omega(\tau_{\mathrm{mix}} \log T)$, then the error is $O(\frac{\tau_{\mathrm{mix}} \log T}{T})$. Combined with the lower bounds in Theorems 2 and 1, this implies that SGD-DD is optimal up to log factors – in particular, the mixing time must appear in the rates. The analysis simply bounds the distance between the iterates of SGD with independent samples and the respective iterates of SGD-DD with Markovian samples.

We now formally describe the algorithm and result. Given samples from an exponentially ergodic finite state Markov Chain, MC with stationary distribution $\pi$ and mixing time $\tau_{\mathrm{mix}}$, for $T \in \mathbb{N}$ we obtain data $(X_t, Y_t)_{t=1}^T$ corresponding to the states of the Markov chain $X_1 \to \cdots \to X_T \sim \mathrm{MC}$. We pick $K = \tau_{\mathrm{mix}} \lceil L \log_2 T \rceil$ for some constant $L > 0$ to be fixed later. For the sake of simplicity we assume that $T/K$ is an integer.

We now present our theorem bounding the bias and variance for SGD-DD.

**Theorem 8** (SGD-DD). *Let $MC$ be any exponentially mixing ergodic finite state Markov Chain with stationary distribution $\pi$ and mixing time $\tau_{\mathrm{mix}}$. For $T \in \mathbb{N}$ we obtain data $(X_t, Y_t)_{t=1}^T$ corresponding to the states of the Markov chain $X_1 \to \cdots \to X_T \sim \mathrm{MC}$. Let $\alpha$ be small enough as given in Theorem 1 of [18]. Then*

$$\mathbb{E}[\mathcal{L}(\hat{w})] - \mathcal{L}(w^*) \leq \underbrace{\exp\left(\frac{-\alpha T}{C \cdot L \cdot \tau_{\mathrm{mix}} \kappa \log_2 T}\right) \|w_0 - w^*\|^2 + \frac{C \cdot L \cdot \tau_{\mathrm{mix}} \mathrm{Tr}\left(A^{-1}\Sigma\right) \log_2 T}{T}}_{\textit{Suboptimality for i.i.d. SGD with } T/K \textit{ samples}}$$

$$+ \underbrace{\frac{16\|w_0\|^2}{T^{L-2}} + \frac{16\alpha^2 v}{T^{L-3}}}_{\textit{error due to leftover correlations}},$$

*where $\hat{w}$ is the output of SGD-DD (Algorithm 1), $A := \mathbb{E}_{x \sim \pi}\left[xx^{\intercal}\right]$ is the data covariance matrix and $\Sigma := \mathbb{E}_{x \sim \pi}\left[n^2 xx^{\intercal}\right]$ is the noise covariance matrix.*

**Remarks**:

- The bound above has two groups of terms. The first group is the error achieved by SGD on i.i.d. samples and the second group is the error due to the fact that the samples we use are only *approximately* independent.
- With $L = 5$, the error is bounded by that of SGD on i.i.d. data plus a $O(1/T^2)$ term.

*Main ideas of the proof.* By Lemma 3 in Section B we can couple $(\tilde{X}_K, \tilde{X}_{2K}, \dots, \tilde{X}_T) \sim \pi^{\otimes(T/K)}$ to $(X_K, X_{2K}, \dots, X_T)$ such that:

$$\mathbb{P}\left(\left(\tilde{X}_K, \tilde{X}_{2K}, \dots, \tilde{X}_T\right) \neq (X_K, X_{2K}, \dots, X_T)\right) \leq \frac{T}{K}d(K) \leq \frac{T}{K}e^{-K/\tau_{\mathrm{mix}}}.$$

We call the data $\left(\tilde{X}_{tK}, Y_{tK}(\tilde{X}_{tK})\right)$ as $(\tilde{X}_{tK}, \tilde{Y}_{tK})$ for $t = 1, \dots, \frac{T}{K}$. We replace $(X_{tK}, Y_{tK})$ in the definition of SGD-DD with $(\tilde{X}_{tK}, \tilde{Y}_{tK})$ (with the exogenous, contextual noise $n_{tK}(\tilde{X}_{tK})$). We call the resulting iterates $\tilde{w}_t$. We can first show that $\mathbb{E}\|w_t - \tilde{w}_t\|^2$ is small and hence that the guarantees for SGD with i.i.d data, run for $T/K$ steps as given in [18] carry over to 'SGD with Data Drop' (Algorithm 1). We refer to Appendix D.3 for a detailed proof. $\qquad\square$

## A.2  Parallel SGD for Independent Noise Setting

We established in Section 4 that SGD with constant step size and averaging cannot achieve the minimax risk for least squares regression with Markovian data and independent noise [16], so we propose Parallel SGD algorithm with parallelization number $K \in \mathbb{N}$ to bridge the gap. For the sake of simplicity, let $\frac{T}{2K}$ be an integer.

In this algorithm, we run $K$ different SGD instances in parallel such that the $i$th instance of the algorithm observes $(X_{K(t-1)+i}, Y_{K(t-1)+i})$ for $t \geq 1$. Therefore, each parallel instance of SGD observes points which are $K$ time units apart and if $K \gg \tau_{\mathrm{mix}}$, the observations used by each of the SGD instance appear to be almost independent.

The following is the main result of this section.

**Theorem 9** (Parallel SGD). *Consider the Parallel SGD algorithm in the independent noise setting. Let the step size $\alpha < \frac{1}{2}$ and the number of parallel instances $K \geq \tau_{\mathrm{mix}} \lceil r \log_2(T) \rceil$ where $r > 5$. Assume $T/K$ is an integer. If $\hat{w}$ is the output of the algorithm using $T$ data points, we have for a universal constant $C > 0$ the bound:*

$$\mathbb{E}[\mathcal{L}(\hat{w})] - \mathcal{L}(w^*) \leq 2 \left(1 - \frac{\alpha}{2\kappa}\right)^{\frac{T}{2K}} \frac{1}{\tau_{\mathrm{mix}} \cdot \log T} \left[\sum_{i=1}^{K} \|w_1^{(i)} - w^*\|^2\right] + \frac{Cd\sigma^2}{T}.$$

Note that compared to the rate for SGD and SGD-DD (Section A.1), the variance term has no dependence on $\tau_{\mathrm{mix}}$. The bias decay is slower by a factor of $\tau_{\mathrm{mix}}$ compared to the i.i.d. data setting, but is optimal up to a logarithmic factor for the Markovian setting. A complete proof can be found in Appendix D.4.

# B    Coupling Lemmas

We give a well known characterization of total variation distance:

**Lemma 1.** *Let $\mu$ and $\nu$ be any two probability measures over a finite set $\Omega$. Then, there exist coupled random variables $(X, Y)$, that is random variables on a common probability space, such that $X \sim \mu$, $Y \sim \nu$ and,*

$$\mathbb{P}(X \neq Y) = \mathrm{TV}(\mu, \nu).$$

**Lemma 2.** *Let $X_0, \ldots, X_t, \ldots$ be a stationary finite state Markov chain $\mathrm{MC}$ with stationary distribution $\pi$. For arbitrary $r, s \in \mathbb{N}$, consider the following random variable:*

$$Y_{t,r,s} := (X_{t+r}, X_{t+r+1}, \ldots, X_{t+r+s}).$$

*Then, we have:*

$$\mathrm{TV}(\mathcal{D}(X_t, Y_{t,r,s}), \pi \otimes \mathcal{D}(Y_{t,r,s})) \leq \mathrm{d}_{\mathrm{mix}}(r),$$

*where $d(r)$ is the mixing metric as defined in Section 1.1.*

*Proof.* Using the fact that $X_t \sim \pi$ and by definition of total variation distance, we have:

$$\mathrm{TV}(\mathcal{D}(X_t, Y_{t,r,s}), \pi \otimes \mathcal{D}(Y_{t,r,s})) = \sum_{x \in \Omega} \pi(x) \mathrm{TV}(\mathcal{D}(Y_{t,r,s}|X_t = x), \mathcal{D}(Y_{t,r,s})).$$

By the Markov property, $\mathrm{TV}(\mathcal{D}(Y_{t,r,s}|X_t = x), \mathcal{D}(Y_{t,r,s})) = \mathrm{TV}(\mathcal{D}(X_{t+r}|X_t = x), \mathcal{D}(X_{t+r})) = \mathrm{TV}(\mathcal{D}(X_{t+r}|X_t = x), \pi)$. Lemma now follows from the definition of $\mathrm{d}_{\mathrm{mix}}(r)$. □

**Lemma 3.** *Let $X_0, \ldots, X_t, \ldots$ be a stationary finite state Markov chain $\mathrm{MC}$ with stationary distribution $\pi$. Let $K, n \in \mathbb{N}$. Then,*

$$\mathrm{TV}\left(\mathcal{D}(X_0, X_K, X_{2K}, \ldots, X_{nK}), \pi^{\otimes(n+1)}\right) \leq n\mathrm{d}_{\mathrm{mix}}(K).$$

*Furthermore, we can couple $(X_0, X_K, \ldots, X_{nK})$ and $(\tilde{X}_0, \tilde{X}_K, \ldots, \tilde{X}_{nK}) \sim \pi^{\otimes(n+1)}$ such that:*

$$\mathbb{P}\left((X_0, X_K, \ldots, X_{nK}) \neq (\tilde{X}_0, \tilde{X}_K, \ldots, \tilde{X}_{nK})\right) \leq n\mathrm{d}_{\mathrm{mix}}(K).$$

*Proof.* We prove this inductively. By Lemma 2, we have:

$$\mathrm{TV}(\mathcal{D}(X_{(n-1)K}, X_{nK}), \pi^{\otimes 2}) \leq \mathrm{d}_{\mathrm{mix}}(K).$$

From this it is easy to show that

$$\mathrm{TV}(\pi \otimes \mathcal{D}(X_{(n-1)K}, X_{nK}), \pi^{\otimes 3}) = \mathrm{TV}(\mathcal{D}(X_{(n-1)K}, X_{nK}), \pi^{\otimes 2}) \leq \mathrm{d}_{\mathrm{mix}}(K). \tag{7}$$

Using the notation in Lemma 2, we have:

$$\mathrm{TV}(\mathcal{D}(X_{(n-2)K}, Y_{(n-2)K,K,K}), \pi \otimes \mathcal{D}(Y_{(n-2)K,K,K})) \leq \mathrm{d}_{\mathrm{mix}}(K).$$

By elementary properties of TV distance, it is clear that the TV between the respective marginals is smaller than the TV between the given measures. Therefore,

$$\mathrm{TV}(\mathcal{D}(X_{(n-2)K}, X_{(n-1)K}, X_{nK}), \pi \otimes \mathcal{D}(X_{(n-1)K}, X_{nK}) \leq \mathrm{d}_{\mathrm{mix}}(K). \tag{8}$$

Using triangle inequality for TV distance along with (7) and (8), we have:

$$\mathrm{TV}(\mathcal{D}(X_{(n-2)K}, X_{(n-1)K}, X_{nK}), \pi^{\otimes 3}) \leq 2\mathrm{d}_{\mathrm{mix}}(K).$$

First part of the Lemma follows by using similar argument for all $i$, $1 \leq i \leq n$. The coupling part of the lemma then follows by Lemma 1. $\qquad\square$

## C  Minimax Lower Bounds: Proofs

We first note some well known and useful results about the square loss.

**Lemma 4.**       *1.* $\mathbb{E}_{x \sim \pi} \mathbb{E}[n_t(x) \cdot x] = 0$

2. $\mathcal{L}(w) - \mathcal{L}(w^*) = (w - w^*)^\mathsf{T} A (w - w^*)$

*Proof.*       1. This follows from the fact that $w^*$ is the minimizer of the square loss $\mathcal{L}(w)$ and hence $\nabla \mathcal{L}(w^*) = 0$.

2. Clearly, $\mathcal{L}(w) = w^\mathsf{T} A w + \mathbb{E}_{x \sim \pi} \mathbb{E}|Y_0(x)|^2 - 2\mathbb{E}_{x \sim \pi} \mathbb{E} Y_0(x) x^\mathsf{T} w$. The result follows after a simple algebraic manipulation involving item 1 above.

$\qquad\square$

### C.1  General Minimax Lower Bound for Bias Decay

**Proof sketch of Theorem 1**: The proof of Theorem 1 proceeds by considering a particular Markov chain and constructing a two point Bayesian lower bound. Let $\Omega = \{e_1, e_2\} \subset \mathbb{R}^2$ where $e_1, e_2 \in \mathbb{R}^2$ are the standard basis vectors. Let $\kappa \geq 2$ be given. Fix $\delta \leq (0, 1/2]$ and define $\epsilon = \frac{\delta}{\kappa - 1}$. Consider the Markov Chain $\mathrm{MC}_3$ defined by its transition matrix:

$$P_3 = \begin{bmatrix} P_3(e_1, e_1) & P_3(e_1, e_2) \\ P_3(e_2, e_1) & P_3(e_2, e_2) \end{bmatrix} = \begin{bmatrix} 1 - \epsilon & \epsilon \\ \delta & 1 - \delta \end{bmatrix} \tag{9}$$

Below given proposition shows that the mixing time $\tau_{\mathrm{mix}}^{(3)}$ of this Markov chain is bounded.

**Proposition 1.** $\tau_{\mathrm{mix}}^{(3)} \leq \frac{C}{\kappa \epsilon} \leq \frac{C}{\delta}$ *for some universal constant $C$.*

We use $\mathrm{MC}_3$ to generate a set of points. We note that if we start in $e_1$, with probability $\sim \left(1 - \frac{C}{\tau_0 \kappa_0}\right)^T$, we do not visit $e_2$ for the first $T$ time steps. In this event, the algorithm does not have any information about $\langle w^*, e_2 \rangle$, giving us the lower bound.

*Proof of Proposition 1.* We consider the metric:

$$\bar{\mathrm{d}}_{\mathrm{mix}}(t) := \sup_{i,j \in \Omega} \mathrm{TV}(P^t(i, \cdot), P^t(j, \cdot)).$$

Clearly, $\bar{\mathrm{d}}_{\mathrm{mix}}(1) = (1 - \frac{\delta \kappa}{\kappa - 1}) = (1 - \epsilon \kappa)$.

By Lemma 4.12 in [17], $\bar{\mathrm{d}}_{\mathrm{mix}}(t)$ is submultiplicative. Therefore, $\bar{\mathrm{d}}_{\mathrm{mix}}(t) \leq (1 - \epsilon \kappa)^t$. Now, by Lemma 4.11 in [17], we conclude that $\mathrm{d}_{\mathrm{mix}}(t) \leq \bar{\mathrm{d}}_{\mathrm{mix}}(t) \leq (1 - \epsilon \kappa)^t \leq e^{-t \epsilon \kappa}$. From this we conclude that $\tau_{\mathrm{mix}}^{(3)} \leq \frac{C}{\kappa \epsilon}$ for some universal constant $C$. $\qquad\square$

*Proof of Theorem 1.* Let the stationary distribution of $MC_3$ be $\pi_3$. We can easily show that $\pi_3(1) = \frac{\delta}{\delta+\epsilon} = 1 - \frac{1}{\kappa}$ and $\pi_3(2) = 1/\kappa$. Let $X_1, X_2, \ldots, X_T \sim \mathrm{MC}_3$. Consider the event $\mathcal{E}_T = \cap_{t=1}^T \{X_t \neq 2\}$. The event $\mathcal{E}_T$ holds if and only if the Markov chain starts in state 1 and remains in state 1 for the next $T$ transitions. Therefore,

$$\mathbb{P}(\mathcal{E}_T) = \pi_3(1) P(1,1)^{T-1} = \left(1 - \tfrac{1}{\kappa}\right)(1-\epsilon)^{T-1} \tag{10}$$

We will first consider the case $\tau_0 \geq 2C$ for the universal constant $C$ given in Proposition 1. Now, we give a two point Bayesian lower bound for the minimax error rate using the Markov chain $\mathrm{MC}_3$ defined above over the set $\{e_1, e_2\}$. Consider the following two observation models associated with the markov chain $\mathrm{MC}_3$ - which we denote with subscripts/ superscripts 1 and 2 respectively. Call these models $Q_1$ and $Q_2$. Let $w_1^*, w_2^* \in \mathbb{R}^2$ and set $w_1^* = e_2, w_2^* = -e_2$. For $k \in \{1,2\}$, and for a stationary sequence $X_1 \to \ldots \to X_T \sim \mathrm{MC}_3$, we obtain the data sequence $(X_t, Y_t^k) \in \mathbb{R}^2 \times R$. We let $Y_t^k = \langle X_t, w_k^* \rangle + \eta_t$ for any sequence of noise random variables considered in the class $\mathcal{Q}$. Now,

$$A_3 := \mathbb{E} X_t X_t^\mathsf{T} = \begin{bmatrix} 1 - \frac{1}{\kappa} & 0 \\ 0 & \frac{1}{\kappa} \end{bmatrix} \tag{11}$$

$A_3 \geq \frac{I}{\kappa}$ and $\kappa$ is the 'condition number'. We take $\kappa = \kappa_0$ and fix $\delta$ such that $\tau_{\mathrm{mix}}^{(3)} \leq \frac{C}{\kappa_0 \epsilon} = \tau_0$. Here $C$ is the universal constant given in Proposition 1. We see that the choice of $\delta$ above can be made using Proposition 1. Clearly, $Q_1, Q_2 \in \mathcal{Q}$.

From Lemma 4, it follows that for any $w \in \mathbb{R}^2$ and $k \in \{1,2\}$, we have

$$\mathcal{L}_{Q_k}(w) - \mathcal{L}_{Q_k}(w_k^*) = \frac{1}{\kappa} \| w - w_k^* \|^2 \,.$$

The following lower bound holds for the LHS of Equation (4):

$$\mathcal{L}(\mathcal{Q}) \geq \inf_{\mathrm{ALG} \in \mathcal{A}} \frac{1}{2\kappa} \mathbb{E} \|\mathrm{ALG}(D_{Q_1}(T)) - w_1^*\|^2 + \frac{1}{2\kappa} \mathbb{E} \|\mathrm{ALG}(D_{Q_2}(T)) - w_2^*\|^2 \tag{12}$$

Now, we can embed $D_{Q_1}(T)$ and $D_{Q_2}(T)$ into the same probability space such that the data is generated by the same sequence of states $X_0, \ldots, X_T$ and they have the same noise sequence $\eta_0, \ldots, \eta_T$ almost surely. It is easy to see that conditioned on the event $\mathcal{E}_T$ described above, $D_{Q_1}(T) = D_{Q_2}(T)$ almost surely. Under this event, $\mathrm{ALG}(D_{Q_1}(T)) = \mathrm{ALG}(D_{Q_2}(T))$. Using this in Equation (12), we concude:

$$\mathcal{L}(\mathcal{Q}) \geq \inf_{\mathrm{ALG} \in \mathcal{A}} \frac{1}{2\kappa} \mathbb{E} \|\mathrm{ALG}(D_{Q_1}(T)) - w_1^*\|^2 \mathbb{1}(\mathcal{E}_T) + \frac{1}{2\kappa} \mathbb{E} \|\mathrm{ALG}(D_{Q_2}(T)) - w_2^*\|^2 \mathbb{1}(\mathcal{E}_T)$$

$$= \inf_{\mathrm{ALG} \in \mathcal{A}} \frac{1}{2\kappa} \mathbb{E} \|\mathrm{ALG}(D_{Q_1}(T)) - w_1^*\|^2 \mathbb{1}(\mathcal{E}_T) + \frac{1}{2\kappa} \mathbb{E} \|\mathrm{ALG}(D_{Q_1}(T)) - w_2^*\|^2 \mathbb{1}(\mathcal{E}_T)$$

$$\geq \inf_{\mathrm{ALG} \in \mathcal{A}} \frac{\|w_1^* - w_2^*\|^2}{4\kappa} \mathbb{P}(\mathcal{E}_T) = \frac{\|w_1^* - w_2^*\|^2}{4\kappa} \mathbb{P}(\mathcal{E}_T) = \frac{\kappa - 1}{\kappa^2}(1-\epsilon)^{T-1}$$

$$\geq \frac{\kappa - 1}{\kappa^2}\left(1 - \frac{C}{\tau_0 \kappa}\right)^{T-1} = \frac{\kappa_0 - 1}{\kappa_0^2}\left(1 - \frac{C}{\tau_0 \kappa_0}\right)^{T-1} \geq \frac{\kappa_0 - 1}{\kappa_0^2}\left(1 - \frac{C}{\tau_0 \kappa_0}\right)^{T}. \tag{13}$$

In the third step above, we have used the fact that due to convexity of the map $a \to \|a\|^2$, we have $\|a - b\|^2 + \|c - b\|^2 \geq \frac{1}{2}\|a - c\|^2$ for arbitrary $a, b, c \in \mathbb{R}^d$ and Equation (10) in the sixth step and the choice of $\epsilon$ and $\delta$ in the seventh step and the choice of $\kappa = \kappa_0$ in the last step.

For the case $1 \leq \tau_0 < 2C$, we take $X_1 \to X_2 \ldots \to X_T$ to be an i.i.d seqence with distribution $\pi_3(\cdot)$ and let $\kappa = \kappa_0$. In this case, $\mathbb{P}(\mathcal{E}_T) = (1 - \frac{1}{\kappa})^T$ and its mixing time is 1. The lower bounds for this case follows using similar reasoning as above. We conclude that even when $1 \leq \tau_0 < 2C$, Equation (13) holds.

$\square$

## C.2 Minimax Lower Bound for Agnostic Setting

*Proof of Theorem 2.*

In the setting considered below, $\sigma^2 = c$ for some constant $c$. But, we note that we can achieve lower bounds for more general $\sigma^2$ by scaling $w^*$ and $Y_t$ below simultaenously by $\sigma$. (This would also require a scaling of the lower bound on $T$ below to ensure $\|w^*\| \leq 1$)

Let $\mathbf{I} = (I_1, \ldots, I_d) \in \{0, 1\}^d$. Let $\epsilon, \delta \in (0, 1)$ be such that $1/2 \geq \epsilon > \delta$. We consider a collection of irreducible Markov chains, indexed by $\{0, 1\}^d$ with a common state space $\Omega$ such that $|\Omega| = 2d$ and $\Omega \subset \mathbb{R}^d$. For now, we denote $\Omega = \{a_1, \ldots, a_{2d}\}$. We denote the Markov chain corresponding to $\mathbf{I}$ by $\mathrm{MC}_{\mathbf{I}}$, the corresponding transition matrix by $P_{\mathbf{I}}$, the stationary distribution by $\pi_{\mathbf{I}}$ and the mixing time by $\tau_{\mathrm{mix}}^{\mathbf{I}}$. Let

$$P_{\mathbf{I}}(a_i, a_j) = \begin{cases} 1 - \epsilon & \text{if } i = j, i \leq d \text{ and } I_i = 0 \\ 1 - \epsilon - \delta & \text{if } i = j, i \leq d \text{ and } I_i = 1 \\ \frac{\epsilon}{2d-1} & \text{if } i \neq j, i \leq d \text{ and } I_i = 0 \\ \frac{\epsilon+\delta}{2d-1} & \text{if } i \neq j, i \leq d \text{ and } I_i = 1 \\ 1 - \epsilon & \text{if } i = j \text{ and } i \geq d \\ \frac{\epsilon}{2d-1} & \text{if } i \neq j \text{ and } i \geq d \end{cases} \tag{14}$$

We consider the data model corresponding to each $\mathrm{MC}_{\mathbf{I}}$. For $i \in \{1, \ldots, d\}$, we take $a_i := e_i$ and $a_{d+i} := -e_i$ where $e_i$ is the standard basis vector in $\mathbb{R}^d$. We let the output corresponding to $a_i$, $Y_t(a_i) = 1$ almost surely for $i \in \{1, 2, \ldots, 2d\}$. Let $w_{\mathbf{I}}^* \in \mathbb{R}^d$ the optimum corresponding to regression problem described in Equation (2). A simple computation shows that:

$$w_{\mathbf{I}}^* = \arg\inf_{w \in \mathbb{R}^d} \sum_{i=1}^d \pi_{\mathbf{I}}(e_i)(\langle w, e_i \rangle - 1)^2 + \pi_{\mathbf{I}}(-e_i)(\langle w, e_i \rangle + 1)^2 \,.$$

Optimizing the RHS by setting the gradient to 0, we conclude that:

$$\langle w_{\mathbf{I}}^*, e_i \rangle = \frac{\pi_{\mathbf{I}}(e_i) - \pi_{\mathbf{I}}(-e_i)}{\pi_{\mathbf{I}}(e_i) + \pi_{\mathbf{I}}(-e_i)} \,.$$

It is clear from an application of Proposition 2 that:

$$\langle w_{\mathbf{I}}^*, e_i \rangle = \begin{cases} 0 & \text{if } I_i = 0 \\ -\frac{\delta}{2\epsilon+\delta} & \text{if } I_i = 1 \end{cases} \tag{15}$$

Denote $A_{\mathbf{I}} = \mathbb{E}_{x \sim \pi_{\mathbf{I}}} x x^\mathsf{T}$. It is easy to show that $A_{\mathbf{I}} \succeq \frac{I_d}{2d}$ from the identity given for $\pi_{\mathbf{I}}$ in Proposition 2. Let $Q_{\mathbf{I}}$ be the regression problem corresponding to $\mathrm{MC}_{\mathbf{I}}$. We define the data set $D_{Q_{\mathbf{I}}} = \{X_1^{(\mathbf{I})}, \ldots, X_T^{(\mathbf{I})}\}$.

We now consider the minimax error rate. In the equations below, we will denote $\mathrm{ALG}(D_{Q_{\mathbf{I}}}(T))$ by just $\hat{w}_{\mathbf{I}}$ for the sake of clarity. From Proposition 2, we conclude that if we take $1/\epsilon \sim \tau_0$ then $Q_{\mathbf{I}} \in \mathcal{Q}$ for every $\mathbf{I} \in \{0, 1\}^d$

$$\begin{aligned} \mathcal{L}(\mathcal{Q}) &= \inf_{\mathrm{ALG} \in \mathcal{A}} \sup_{Q \in \mathcal{Q}} \mathbb{E}[\mathcal{L}_Q(\mathrm{ALG}(D_Q(T)))] - \mathcal{L}_Q(w_Q^*) \\ &\geq \inf_{\mathrm{ALG} \in \mathcal{A}} \sup_{\mathbf{I} \in \{0,1\}^d} \mathbb{E}[\mathcal{L}_{Q_{\mathbf{I}}}(\hat{w}_{\mathbf{I}})] - \mathcal{L}_{Q_{\mathbf{I}}}(w_{\mathbf{I}}^*) \\ &= \inf_{\mathrm{ALG} \in \mathcal{A}} \sup_{\mathbf{I} \in \{0,1\}^d} \mathbb{E}(\hat{w}_{\mathbf{I}} - w_{\mathbf{I}}^*)^\mathsf{T} A_{\mathbf{I}}(\hat{w}_{\mathbf{I}} - w_{\mathbf{I}}^*) \\ &\geq \inf_{\mathrm{ALG} \in \mathcal{A}} \sup_{\mathbf{I} \in \{0,1\}^d} \frac{1}{2d} \mathbb{E}\|\hat{w}_{\mathbf{I}} - w_{\mathbf{I}}^*\|^2 \\ &\geq \inf_{\mathrm{ALG} \in \mathcal{A}} \mathbb{E}_{\mathbf{I} \sim \mathsf{Unif}\{0,1\}^d} \frac{1}{2d} \mathbb{E}\|\hat{w}_{\mathbf{I}} - w_{\mathbf{I}}^*\|^2 \\ &= \frac{1}{2d} \inf_{\mathrm{ALG} \in \mathcal{A}} \sum_{i=1}^d \mathbb{E}_{\mathbf{I} \sim \mathsf{Unif}\{0,1\}^d} \mathbb{E}|\langle \hat{w}_{\mathbf{I}}, e_i \rangle - \langle w_{\mathbf{I}}^*, e_i \rangle|^2, \end{aligned} \tag{16}$$

The third step follows by an application of Lemma 4. The fourth step follows from the fact that $A_{\mathbf{I}} \succeq \frac{1_d}{2d}$ as shown above. In the fourth and fifth steps, the inner expectation is with respect to the randomness in the data and the outer expectation is with respect to the randomness in $I \sim \mathsf{Unif}\{0,1\}^d$. We refer to Lemma 7, proved below, which essentially argues that whenever $\mathbf{I}, \mathbf{J} \in \{0,1\}^d$ are such that they differ only in one position, the outputs of $\mathrm{MC}_{\mathbf{I}}$ and $\mathrm{MC}_{\mathbf{J}}$ have similar distribution whenever $\delta$ is 'small enough' (as given in the lemma). Therefore, with constant probability, any given algorithm fails to distinguish between the data from the two Markov chains. Applying lemma 7 to Equation (16), we conclude that for some absolute constant $C, C_1, C_2$, whenever $T \geq C\frac{d^2}{\epsilon}$ and $\delta \leq C_1\sqrt{\frac{d\epsilon}{T}}$, we have $\|w_{\mathbf{I}}^*\| \leq 1$ and:

$$\sup_{\mathbf{I}\in\{0,1\}^d} \mathbb{E}\mathcal{L}_{\mathbf{I}}(\hat{w}_{\mathbf{I}}) - \mathcal{L}_{\mathbf{I}}(w_{\mathbf{I}}^*) \geq C_2 \frac{\delta^2}{\epsilon^2 + \delta^2} \tag{17}$$

The lower bounds follow from the equation above after noting that $\tau_0 \sim 1/\epsilon$

$\square$

The following proposition gives a uniform bound for the mixing times for the class of Markov chains and determines their stationary distributions considered in the proof of Theorem 2 above.

**Proposition 2.** $\tau_{\mathrm{mix}}^{\mathbf{I}} \leq \frac{C_0}{\epsilon}$ for some universal constant $C_0$. Let $|\mathbf{I}| := \sum_{i=1}^d I_i$. Then,

$$\pi_{\mathbf{I}}(a_i) = \begin{cases} \frac{\epsilon}{2d\epsilon + (2d-|\mathbf{I}|)\delta} & \text{if } i \leq d \text{ and } I_i = 1 \\ \frac{\epsilon+\delta}{2d\epsilon + (2d-|\mathbf{I}|)\delta} & \text{otherwise} \end{cases} \tag{18}$$

*Proof.* Consider the distance measure for mixing: $\bar{\mathrm{d}}_{\mathrm{mix}}(t) = \sup_{a,b\in\Omega} \mathrm{TV}(P_{\mathbf{I}}^t(a,\cdot), P_{\mathbf{I}}^t(b,\cdot))$. A simple calculation, using the fact that $1/2 \geq \epsilon > \delta$ shows that for any $\mathbf{I} \in \{0,1\}^d$, we have:

$$\bar{\mathrm{d}}_{\mathrm{mix}}(1) \leq 1 - \frac{\epsilon}{2}.$$

Using Lemma 4.12 in [17], we conclude that $\bar{\mathrm{d}}_{\mathrm{mix}}$ is submultiplicative and therefore, $\bar{\mathrm{d}}_{\mathrm{mix}}(t) \leq (1 - \frac{\epsilon}{2})^t$. By Lemma 4.11 in [17], $\mathrm{d}_{\mathrm{mix}}(t) \leq \bar{\mathrm{d}}_{\mathrm{mix}}(t) \leq (1 - \frac{\epsilon}{2})^t$. From this inequality, we conclude the result.

The identity for the stationary distribution follows from the definition. $\square$

Suppose $\mathbf{I}, \mathbf{J} \in \{0,1\}^d$ and that they differ only in one co-ordinate. Let $X_1^{(\mathbf{I})} \to X_2^{(\mathbf{I})} \to \ldots \to X_T^{(\mathbf{I})} \sim \mathrm{MC}_{\mathbf{I}}$ and $X_1^{(\mathbf{J})} \to X_2^{(\mathbf{J})} \to \ldots \to X_T^{(\mathbf{J})} \sim \mathrm{MC}_{\mathbf{J}}$ be stationary sequences. We will denote them as $\mathbf{X}^{(k)}$ for $k \in \{\mathbf{I}, \mathbf{J}\}$ respectively.

**Lemma 5.** *There exist universal constants $C, C_1$ such that whenever $T \geq C\frac{d}{\epsilon}$ and $\delta \leq C_1\sqrt{\frac{d\epsilon}{T}}$, we have*

$$\mathrm{TV}(\mathbf{X}^{(\mathbf{J})}, \mathbf{X}^{(\mathbf{I})}) \leq \frac{1}{2},$$

*where $\mathrm{TV}(\mathbf{X}^{(\mathbf{J})}, \mathbf{X}^{(\mathbf{I})})$ is the total variation distance between random variables $\mathbf{X}^{(\mathbf{J})}$ and $\mathbf{X}^{(\mathbf{J})}$.*

*Proof.* We will bound the total variation distance between $\mathbf{X}^{(\mathbf{J})}$ and $\mathbf{X}^{(\mathbf{J})}$ below by first bounding KL divergence between the sequences and then using Pinsker's inequality. Without loss of generality, we assume that $I_1 = 0$, $J_1 = 1$.

Let $(z_1, \ldots, z_T) \in \Omega^T$. Henceforth, we will denote this tuple by $\mathbf{z}$. For $k \in \{\mathbf{I}, \mathbf{J}\}$, we have:

$$\mathbb{P}(\mathbf{X}^{(k)} = \mathbf{z}) = \pi_k(z_1) \prod_{t=2}^T P_k(z_{t-1}, z_t). \tag{19}$$

Define the function $\eta_{ab}: \Omega^T \to \mathbb{N}$ for $a, b \in \Omega$ by: $\eta_{ab}(\mathbf{z}) = |\{2 \leq t \leq T : z_{t-1} = a \text{ and } z_t = b\}|$. $\eta_{ab}(\mathbf{z})$ counts the number of transitions from state $a$ to state $b$ in $\mathbf{z}$. Equation 19 can be rewritten using functions $\eta_{ab}$ as:

$$\mathbb{P}(\mathbf{X}^{(k)} = \mathbf{z}) = \pi_k(z_1) \prod_{a,b \in \Omega} P_k(a,b)^{\eta_{ab}(\mathbf{z})}.$$

Abusing notation to use $\mathbf{X}^{(k)}$ and $\mathcal{D}(\mathbf{X}^{(k)})$ interchangably, and by using definition of the KL divergence, we have:

$$\mathrm{KL}(\mathbf{X}^{(\mathbf{J})}||\mathbf{X}^{(\mathbf{I})}) = \sum_{\mathbf{z} \in \Omega^T} \mathbb{P}(\mathbf{X}^{(\mathbf{J})} = \mathbf{z}) \log \frac{\mathbb{P}(\mathbf{X}^{(\mathbf{J})} = \mathbf{z})}{\mathbb{P}(\mathbf{X}^{(\mathbf{I})} = \mathbf{z})}$$

$$= \sum_{\mathbf{z} \in \Omega^T} \mathbb{P}(\mathbf{X}^{(\mathbf{J})} = \mathbf{z}) \left[ \log \left( \frac{\pi_{\mathbf{J}}(z_1)}{\pi_{\mathbf{I}}(z_1)} \right) + \sum_{a,b \in \Omega} \eta_{ab}(\mathbf{z}) \log \frac{P_{\mathbf{J}}(a,b)}{P_{\mathbf{I}}(a,b)} \right]$$

$$= \mathrm{KL}(\pi_{\mathbf{J}}||\pi_{\mathbf{I}}) + \sum_{j=1}^{d} \mathbb{E}\eta_{a_1 a_j}(\mathbf{X}^{(2)}) \log \frac{P_{\mathbf{J}}(a_1, a_j)}{P_{\mathbf{I}}(a_1, a_j)}$$

$$= \mathrm{KL}(\pi_{\mathbf{J}}||\pi_{\mathbf{I}}) + (T-1)\pi_{\mathbf{J}}(a_1)\mathrm{KL}(P_{\mathbf{J}}(a_1, \cdot)||P_{\mathbf{I}}(a, \cdot)) \qquad (20)$$

In the third step we have used that fact that $P_{\mathbf{J}}(a,b) \neq P_{\mathbf{I}}(a,b)$ only when $a = a_1$ since $\mathbf{J}$ and $\mathbf{I}$ differ only in the first co-ordinate. In the fourth step we have used the fact that $\mathbb{E}\eta_{a_1 a_j}(\mathbf{X}^{(2)}) = (T-1)\pi_{\mathbf{J}}(a_1)P_{\mathbf{J}}(a_1, a_j)$

For any two probability measures $P$ and $Q$ on the same finite space, the following holds by the Pinsker's inequality:

$$\mathrm{TV}(P,Q) \leq \sqrt{2\mathrm{KL}(P||Q)} \qquad (21)$$

We now state the 'reverse Pinkser's inequality' to bound the KL divergence.

**Lemma 6.** *[Lemma 6.3 in [22]] Let $P_1$ and $P_2$ be probability distributions over some finite space $E$. Then,*

$$\mathrm{KL}(P_2||P_1) \leq \sum_{a \in E} \frac{|P_2(a) - P_1(a)|^2}{P_1(a)} .$$

*In particular, when $E = \{0, 1\}$, $P_i = \mathrm{Ber}(p_i)$, we have:*

$$\mathrm{KL}(P_2||P_1) \leq \frac{|p_1 - p_2|^2}{p_1(1 - p_1)}$$

An easy computation using Lemma 6 shows that for some universal constant $C_3$:

$$\mathrm{KL}(\pi_{\mathbf{J}}||\pi_{\mathbf{I}}) \leq \frac{C_3 \delta^2}{d\epsilon^2} .$$

By Proposition 2, we have $\pi_{\mathbf{J}}(a_1) \leq 1/2d$. By a similar application of Lemma 6 we have:

$$\mathrm{KL}(P_{\mathbf{J}}(a_1, \cdot)||P_{\mathbf{I}}(a_1, \cdot)) \leq \frac{\delta^2}{\epsilon(1 - \epsilon)}$$

Combining these bounds with Equation (20) and using the fact that $\epsilon < 1/2$, we have, for some universal constant $C_3$,

$$\mathrm{KL}(\mathbf{X}^{(\mathbf{J})}||\mathbf{X}^{(\mathbf{I})}) \leq C_3 \left[ \frac{\delta^2}{d\epsilon^2} + \frac{T\delta^2}{d\epsilon} \right] \qquad (22)$$

We let $T \geq Cd/\epsilon$ and take $\delta \leq C_1 \sqrt{\frac{d\epsilon}{T}}$ for appropriate constants $C, C_1$. Applying this in Equation (22) and then using Equation (21), we obtain the desired result. $\qquad \square$

**Lemma 7.** $T \geq \frac{Cd}{\epsilon}$ and $\delta \leq C_1\sqrt{\frac{d\epsilon}{T}}$ For any output $\hat{w}_\mathbf{I}$ (as decribed in the proof of Theorem 2),

$$\mathbb{E}_{\mathbf{I}\sim\mathsf{Unif}\{0,1\}^d}\mathbb{E}|\langle\hat{w}_\mathbf{I}, e_i\rangle - \langle w_\mathbf{I}^*, e_i\rangle|^2 \geq \frac{\delta^2}{8(2\epsilon+\delta)^2}$$

*Proof.* Let $\mathbf{I}_{\sim i}$ denote all the co-ordinates of $\mathbf{I}$ other than $i$, let $\mathbf{I}_i^+ \in \{0,1\}^d$ be such that its $i$-th co-ordinate is 1 and the rest of the co-ordinates are $\mathbf{I}_{\sim i}$. Similarly $\mathbf{I}_i^-$ be such that its $i$-th co-ordinate is 0 and the rest of the co-ordinates are $\mathbf{I}_{\sim i}$.

By Lemma 1 and Lemma 5, we conclude that whenever $T \geq \frac{Cd}{\epsilon}$ and $\delta \leq C_1\sqrt{\frac{d\epsilon}{T}}$ we can couple the sequences $\mathbf{X}^{(\mathbf{I}_i^+)}$ and $\mathbf{X}^{(\mathbf{I}_i^-)}$ such that:

$$\mathbb{P}(\{\mathbf{X}^{(\mathbf{I}_i^+)} = \mathbf{X}^{(\mathbf{I}_i^-)}\}) \geq \frac{1}{2},.$$

Define the event $\mathcal{E}'_T := \{D_{Q_1}(T) = D_{Q_2}(T)\}$.

$$\mathbb{E}_{\mathbf{I}\sim\mathsf{Unif}\{0,1\}^{d-1}}\mathbb{E}|\langle\hat{w}_\mathbf{I}, e_i\rangle - \langle w_\mathbf{I}^*, e_i\rangle|^2 = \mathbb{E}_{\mathbf{I}_{\sim i}\sim\mathsf{Unif}\{0,1\}^{d-1}}\mathbb{E}_{I_i\sim\mathsf{Unif}\{0,1\}}\mathbb{E}|\langle\hat{w}_\mathbf{I}, e_i\rangle - \langle w_\mathbf{I}^*, e_i\rangle|^2$$

$$= \mathbb{E}_{\mathbf{I}_{\sim i}\sim\mathsf{Unif}\{0,1\}^{d-1}}\frac{1}{2}\left[\mathbb{E}|\langle\hat{w}_{\mathbf{I}_i^+}, e_i\rangle - \langle w_{\mathbf{I}_i^+}^*, e_i\rangle|^2 + \mathbb{E}|\langle\hat{w}_{\mathbf{I}_i^-}, e_i\rangle - \langle w_{\mathbf{I}_i^-}^*, e_i\rangle|^2\right]$$

$$\geq \frac{1}{2}\mathbb{E}_{\mathbf{I}_{\sim i}\sim\mathsf{Unif}\{0,1\}^{d-1}}\mathbb{E}\left[|\langle\hat{w}_{\mathbf{I}_i^+}, e_i\rangle - \langle w_{\mathbf{I}_i^+}^*, e_i\rangle|^2 + |\langle\hat{w}_{\mathbf{I}_i^-}, e_i\rangle - \langle w_{\mathbf{I}_i^-}^*, e_i\rangle|^2\right]\mathbb{1}(\mathcal{E}'_T)$$

$$\geq \frac{1}{4}\mathbb{E}_{\mathbf{I}_{\sim i}\sim\mathsf{Unif}\{0,1\}^{d-1}}\left[|\langle w_{\mathbf{I}_i^-}^*, e_i\rangle - \langle w_{\mathbf{I}_i^+}^*, e_i\rangle|^2\right]\mathbb{P}(\mathcal{E}'_T)$$

$$= \frac{\delta^2}{8(2\epsilon+\delta)^2} \tag{23}$$

In the fourth step we have used the fact that in the event $\mathcal{E}'_T$, that is when $\mathbf{X}^{(\mathbf{I}_i^+)} = \mathbf{X}^{(\mathbf{I}_i^-)}$, the corresponding outputs of the algorithm are the same. That is $\hat{w}_\mathbf{I}^+ = \hat{w}_\mathbf{I}^-$. We have also used the convexity of the map $x \to \|x\|^2$ to show that $\|a-b\|^2 + \|b-c\|^2 \geq \frac{\|a-c\|^2}{2}$. In the last step, we have used Equation (15). $\qquad\square$

# D  SGD algorithms: Proofs

## D.1  SGD with Constant Step Size suffers Asymptotic Bias in the Agnostic Setting

*Proof of Theorem 4.* Fix $\epsilon \in (0,1)$. We describe the Markov chain $\mathrm{MC}_1$ over the space $\Omega = \{a,b\} \subset \mathbb{R}$ and the corresponding data model that we consider. Let the corresponding stationary distribution be $\pi_1$, mixing time be $\tau_{\mathrm{mix}}^1$ and the transition matrix be $P_1$, given by:

$$P_1(a,a) = P_1(b,b) = 1 - \epsilon, P_1(a,b) = P_1(b,a) = \epsilon.$$

It is clear from Proposition 2 with $d = 1$ and $\delta = 0$ that $\tau_{\mathrm{mix}}^1 \leq C/\epsilon$ for some universal constant $C$ and that the stationary distribution is uniform over $\Omega$. We set $a = 1/2$ and $b = -1$. The output $Y_t(a) = Y_t(b) = 1/2$ almost surely. It is easy to show that the corresponding optimal parameter $w_1^* = -\frac{1}{5}$. Let the $\mathrm{SGD}_\alpha$ be run on an instance using the data from $\mathrm{MC}_1$ as described above. We call the iterates $w_t$.

We will first bound the Wasserstein distance between $w_{t+1}$ and $w_t$. Let $X_1 \to X_2 \to \ldots \to X_T \ldots \sim \mathrm{MC}_1$ be a stationary sequence. We consider another stationary sequence $\tilde{X}_1 \to \tilde{X}_2 \to \ldots \sim \mathrm{MC}_1$ such that $\tilde{X}_t = X_{t+1}$ for every $t \geq 1$ almost surely. We can run the SGD with data from the chain $X_t$ or from the chain $\tilde{X}_t$. Let the data corresponding to $\tilde{X}_t$ be $(\tilde{X}_t, \tilde{Y}_t)$. We let the iterates be $w_t$ and $\tilde{w}_t$ respectively and start both from the same initial point $w_1$. Now, $w_t$ and $\tilde{w}_t$ are identically distributed. For $t \geq 1$, consider:

$$w_{t+2} - \tilde{w}_{t+1} = w_{t+1} - \tilde{w}_t - \alpha(X_{t+1}X_{t+1}^\intercal w_{t+1} - X_{t+1}Y_{t+1}) + \alpha(\tilde{X}_t\tilde{X}_t^\intercal w_t - \tilde{X}_t\tilde{Y}_t)$$

Clearly, $\tilde{X}_t = X_{t+1}$ and $\tilde{Y}_t = Y_{t+1}$ almost surely. Hence,

$$w_{t+2} - \tilde{w}_{t+1} = (1 - \alpha\tilde{X}_t\tilde{X}_t^\intercal)(w_{t+1} - \tilde{w}_t).$$

Now, $\tilde{X}_t \in \mathbb{R}$ and $|\tilde{X}_t|^2 \geq \frac{1}{4}$ almost surely. Therefore, when $\alpha \in (0,1)$ we have:

$$|w_{t+2} - \tilde{w}_{t+1}|^2 \leq \left(1 - \frac{\alpha}{4}\right)^2 |w_{t+1} - \tilde{w}_t|^2. \tag{24}$$

Applying the above inequality for $t$ iterations and by applying expectation on both sides: Therefore we conclude that:

$$\mathbb{E}\left[|w_{t+1} - \tilde{w}_t|^2\right] \leq e^{-t\alpha/2}\mathbb{E}\left[|w_1 - w_2|^2\right] = e^{-t\alpha/2}|w_2|^2 \leq e^{-(t-1)\alpha/2}.$$

Applying Jensen's inequality to the LHS and using the fact that $\tilde{w}_t$ has the same distribution as $w_t$, we get:

$$|\mathbb{E}w_{t+1} - \mathbb{E}w_t| \leq \frac{\alpha}{2}e^{-(t-1)\alpha/4}.$$

Similarly, since Equation (24) holds almost surely, we have for $k \in \{1,2\}$:

$$\mathbb{E}\left[|w_{t+2} - \tilde{w}_{t+1}|^2 \Big| X_{t+1} = k\right] \leq \frac{\alpha}{2}e^{-(t-1)\alpha/2}. \tag{25}$$

Using the fact that $X_{t+1} = \tilde{X}_t$ almost surely, we conclude that:

$$\mathbb{E}\left[\tilde{w}_{t+1}|X_{t+1} = k\right] = \mathbb{E}\left[\tilde{w}_{t+1}|\tilde{X}_t = k\right] = \mathbb{E}\left[w_{t+1}|X_t = k\right].$$

Using the equation above and applying Jensen's inequality to Equation (25), we obtain:

$$\left|\mathbb{E}\left[w_{t+1}|X_t = k\right] - \mathbb{E}\left[w_t|X_{t-1} = k\right]\right| \leq \frac{\alpha}{2}e^{-(t-1)\alpha/4}. \tag{26}$$

For the sake of simplicity, we will denote $\mathbb{E}\left[w_t|X_{t-1} = k\right]$ by $e_k$ and $\mathbb{E}\left[w_{t+1}|X_t = k\right]$ by $e_k + \lambda_k$, where $|\lambda_k| \leq \frac{\alpha}{2}e^{-\alpha(t-1)/4}$. We hide the dependence on $t$ for the sake of clarity.

Taking conditional expectation with respect to the event $X_t = 1$ in the recursion in Algorithm 3, we have:

$$e_1 + \lambda_1 = \left(1 - \frac{\alpha}{4}\right)\mathbb{E}\left[w_t|X_t = 1\right] + \frac{\alpha}{4}. \tag{27}$$

Now, consider:

$$\mathbb{E}\left[w_t|X_t = 1\right] = 2\mathbb{E}\left[w_t \mathbb{1}(X_t = 1)\right] = 2\mathbb{E}\left[w_t \mathbb{1}(X_t = 1)\mathbb{1}(X_{t-1} = 1) + w_t \mathbb{1}(X_t = 1)\mathbb{1}(X_{t-1} = 2)\right]$$
$$= \mathbb{E}\left[w_t \mathbb{1}(X_t = 1)\big|X_{t-1} = 1\right] + \mathbb{E}\left[w_t \mathbb{1}(X_t = 1)\big|X_{t-1} = 2\right] = (1 - \epsilon)e_1 + \epsilon e_2.$$

Using this in Equation (27), we conclude:

$$e_1 + \lambda_1 = (1 - \alpha/4)\left[(1 - \epsilon)e_1 + \epsilon e_2\right] + \alpha/4$$

Similarly, we have:

$$e_2 + \lambda_2 = (1 - \alpha)\left[\epsilon e_1 + (1 - \epsilon)e_2\right] - \alpha/2$$

Using the above two equations, we have

$$\begin{bmatrix} \alpha/4 + \epsilon - \alpha\epsilon/4 & -\epsilon(1 - \alpha/4) \\ -(1-\alpha)\epsilon & \alpha + \epsilon - \alpha\epsilon \end{bmatrix}\begin{bmatrix} e_1 \\ e_2 \end{bmatrix} = \begin{bmatrix} \alpha/4 + \lambda_1 \\ -\alpha/2 + \lambda_2 \end{bmatrix} \tag{28}$$

Solving the equations above we get:

$$\begin{bmatrix} e_1 \\ e_2 \end{bmatrix} = \begin{bmatrix} \frac{\alpha/2 - \alpha\epsilon/4 - \epsilon/2}{\alpha/2 + 5\epsilon/2 - \alpha\epsilon} \\ \frac{-\alpha/4 - \alpha\epsilon/4 - \epsilon/2}{\alpha/2 + 5\epsilon/2 - \alpha\epsilon} \end{bmatrix} + O(C(\alpha, \epsilon)e^{-t\alpha/4}). \tag{29}$$

As $\mathbb{E}[w_t] = \frac{e_1 + e_2}{2}$,

$$\mathbb{E}[w_t] = \frac{1}{2}\left[\frac{\alpha/4 - \alpha\epsilon/2 - \epsilon}{\alpha/2 - \alpha\epsilon + 5\epsilon/2}\right] + O(C(\alpha, \epsilon)e^{-t\alpha/4}). \tag{30}$$

It is easy to check that when $\epsilon = 1/2$, $X_0, X_1, \ldots,$ is infact a sequence of i.i.d $\mathrm{Ber}(1/2)$ random variables and we'd expect $w_t$ to be an unbiased estimator as $t \to \infty$. This can be verified by plugging in $\epsilon = 1/2$ in Equation (30). When $\epsilon = 1/4$, the corresponding value becomes $\frac{1}{2}\frac{\alpha-2}{2\alpha+5} + o_t(1)$, which does not tend to $w_1^* = -1/5$ as $t \to \infty$. $\qquad\square$

## D.2 A Lower Bound for SGD with Constant Step Size in the Independent Noise Setting

*Proof of Theorem 5.* Recall the class of Markov chains $\mathrm{MC_I}$ for $\mathbf{I} \in \{0,1\}^d$ defined in the proof of Theorem 2 in Appendix C.2. We consider a similar Markov chain $\mathrm{MC_0}$ with state space $\Omega = \{e_1, \ldots, e_d\}$. Let its transition matrix be $P_0$, the stationary distribution be $\pi_0$ and the mixing time be $\tau_{\mathrm{mix}}^0$. Let $0 < \epsilon < 1/2$. We define:

$$P_0(e_i, e_j) = \begin{cases} 1 - \epsilon & \text{if } i = j \\ \frac{\epsilon}{d-1} & \text{if } i \neq j \end{cases} \tag{31}$$

Through steps analogous to the proof of Proposition 2, we can show that $\tau_{\mathrm{mix}}^0 \leq \frac{C}{\epsilon}$ for some universal constant $C$. It follows from definitions that $\pi_0$ is the uniform distribution over $\Omega$.

Let $X_1 \to X_2 \ldots \to X_T \sim \mathrm{MC}^0$ is a stationary seqence. We let $w_0^* = 0$ and let the output be $Y_t = \langle X_t, w_0^* \rangle + n_t = n_t$ such that $n_t \sim \mathcal{N}(0, \sigma^2)$. Since this is the independent noise case, $n_t$ is taken to be i.i.d. and independent of $X_t$. The matrix $A_0 = \mathbb{E} X_t X_t^\mathsf{T} = \frac{\mathrm{I}_d}{d}$. Consider the SGD algorithm with iterate averaged output which achieves the information theoretically optimal rates in the i.i.d data case. Suppose $(X_t, Y_t)_{t=1}^T$ is drawn from the model associated with $\mathrm{MC_0}$ described above. The evolution equations become:

$$\langle w_{t+1}, e_i \rangle = \begin{cases} \langle w_t, e_i \rangle & \text{if } X_t \neq e_i \\ (1 - \alpha)\langle w_t, e_i \rangle + \alpha n_t & \text{if } X_t = e_i \end{cases} \tag{32}$$

Let the averaged output of SGD be $\hat{w} := \frac{2}{T} \sum_{t=T/2+1}^T w_t$. Now we will directly give a lower bound for the excess loss of the estimator $\hat{w}$ for $w_0^*$. We let $w_0 = 0$. For the problem under consideration, $w_0^* = 0$ and $A_0 = \frac{\mathrm{I}_d}{d}$. Therefore, using Lemma 4

$$\begin{aligned} \mathbb{E}\mathcal{L}(\hat{w}) - \mathcal{L}(w_0^*) &= \frac{1}{d}\mathbb{E}\|\hat{w}\|^2 \\ &= \frac{4}{T^2 d} \sum_{t,s=T/2+1}^T \mathbb{E}\langle w_t, w_s \rangle \\ &= \frac{4}{T^2 d} \sum_{t,s=T/2+1}^T \sum_{i=1}^d \mathbb{E}\langle w_t, e_i \rangle \langle w_s, e_i \rangle \end{aligned} \tag{33}$$

Consider the case $s > t$. For $i \in \{1, \ldots, d\}$, let $N_i(s-1,t) := |\{t \leq l \leq s-1 : X_l = e_i\}|$ and let $t \leq t_1 < \ldots < t_{N_i(s-1,t)} \leq s-1$ be the sequence of times such that $X_{t_p} = e_i$. We have $\langle w_s, e_i \rangle = (1-\alpha)^{N_i(s-1,t)}\langle w_t, e_i \rangle + \sum_{p=1}^{N_i(t,s)}(1-\alpha)^{N_i(t,s)-p}\alpha n_{t_p}$. Therefore, multiplying by $\langle w_t, e_i \rangle$ on both sides and taking expectation, we conclude:

$$\begin{aligned} \mathbb{E}\langle w_t, e_i \rangle \langle w_s, e_i \rangle &= \mathbb{E}(1-\alpha)^{N_i(s-1,t)}|\langle w_t, e_i \rangle|^2 \\ &\geq \sum_{j \neq i} \mathbb{E}\left[|\langle w_t, e_i \rangle|^2 \big| X_{t-1} = j, N_i(s-1,t) = 0\right] \mathbb{P}\left(X_{t-1} = j, N_i(s-1,t) = 0\right) \\ &= \sum_{j \neq i} \mathbb{E}\left[|\langle w_t, e_i \rangle|^2 \big| X_{t-1} = j\right] \mathbb{P}\left(X_{t-1} = j, N_i(s-1,t) = 0\right) \\ &= \sum_{j \neq i} \mathbb{E}\left[|\langle w_t, e_i \rangle|^2 \big| X_{t-1} = j\right] \mathbb{P}\left(N_i(s-1,t) = 0 \big| X_{t-1} = j\right) \mathbb{P}(X_{t-1} = j) \\ &= \sum_{j \neq i} \frac{(1 - \frac{\epsilon}{d-1})^{s-t}}{d} \mathbb{E}\left[|\langle w_t, e_i \rangle|^2 \big| X_{t-1} = j\right] \end{aligned} \tag{34}$$

The first equality follows from fact that $n_l$ are i.i.d mean 0 and independent of $X_l$. In the second step we have used the fact that conditioned on the event $N_2(s-1,t) = 0$, $(1-\alpha)^{N_2(s-1,t)} = 1$. In the third step we have used the fact that $w_t$ depends only on $X_1, \ldots, X_{t-1}$, and $n_1, \ldots, n_{t-1}$ and

$N_2(s-1,t)$ depends only on $X_t, \ldots, X_{s-1}$ and therefore are conditionally independent given $X_{t-1}$. The last step follows from the fact that $\mathbb{P}\left(N_i(s-1,t) = 0 \big| X_{t-1} = j\right) = (1 - \frac{\epsilon}{d-1})^{s-t}$.

Using Equation (34) in Equation (33), we have:

$$
\begin{aligned}
\mathbb{E}\mathcal{L}(\hat{w}) - \mathcal{L}(w^*) &\geq \frac{2}{T^2 d^2} \sum_{t=T/2+1}^{T} \sum_{s=t}^{T} \sum_{i=1}^{d} \sum_{j \neq i} (1 - \tfrac{\epsilon}{d-1})^{s-t} \mathbb{E}\left[|\langle w_t, e_i \rangle|^2 \big| X_{t-1} = j\right] \\
&= \frac{2(d-1)}{T^2 d^2 \epsilon} \sum_{t=T/2+1}^{T} \sum_{i=1}^{d} \sum_{j \neq i} \left(1 - (1 - \tfrac{\epsilon}{d-1})^{T-t+1}\right) \mathbb{E}\left[|w_t|^2 \big| X_{t-1} = 0\right] \\
&\geq \frac{2(1 - (1 - \tfrac{\epsilon}{d-1})^{T/4})(d-1)}{d^2 T^2 \epsilon} \sum_{t=T/2+1}^{3T/4} \sum_{i=1}^{d} \sum_{j \neq i} \mathbb{E}\left[|\langle w_t, e_i \rangle|^2 \big| X_{t-1} = j\right] \\
&\geq \frac{c \alpha \sigma^2 (d-1)^2}{T d \epsilon (2 - \alpha)} \left[1 - O\left((1 - \tfrac{\epsilon}{d-1})^{T/4} + d(1-\alpha)^{T/2d} + d e^{-T/36 d^2 \tau_{\mathrm{mix}}^0}\right)\right] \\
&\geq \frac{c' \alpha \tau_{\mathrm{mix}}^0 \sigma^2 d}{T(2 - \alpha)} \left[1 - O\left((1 - \tfrac{\epsilon}{d-1})^{T/4} + d(1-\alpha)^{T/2d} + d e^{-T/36 d^2 \tau_{\mathrm{mix}}^0}\right)\right]
\end{aligned}
\tag{35}
$$

Where $c$ in the third step is some positive universal constant. In the third step we have used Lemma 8. In the last step we have used the bounds on $\tau_{\mathrm{mix}}^0$. This establishes the lower bound. $\qquad \square$

**Lemma 8.** *For $j \in \{1, \ldots, d\}$,*

$$
\frac{\alpha \sigma^2}{2 - \alpha}\left(1 - d(1-\alpha)^{\frac{t}{d}} - d e^{-\frac{t}{72 d^2 \tau_{\mathrm{mix}}^0}}\right) \leq \mathbb{E}\left[|\langle e_i, w_{t+1} \rangle|^2 \big| X_t = e_j\right] \leq \frac{\alpha \sigma^2}{2 - \alpha}.
$$

*Proof.* It is clear from Equation (32) that

$$
\langle w_{t+1}, e_i \rangle = \sum_{s=1}^{N_i(t)} (1-\alpha)^{N_i(t)-s} \alpha \epsilon_{t_s}.
$$

Where $N_2(t) = |\{1 \leq l \leq t : X_l = e_i\}|$ and $1 \leq t_1 \leq t_2 \ldots t_{N_2(t)} \leq t$ is the increasing and exhaustive sequence of times such that $X_{t_s} = e_i$. We understand an empty summation to be 0. Therefore we have:

$$
\begin{aligned}
&\mathbb{E}\left[|\langle w_{t+1}, e_i \rangle|^2 \big| X_t = e_j\right] \\
&= \sum_{n=0}^{t} \mathbb{E}\left[\sum_{s,p=1}^{n} (1-\alpha)^{2n-s-p} \alpha^2 \epsilon_{t_s} \epsilon_{t_p} \bigg| X_t = e_j, N_i(t) = n\right] \mathbb{P}(N_i(t) = n | X_t = e_j) \\
&= \sum_{n=0}^{t} \mathbb{E}\left[\sum_{s=1}^{n} (1-\alpha)^{2n-2s} \alpha^2 \sigma^2 \bigg| X_t = e_j, N_i(t) = n\right] \mathbb{P}(N_i(t) = n | X_t = e_j) \\
&= \alpha^2 \sigma^2 \mathbb{E}\left[\frac{1 - (1-\alpha)^{2N_i(t)}}{1 - (1-\alpha)^2} \bigg| X_t = e_j\right] \tag{36} \\
&= \frac{\alpha \sigma^2}{2 - \alpha}\left(1 - \mathbb{E}\left[(1-\alpha)^{2N_i(t)} \big| X_t = e_j\right]\right) \tag{37}
\end{aligned}
$$

In the second step we have used the fact that the sequence $(\epsilon_s)$ is i.i.d mean 0 and independent of the sequence $(X_s)$. It is now sufficient to show that

$$
\mathbb{E}\left[(1-\alpha)^{2N_i(t)} \big| X_t = e_j\right] \to 0
$$

as $t \to \infty$.

Clearly, $\mathbb{E}N_i(t) = t/d$. We will now bound $\mathbb{E}(1-\alpha)^{2N_i(t)}$. By a direct application of Corollary 2.10 in [23], we conclude that for any $x \geq 0$

$$\mathbb{P}(N_i(t) \le \mathbb{E}N_i(t) - x) \le \exp\left(-\frac{2x^2}{9t\tau_{\mathrm{mix}}^0}\right).$$

Taking $x = t/2d$, we conclude:

$$\mathbb{P}(N_i(t) \le \tfrac{t}{2d}) \le \exp\left(-\frac{t}{18d^2\tau_{\mathrm{mix}}^0}\right) \tag{38}$$

Now consider:

$$\mathbb{E}(1-\alpha)^{2N_i(t)} \le \mathbb{E}(1-\alpha)^{\frac{t}{d}}\mathbb{1}(N_i(t) \ge t/2d) + \mathbb{E}\mathbb{1}(N_i(t) \le \tfrac{t}{2d})$$

$$\le (1-\alpha)^{\frac{t}{d}} + \mathbb{P}(N_2(t) \le \tfrac{t}{2d})$$

$$\le (1-\alpha)^{\frac{t}{d}} + \exp\left(-\frac{t}{18d^2\tau_{\mathrm{mix}}^0}\right)$$

In the last step we have used Equation (38). Now,

$$\mathbb{E}\left[(1-\alpha)^{2N_i(t)}\big|X_t = e_j\right] = \frac{1}{\mathbb{P}(X_t = e_j)}\mathbb{E}\left[(1-\alpha)^{2N_i(t)}\mathbb{1}(X_t = k)\right]$$

$$\le \frac{1}{\mathbb{P}(X_t = e_j)}\mathbb{E}\left[(1-\alpha)^{2N_i(t)}\right]$$

$$= d\mathbb{E}\left[(1-\alpha)^{2N_i(t)}\right]$$

$$\le d(1-\alpha)^{\frac{t}{d}} + d\exp\left(-\frac{t}{18d^2\tau_{\mathrm{mix}}^0}\right)$$

From this the result of the lemma follows.

$\square$

### D.3 SGD with Data Drop is Unbiased and Minimax Optimal in the Agnostic Setting

*Proof of Theorem 8.* Let $(\tilde{X}_K, \tilde{X}_{2K}, \ldots, \tilde{X}_T) \sim \pi^{\otimes(T/K)}$ and let $\tilde{w}_t$ be $t$-th iterate of standard SGD when applied to $(\tilde{X}_K, \tilde{X}_{2K}, \ldots, \tilde{X}_T)$.

Define $\Delta_t := w_t - \tilde{w}_t$. We will bound $\mathbb{E}\|\Delta_t\|^2$ for every $t$. Clearly, if $\{(\tilde{X}_K, \tilde{X}_{2K}, \ldots, \tilde{X}_T) = (X_K, X_{2K}, \ldots, X_T)\}$, then $\Delta_t = 0$. We call this event $\mathcal{C}$. In the event $\mathcal{C}^c$, we use the coarse bound given in Lemma 9 to bound $\Delta_t$.

We have the following comparison theorem between i.i.d SGD and Markovian SGD-DD. Recall that,q

$$\hat{w} = \frac{2K}{T}\sum_{s=T/2K+2}^{T/K+1} w_s, \quad \hat{\tilde{w}} = \frac{2K}{T}\sum_{s=T/2K+2}^{T/K+1} \tilde{w}_s .$$

Using Lemma 4, we have:

$$\mathcal{L}(\hat{w}) - \mathcal{L}(w^*) = (\hat{w} - w^*)^\mathsf{T} A(\hat{w} - w^*) = (\hat{w} - \hat{\tilde{w}} + \hat{\tilde{w}} - w^*)^\mathsf{T} A(\hat{w} - \hat{\tilde{w}} + \hat{\tilde{w}} - w^*)$$

$$= 4(\tfrac{\hat{w}-\hat{\tilde{w}}}{2} + \tfrac{\hat{\tilde{w}}-w^*}{2})^\mathsf{T} A(\tfrac{\hat{w}-\hat{\tilde{w}}}{2} + \tfrac{\hat{\tilde{w}}-w^*}{2}) \le 2(\hat{\tilde{w}} - w^*)^\mathsf{T} A(\hat{\tilde{w}} - w^*) + 2(\hat{w} - \hat{\tilde{w}})^\mathsf{T} A(\hat{w} - \hat{\tilde{w}})$$

$$\le 2(\hat{\tilde{w}} - w^*)^\mathsf{T} A(\hat{\tilde{w}} - w^*) + 2\|\hat{w} - \hat{\tilde{w}}\|^2 \tag{39}$$

In the fourth step we have used the fact that $A$ is a PSD matrix and hence $z \to z^\mathsf{T} Az$ is a convex function. In the fifth step we have used the fact that $\|A\|_{\mathrm{op}} \le 1$.

Now, to conclude the statement of the theorem from the equation above, we need to bound $\mathbb{E}\left[\|\hat{w} - \hat{\tilde{w}}\|^2\right]$. By an application of Jensen's inequality, it is clear that:

$$\mathbb{E}\left[\|\hat{w} - \hat{\tilde{w}}\|^2\right] \le \sup_{\frac{T}{2K}+2 \le t \le \frac{T}{K}+1} \mathbb{E}\left[\|w_s - \tilde{w}_s\|^2\right] .$$

Now, under the event $\mathcal{C}$, $w_s - \tilde{w}_s = 0$ and under the event $\mathcal{C}^c$, we use the bounds on $\mathbb{E}\left[\|\Delta_t\|^2|\mathcal{C}^c\right]$ given in Lemma 9 to conclude:

$$\mathbb{E}\left[\|\hat{w} - \hat{\tilde{w}}\|^2\right] \leq \sup_{\frac{T}{2K}+2\leq s\leq \frac{T}{K}+1} \mathbb{E}\left[\|w_s - \tilde{w}_s\|^2\right] = \sup_{\frac{T}{2K}+2\leq s\leq \frac{T}{K}+1} \mathbb{P}(\mathcal{C}^c)\mathbb{E}\left[\|w_s - \tilde{w}_s\|^2\big|\mathcal{C}^c\right]$$

$$\leq \left[4(T/K+1)T/K\|w_1\|^2 + 4(T/K)^2(T/K+1)\alpha^2 v\right]e^{-K/\tau_{\text{mix}}}. \quad (40)$$

For $T, K \geq 3$, we have $T/K + 1 \leq T$ and by definition of $K$, $e^{-K/\tau_{\text{mix}}} \leq \frac{1}{T^L}$. Combining this with Equations (39) and (40), we have:

$$\mathbb{E}[\mathcal{L}(\hat{w})] - \mathcal{L}(w^*) \leq 2\left[\mathbb{E}[\mathcal{L}(\hat{\tilde{w}})] - \mathcal{L}(w^*)\right] + \frac{8\|w_0\|^2}{T^{L-2}} + \frac{8\alpha^2 v}{T^{L-3}}.$$

$\square$

**Lemma 9.** *Fix a sequence $\{x_K, x_{2K}, \ldots, x_T\}$ in $\Omega$. Call this vector $\mathbf{x}$. Similarly, we let $\mathbf{X}$ and $\tilde{\mathbf{X}}$ respectively denote $(X_{tK})_{t=1}^{T/K}$ and $(\tilde{X}_{tK})_{r=1}^{T/K}$ where $X$ and $\tilde{X}$ are as defined in the proof of Theorem 8. Now, the following holds for any $\alpha \leq 1$:*

1. $\mathbb{E}\left[\|w_t\|^2|\mathbf{X} = \mathbf{x}, \tilde{\mathbf{X}} = \tilde{\mathbf{x}}\right] \leq t\|w_1\|^2 + t(t-1)\alpha^2 v$.

2. $\mathbb{E}\left[\|\tilde{w}_t\|^2|\mathbf{X} = \mathbf{x}, \tilde{\mathbf{X}} = \tilde{\mathbf{x}}\right] \leq t\|w_1\|^2 + t(t-1)\alpha^2 v$.

   *We recall that $v$ is the uniform bound on $\mathbb{E}\|xY_t(x)\|^2$ as given in Section 2. Therefore,*

   $$\mathbb{E}\left[\|\Delta_t\|^2|\mathcal{C}^c\right] \leq 4t\|w_1\|^2 + 4t(t-1)\alpha^2 v.$$

*Proof.* We will prove the inequality given in item 1. The inequality given in item 2 follows similarly. Define the matrices $B_s = I - \alpha X_{sK}X_{sK}^{\intercal}$ and $E_s = X_{sK}y_{sK}$. Clearly, $w_{s+1} = B_s w_s + \alpha E_s$.

Clearly, $\|B_s\|_{\text{op}} \leq 1$ almost surely. Therefore, almost surely: $\|w_{s+1}\| \leq \|w_s\| + \alpha\|E_s\|$. Summing the telescoping series from 1 to $t-1$, we have almost surely: $\|w_t\| \leq \|w_1\| + \sum_{s=1}^{t-1}\alpha\|E_s\|$. By Jensen's inequality,

$$\|w_t\|^2 \leq t\|w_1\|^2 + t\sum_{s=1}^{t-1}\alpha^2\|E_s\|^2.$$

Lemma items 1 and 2 now follow by taking the necessary conditional expectation on both sides, using the uniform bound $\mathbb{E}[\|xy_t(x)\|^2] \leq v$ for all $x \in \Omega$ and using $t \in \mathbb{N}$ as given in Section 2. The conditional expectation bound follows from the fact that $\|\Delta_t\|^2 = \|w_t - \tilde{w}_t\|^2 \leq 2\|w_t\|^2 + 2\|\tilde{w}_t\|^2$ and using the bounds in items 1 and 2. $\square$

### D.4 Parallel SGD accelerates Noise Decay in the Independent Noise Setting

For ease of notation, for the rest of the section, we define:

$$X_{l,i} := X_{(l-1)K+i}, \quad \epsilon_{l,i} := \epsilon_{(l-1)K+i}, \quad \Gamma_{t,s}^{(i)} := \prod_{l=s}^{l=t-1}\left(I - \alpha\left(X_{l,i}X_{l,i}^{\intercal}\right)\right),$$

where $t \geq s+1$. For $t = s$, we use the convention that this product denotes $I$. We unroll the recursion in Algorithm 2 to show:

$$w_t^{(i)} = w^* + \Gamma_{t,1}^{(i)}(w_1^{(i)} - w^*) + \alpha\sum_{l=1}^{t-1}\epsilon_{l,i}\Gamma_{t,l+1}^{(i)}X_{l,i} = w_{t,i}^{\text{bias}} + w_{t,i}^{\text{var}}, \quad (41)$$

where $w_{t,i}^{\text{bias}} := \Gamma_{t,1}^{(i)}(w_1^{(i)} - w^*)$ and $w_{t,i}^{\text{var}} = w^* + \alpha\sum_{l=1}^{t-1}\epsilon_{l,i}\Gamma_{t,l+1}^{(i)}X_{l,i}$.

We first state elementary results to understand the bias and the variance term of $w_{t,i}^{\text{var}}$.

**Lemma 10.**      *1. $w_{t,i}^{\text{var}}$ is the output of SGD when $w_1^{(i)} = w^*$ and $\mathbb{E}w_{t,i}^{\text{var}} = w^*$*

2. *Every entry of $w_{t,i}^{\text{var}}$ is uncorrelated with every entry of $w_{s,j}^{\text{bias}}$ for every $t, i, s, j$.*

3. *Every entry of $w_{t,i}^{\text{var}}$ is uncorrelated with every entry of $w_{s,j}^{\text{var}}$ for every $t, s$ when $i \neq j$*

*Proof.*     1. This follows from Equation 41 and the fact that $\epsilon_{l,i}$ are mean 0 random variables independent of $a_{l,i}$.

2. This follows from the fact that $\epsilon_t$ are i.i.d. mean 0 and independent of the Markov chain.

3. The proof is similar to the proof of item 2.

$\square$

Define

$$\hat{w}^{\text{bias}} := \frac{2}{T} \sum_{i=1}^{K} \sum_{t=T/2K+1}^{T/K} w_{t,i}^{\text{bias}}, \qquad \hat{w}^{\text{var}} := \frac{2}{T} \sum_{i=1}^{K} \sum_{t=T/2K+1}^{T/K} w_{t,i}^{\text{var}}.$$

Now, $\hat{w} = \hat{w}^{\text{bias}} + \hat{w}^{\text{var}}$, where $\hat{w}$ is the output of the parallel SGD algorithm (Algorithm 2). The following lemma follows from a simple application of item 2 of Lemma 4 and Lemma 10.

**Lemma 11.**

$$\mathbb{E}[\mathcal{L}(\hat{w})] = \mathbb{E} \left( \hat{w}^{\text{var}} - w^* \right)^\mathsf{T} A \left( \hat{w}^{\text{var}} - w^* \right) + \mathbb{E} \left[ \left( \hat{w}^{\text{bias}} \right)^\mathsf{T} A \left( \hat{w}^{\text{bias}} \right) \right] + \mathcal{L}(w^*)$$

We will bound the two terms in the above lemma separately. Bound for each of the terms is provided in Appendix D.4.1 and Appendix D.4.2, respectively.

### D.4.1    The Bias Term

In this section we will show that bias decays exponentially in $T$ when $K$ is large enough. Define sigma algebra $\mathcal{F}_{t,i} := \sigma(\epsilon_{s,i}, X_{s,i} : 1 \leq s \leq t)$

**Lemma 12.** *Let $K > \tau_{\text{mix}} \lceil r \log_2 T \rceil$ and $\Gamma_{t,s}^{(i)}$ and other notation be as defined in Section A.2. Then,*

1. *For every $t > s$, $\mathbb{E} \left[ \Gamma_{t,s}^{(i)} \big| \mathcal{F}_{t-2,i} \right] = (I - \alpha A + E_t) \Gamma_{t-1,s}^{(i)}$ where $E_t$ is a random matrix such that $\|E_t\| \leq \frac{\alpha}{T^r}$ almost surely.*

2. *For every random vector $X \in \mathcal{F}_{t-2,i}$ such that $\mathbb{E}\|X\|^2 < \infty$. Let $\alpha < 1$ and $T^r > 2\kappa$. we have:*

$$\mathbb{E}\|\Gamma_{t,t-1}^{(i)} X\|^2 \leq \left(1 - \frac{\alpha}{2\kappa}\right) \mathbb{E}\|X\|^2.$$

*Proof.* We first observe that $\Gamma_{t,s}^{(i)} = \left[ I - \alpha X_{i,t-1} X_{i,t-1}^\mathsf{T} \right] \Gamma_{t-1,s}^{(i)}$ and $\Gamma_{t-1,s}^{(i)} \in \mathcal{F}_{t-2,i}$. Therefore,

$$\mathbb{E} \left[ \Gamma_{t,s}^{(i)} \big| \mathcal{F}_{t-2,i} \right] = \mathbb{E} \left[ I - \alpha X_{i,t-1} X_{i,t-1}^\mathsf{T} \big| \mathcal{F}_{t-2,i} \right] \Gamma_{t-1,s}^{(i)}. \tag{42}$$

Let $P^K$ denote the law of $X_{(t-1)K+i}$ conditioned on $\mathcal{F}_{t-2,i}$. From equation (1), $\text{TV}(P^K, \pi) \leq \frac{1}{T^r}$ almost surely. Now,

$$\mathbb{E} \left[ I - \alpha X_{i,t-1} X_{i,t-1}^\mathsf{T} \big| \mathcal{F}_{t-2,i} \right] = I - \alpha \sum_{x \in \Omega} [xx^\mathsf{T}] P^K(x)$$

$$= I - \alpha \sum_{x \in \Omega} [xx^\mathsf{T}] \pi(x) + \alpha \sum_{x \in \Omega} xx^\mathsf{T}(P^K(x) - \pi(x)) = I - \alpha A + \alpha \sum_{x \in \Omega} xx^\mathsf{T}(P^K(x) - \pi(x))$$

$$\tag{43}$$

We take $E_t := \alpha \sum_{x \in \Omega} xx^\intercal (P^K(x) - \pi(x))$. Define the event, $\mathcal{A} := \{x \in \Omega : P^K(x) \geq \pi(x)\}$. For any arbitrary $\theta \in \mathbb{R}^d$, we have:

$$
\begin{aligned}
|\theta^\intercal E_t \theta| &= \alpha \Big| \sum_x \langle x, \theta \rangle^2 (P^K(x) - \pi(x)) \Big| \\
&\leq \alpha \max \left( \sum_{x \in \mathcal{A}} \langle x, \theta \rangle^2 (P^K(x) - \pi(x)), \sum_{x \in \mathcal{A}^c} \langle x, \theta \rangle^2 (\pi(x) - P^K(x)) \right) \\
&\leq \alpha \|\theta\|^2 \mathrm{TV}(P^K, \pi) \leq \frac{\alpha}{T^r} \|\theta\|^2 \text{ a.s.}
\end{aligned}
\tag{44}
$$

In the third step above, we have used the fact that $\|x\| \leq 1$. First part of the Lemma now follows using Equations (43), (44) with Equation (42).

Next, we consider:

$$
\begin{aligned}
\mathbb{E}\left[ \|\Gamma_{t,t-1}^{(i)} X\|^2 \right] &= \mathbb{E}\left[ X^\intercal \mathbb{E}\left[ I - 2\alpha X_{t-1,i} X_{t-1,i}^\intercal + \alpha^2 \|X_{t-1,i}\|^2 X_{t-1,i} X_{t-1,i}^\intercal \big| \mathcal{F}_{t-2,i} \right] X \right] \\
&\leq \mathbb{E}\left[ X^\intercal \mathbb{E}\left[ I - (2\alpha - \alpha^2) X_{t-1,i} X_{t-1,i}^\intercal \big| \mathcal{F}_{t-2,i} \right] X \right]
\end{aligned}
\tag{45}
$$

In the second step we have used the fact that $\|X_{t-1,i}\| \leq 1$ almost surely. Substituting $s = t - 1$ and replacing $\alpha$ by $2\alpha - \alpha^2$ in item 1 above, we conclude:

$$
\mathbb{E}\left[ I - (2\alpha - \alpha^2) X_{t,i} X_{t,i}^\intercal \big| \mathcal{F}_{t-2,i} \right] = I - (2\alpha - \alpha^2) A + E_t,
$$

where $\|E_t\| \leq \frac{2\alpha - \alpha^2}{T^r}$ a.s. Combining the above equation with (45) and using $A \succeq \frac{1}{\kappa} I$, we obtain:

$$
\begin{aligned}
\mathbb{E}\left[ \|\Gamma_{t,t-1}^{(i)} X\|^2 \right] &\leq \mathbb{E}\left[ X^\intercal \mathbb{E}\left[ I - (2\alpha - \alpha^2) A + E_t | \mathcal{F}_{t-2,i} \right] X \right] \\
&\leq \left( 1 - (2\alpha - \alpha^2)(\tfrac{1}{\kappa} - \tfrac{1}{T^r}) \right) \mathbb{E}\left[ \|X\|^2 \right] \leq \left( 1 - \tfrac{\alpha}{2\kappa} \right) \mathbb{E}\left[ \|X\|^2 \right]
\end{aligned}
\tag{46}
$$

Second part of the Lemma now follows by using the above equation. $\qquad \square$

**Lemma 13.** *Let the data $X_1, \ldots, X_T$ be generated from an exponentially mixing Markov Chain MC. Let $K > \tau_{\mathrm{mix}} \lceil r \log_2 T \rceil$, $\alpha < 1$ and $T^r > 2\kappa$ for some $r > 0$. Then, we have:*

$$
\mathbb{E}[\|w_{t,i}^{\mathrm{bias}}\|^2] \leq \left( 1 - \tfrac{\alpha}{2\kappa} \right)^{t-1} \|w_1^{(i)}\|^2
\tag{47}
$$

*Consequently,*

$$
\mathbb{E}\left( \hat{w}^{\mathrm{bias}} \right)^\intercal A \left( \hat{w}^{\mathrm{bias}} \right) \leq \left( 1 - \tfrac{\alpha}{2\kappa} \right)^{\frac{T}{2K}} \frac{1}{K} \left[ \sum_{i=1}^{K} \|w_1^{(i)} - w^*\|^2 \right]
\tag{48}
$$

*Proof of Lemma 13.* Clearly, $w_{t,i}^{\mathrm{bias}} = \Gamma_{t,t-1}^{(i)} \left( w_{t-1,i}^{\mathrm{bias}} \right)$. It is clear that $w_{t-1,i}^{\mathrm{bias}} \in \mathcal{F}_{t-2,i}$. Applying item 2 in Lemma 12, we get: $\mathbb{E}\|w_{t,i}^{\mathrm{bias}}\|^2 \leq \left( 1 - \tfrac{\alpha}{2\kappa} \right) \mathbb{E}\|w_{t-1,i}^{\mathrm{bias}}\|^2$. By induction, we conclude Equation (47).

Now, $\left( \hat{w}^{\mathrm{bias}} \right)^\intercal A \left( \hat{w}^{\mathrm{bias}} \right) \leq \|\hat{w}^{\mathrm{bias}}\|^2$ since $\|A\|_{\mathrm{op}} \leq 1$. By Jensen's inequality,

$$
\|\hat{w}^{\mathrm{bias}}\|^2 \leq \frac{2}{T} \sum_{i=1}^{K} \sum_{t=T/2K+1}^{T/K} \|w_{t,i}^{\mathrm{bias}}\|^2.
$$

Lemma now follows by taking expectation on both sides and using Equation (47). $\qquad \square$

We have therefore bound the bias term of Theorem 9 in Lemma 13.

### D.4.2 The Variance Term

We will now bound the variance term. It is clear that,

$$\left(\hat{w}^{\mathrm{var}} - w^*\right)^{\mathsf{T}} A\left(\hat{w}^{\mathrm{var}} - w^*\right) = \frac{4}{T^2} \sum_{i,j=1}^{K} \sum_{t,s=T/2K+1}^{T/K} \left(w_{t,i}^{\mathrm{var}} - w^*\right)^{\mathsf{T}} A\left(w_{s,j}^{\mathrm{var}} - w^*\right)$$

Using the item 3 in Lemma 10, we get:

$$\mathbb{E}\left[\left(\hat{w}^{\mathrm{var}} - w^*\right)^{\mathsf{T}} A\left(\hat{w}^{\mathrm{var}} - w^*\right)\right] = \frac{4}{T^2} \sum_{i=1}^{K} \sum_{t,s=T/2K+1}^{T/K} \mathbb{E}\left[\left(w_{t,i}^{\mathrm{var}} - w^*\right)^{\mathsf{T}} A\left(w_{s,i}^{\mathrm{var}} - w^*\right)\right] \quad (49)$$

Consider the following term in the RHS of Equation (49)

$$\mathbb{E}\left[\left(w_{t,i}^{\mathrm{var}} - w^*\right)^{\mathsf{T}} A\left(w_{s,i}^{\mathrm{var}} - w^*\right)\right] = \alpha^2 \sum_{l=1}^{t-1} \sum_{m=1}^{s-1} \mathbb{E}\left[\epsilon_{l,i}\epsilon_{m,i} X_{l,i}^{\mathsf{T}} \left(\Gamma_{t,l+1}^{(i)}\right)^{\mathsf{T}} A\Gamma_{s,m+1}^{(i)} X_{m,i}\right]$$

$$= \alpha^2\sigma^2 \sum_{l=1}^{\min(t-1,s-1)} \mathbb{E}\left[X_{l,i}^{\mathsf{T}} \left(\Gamma_{t,l+1}^{(i)}\right)^{\mathsf{T}} A\Gamma_{s,l+1}^{(i)} X_{l,i}\right], \quad (50)$$

where the last step holds as $\epsilon_t$ are i.i.d., mean zero random variables with variance $\sigma^2$ and are independent of $(X_s)_{s\in\mathbb{N}}$. For the sake of clarity, we will take $i$ to sum from 1 to $K$ and $t,s$ to sum from $T/2K+1$ to $T/K$ in the equations below without stating this explicitly. Using equations (49) and (50), we conclude:

$$\mathbb{E}\left[\left(\hat{w}^{\mathrm{var}} - w^*\right)^{\mathsf{T}} A\left(\hat{w}^{\mathrm{var}} - w^*\right)^{\mathsf{T}}\right] = \frac{4\alpha^2\sigma^2}{T^2} \sum_{i,t,s} \sum_{l=1}^{\min(t-1,s-1)} \mathbb{E}\left[X_{l,i}^{\mathsf{T}} \left(\Gamma_{t,l+1}^{(i)}\right)^{\mathsf{T}} A\Gamma_{s,l+1}^{(i)} X_{l,i}\right].$$
$$(51)$$

Now, we bound RHS above using the following lemma:

**Lemma 14.** *Let $\alpha < 1$, $K \geq \tau_{\mathrm{mix}}\lceil r \log_2 T\rceil$ and $l \leq s - 1$. Then:*

$$\sum_{t=s}^{T/K} \mathbb{E} X_{l,i}^{\mathsf{T}} \left(\Gamma_{t,l+1}^{(i)}\right)^{\mathsf{T}} A\Gamma_{s,l+1}^{(i)} X_{l,i} \leq \frac{\alpha}{4K^2 T^{r-2}} + \frac{1}{\alpha}\mathbb{E}\|\Gamma_{s,l+1}^{(i)} X_{l,i}\|^2 \quad (52)$$

*Proof.* For $t = s$, $X_{l,i}^{\mathsf{T}} \left(\Gamma_{t,l+1}^{(i)}\right)^{\mathsf{T}} A\Gamma_{s,l+1}^{(i)} X_{l,i} = X_{l,i}^{\mathsf{T}} \left(\Gamma_{s,l+1}^{(i)}\right)^{\mathsf{T}} A\Gamma_{s,l+1}^{(i)} X_{l,i}$. Similarly, for $t > s$, we have: $\Gamma_{t,l+1}^{(i)} = \Gamma_{t,t-1}^{(i)}\Gamma_{t-1,l+1}^{(i)}$. Clearly, $X_{l,i}, \Gamma_{s,l+1}^{(i)}$ and $\Gamma_{t-1,l+1}^{i}$ are $\mathcal{F}_{t-2,i}$ measurable. Therefore,

$$\mathbb{E}\left[X_{l,i}^{\mathsf{T}} \left(\Gamma_{t,l+1}^{(i)}\right)^{\mathsf{T}} A\Gamma_{s,l+1}^{(i)} X_{l,i}\right] = \mathbb{E}\left[X_{l,i}^{\mathsf{T}} \left(\Gamma_{t-1,l+1}^{(i)}\right)^{\mathsf{T}} \mathbb{E}\left[\Gamma_{t,t-1}^{(i)}|\mathcal{F}_{t-2,i}\right] A\Gamma_{s,l+1}^{(i)} X_{l,i}\right]$$

$$= \mathbb{E}\left[X_{l,i}^{\mathsf{T}} \left(\Gamma_{t-1,l+1}^{(i)}\right)^{\mathsf{T}} [I - \alpha A + E_t] A\Gamma_{s,l+1}^{(i)} X_{l,i}\right] \leq \mathbb{E}\left[X_{l,i}^{\mathsf{T}} \left(\Gamma_{t-1,l+1}^{(i)}\right)^{\mathsf{T}} [I - \alpha A] A\Gamma_{s,l+1}^{(i)} X_{l,i}\right] + \frac{\alpha}{T^r},$$
$$(53)$$

where the second step follows using Lemma 12. The third step follows as $\|E_t\| \leq \frac{\alpha}{T^r}$ almost surely and the fact that $\|A\|_{\mathrm{op}}, \|X_{l,i}\|, \|\Gamma_{a,b}^{(i)}\| \leq 1$. Continuing in a similar way as Equation (53), we have:

$$\mathbb{E}\left[X_{l,i}^{\mathsf{T}} \left(\Gamma_{t,l+1}^{(i)}\right)^{\mathsf{T}} A\Gamma_{s,l+1}^{(i)} X_{l,i}\right] \leq \mathbb{E}\left[X_{l,i}^{\mathsf{T}} \left(\Gamma_{s,l+1}^{(i)}\right)^{\mathsf{T}} [I - \alpha A]^{t-s} A\Gamma_{s,l+1}^{(i)} X_{l,i}\right] + \frac{\alpha(t-s)}{T^r}.$$
$$(54)$$

Now, $(I - \alpha A)^{t-s}$ is a PSD matrix and it commutes with $A$. Therefore, from Equation (54), it follows that:

$$\sum_{t=s}^{T/K-1} \mathbb{E}\left[X_{l,i}^{\mathsf{T}} \left(\Gamma_{t,l+1}^{(i)}\right)^{\mathsf{T}} A \Gamma_{s,l+1}^{(i)} X_{l,i}\right]$$

$$\leq \sum_{t=s}^{T/K-1} \left[\mathbb{E}X_{l,i}^{\mathsf{T}} \left(\Gamma_{s,l+1}^{(i)}\right)^{\mathsf{T}} [I-\alpha A]^{t-s} A\Gamma_{s,l+1}^{(i)} X_{l,i} + \frac{\alpha(t-s)}{T^r}\right]$$

$$\leq \mathbb{E}\left[X_{l,i}^{\mathsf{T}} \left(\Gamma_{s,l+1}^{(i)}\right)^{\mathsf{T}} \left[\sum_{t=s}^{\infty}(I-\alpha A)^{t-s}\right] A\Gamma_{s,l+1}^{(i)} X_{l,i}\right] + \frac{\alpha T^2}{4K^2 T^r}$$

$$= \frac{1}{\alpha}\mathbb{E}X_{l,i}^{\mathsf{T}} \left(\Gamma_{s,l+1}^{(i)}\right)^{\mathsf{T}} \Gamma_{s,l+1}^{(i)} X_{l,i} + \frac{\alpha}{4K^2 T^{r-2}} \qquad (55)$$

In the third step we have used the fact that $\sum_{i=0}^{\infty}(I-\alpha A)^i = \frac{A^{-1}}{\alpha}$. Equation (55) establishes the result of the Lemma. $\qquad\square$

Consider the following operator on $\mathcal{S}(d)$ - the space of $d \times d$ symmetric matrices:

$$\Lambda(M) = \mathbb{E}_{x\sim\pi}\mathbb{E}(I - \alpha xx^{\mathsf{T}})M(I - \alpha xx^{\mathsf{T}}).$$

Now, a (linear)PSD map is a linear operator over $\mathcal{S}(d)$ which maps PSD matrices to PSD matrices. We list some important properties of $\Lambda$ below.

**Lemma 15.** *1. $\Lambda$ is a PSD map.*

*2. If $A, B \in \mathcal{S}(d)$ such that $B \preceq A$ then $\Lambda(B) \preceq \Lambda(A)$.*

*3. Let $M$ be a PSD operator. Then $\|\Lambda(M)\|_2 \leq \|M\|_2$ and in particular $\Lambda(I) \preceq I$.*

*Proof.* 1. The proof follows from the definition of PSD matrices and PSD maps.

2. $A - B \succeq 0$. By item 1, $\Lambda(A - B) \succeq 0$. Therefore, by linearity of $\Lambda$, $\Lambda(A) \succeq \Lambda(B)$.

3. This follows easily from the definition of $\Lambda$ and submultiplicativity of operator norm and the fact that $\|a_x\| \leq 1$ almost surely.

$\qquad\square$

**Lemma 16.** *Let $\alpha < 1$, $l \leq s - 1$ and $K > \tau_{\mathrm{mix}}\lceil r\log_2 T\rceil$. Then:*

$$\sum_{s=l+1}^{T/K} \mathbb{E}\|\Gamma_{s,l+1}^{(i)} X_{l,i}\|^2 \leq \frac{d\left(4\alpha + 2\alpha^2\right)}{K^2 T^{r-2}} + \sum_{s\geq l+1} \mathrm{Tr}(\Lambda^{s-l-1}(A)).$$

*Here $\Lambda^0$ is understood to be the identity operator.*

*Proof.* Consider

$$\|\Gamma_{s,l+1}^{(i)} X_{l,i}\|^2 = \mathrm{Tr}\left[\Gamma_{s,l+1}^{(i)} X_{l,i} X_{l,i}^{\mathsf{T}} \left(\Gamma_{s,l+1}^{(i)}\right)^{\mathsf{T}}\right] \qquad (56)$$

When $s = l + 1$, it is clear that $\Gamma_{s,l+1}^{(i)} = I$ and $\mathbb{E}\|\Gamma_{s,l+1}^{(i)} X_{l,i}\|^2 = \mathbb{E}\mathrm{Tr}(X_{l,i} X_{l,i}^{\mathsf{T}}) = \mathrm{Tr}(A)$. When $s > l + 1$, $\Gamma_{s,l+1}^{(i)} = \Gamma_{s,s-1}^{(i)}\Gamma_{s-1,l+1}^{(i)}$. For the sake of clarity, we will denote $A_{s,l,i} := \Gamma_{s-1,l+1}^{(i)} X_{l,i} X_{l,i}^{\mathsf{T}} \left(\Gamma_{s-1,l+1}^{(i)}\right)^{\mathsf{T}}$. Since $\Gamma_{s-1,l+1}^{(i)}, X_{l,i} \in \mathcal{F}_{s-2,i}$, we have:

$$\mathbb{E}\left[\Gamma_{s,s-1}^{(i)} A_{s,l,i} \left(\Gamma_{s,s-1}^{(i)}\right)^{\mathsf{T}}\Big|\mathcal{F}_{s-2,i}\right] = \mathbb{E}\left[\left(I - \alpha X_{i,s-1} X_{i,s-1}^{\mathsf{T}}\right) A_{s,l,i} \left(I - \alpha X_{i,s-1} X_{i,s-1}^{\mathsf{T}}\right)\Big|\mathcal{F}_{s-2,i}\right].$$

Let $P^K$ be the distribution of $X_{(s-1)K+i}$ given $\mathcal{F}_{s-2,i}$. From Equation (1), we have: $\mathrm{TV}(P^K, \pi) \leq \frac{1}{T^r}$. Now, using similar arguments as in the proof of Lemma 12 and using the fact that $A_{s,l,i}$ is $\mathcal{F}_{s-2,i}$ measurable, we show that:

$$\mathbb{E}\left[\Gamma_{s,s-1}^{(i)} A_{s,l,i} \left(\Gamma_{s,s-1}^{(i)}\right)^{\mathsf{T}}\Big|\mathcal{F}_{s-2,i}\right] \preceq \Lambda(A_{s,l,i}) + \frac{4\alpha + 2\alpha^2}{T^r}I.$$

Taking expectation on both sides, we get:

$$\mathbb{E}\left[\Gamma^{(i)}_{s,s-1}A_{s,l,i}\left(\Gamma^{(i)}_{s,s-1}\right)^{\intercal}\right] \preceq \mathbb{E}\Lambda(A_{s,l,i}) + \frac{4\alpha + 2\alpha^2}{T^r}\mathrm{I} = \Lambda(\mathbb{E}A_{s,l,i}) + \frac{4\alpha + 2\alpha^2}{T^r}\mathrm{I}, \qquad (57)$$

where in the last step we have used the linearity of the operator $\Lambda$.

We now use induction and results in Lemma 15, to prove:

$$\mathbb{E}\left[\mathrm{Tr}\left(\Gamma^{(i)}_{s,l+1}X_{l,i}X_{l,i}^{\intercal}\left(\Gamma^{(i)}_{s,l+1}\right)^{\intercal}\right)\right] \preceq \Lambda^{s-l-1}(A) + (4\alpha + 2\alpha^2)(s-l-1)\frac{\mathrm{I}}{T^r},$$

where $A = \mathbb{E}\left[X_l^{\intercal}X_l\right]$.

The statement is clearly true when $s = l+1$. If the result is true for $s = s_0 - 1$, by Equation (57) and the definition of $A_{s,l,i}$, we have:

$$\mathbb{E}\left[\mathrm{Tr}\left(\Gamma^{(i)}_{s_0,l+1}X_{l,i}X_{l,i}^{\intercal}\left(\Gamma^{(i)}_{s_0,l+1}\right)^{\intercal}\right)\right] \preceq \Lambda\left[\mathbb{E}\Gamma^{(i)}_{s_0-1,l+1}X_{l,i}X_{l,i}^{\intercal}\left(\Gamma^{(i)}_{s_0-1,l+1}\right)^{\intercal}\right] + (4\alpha + 2\alpha^2)\frac{\mathrm{I}}{T^r}$$

$$\preceq \Lambda\left[\Lambda^{s_0-l-2}(A) + (s_0 - l - 2)(4\alpha + 2\alpha^2)\frac{\mathrm{I}}{T^r}\right] + (4\alpha + 2\alpha^2)\frac{\mathrm{I}}{T^r}$$

$$\preceq \Lambda^{s_0-l-1}(A) + (s_0 - l - 1)(4\alpha + 2\alpha^2)\frac{\mathrm{I}}{T^r},$$

where in the first step we have used Equation (57). In the second step we have use item 2 of Lemma 15 and the induction hypothesis for $s = s_0 - 1$. In the third step we have used linearity of $\Lambda$ and item 3 of Lemma 15 . We use the equation above along with Equation (56) to conclude the result. $\qquad\square$

We will use some results proved in [18] - there we take the batch size $b = 1$ and consider the homoscedastic (independent) noise case. Consider Lemmas 13,14 and 15 in [18]. We have the following correspondences between terms in our work and [18]

1. The step size $\alpha$ in this work corresponds to $\gamma$.

2. The operator $\frac{\mathcal{I}-\Lambda}{\alpha} : \mathcal{S}(d) \to \mathcal{S}(d)$ here corresponds to the operator $\mathcal{T}_b$.

3. The matrix $\sigma^2 A$ here corresponds $\Sigma$.

4. The matrix $A$ here corresponds to $H$.

Under the step size condition becomes $\alpha < \frac{1}{2}$,

$$\sum_{s\geq l+1} \mathrm{Tr}(\Lambda^{s-l-1}(A)) = \mathrm{Tr}\left((\mathcal{I}-\Lambda)^{-1}A\right) = \frac{1}{\alpha}\mathrm{Tr}(\mathcal{T}_b^{-1}A) \leq \frac{2}{\alpha}\mathrm{Tr}(\mathrm{I}) = \frac{2d}{\alpha}. \qquad (58)$$

The first step follows from the proof of Lemma 13 in [18], the second step follows from Lemma 15 item 4 in [18].

We will combine the inequalities proved above to obtain bounds for the RHS of Equation (51). When $\alpha < \frac{1}{2}$ and $K > \tau_{\text{mix}}\lceil r \log_2 T \rceil$, we have:

$$\mathbb{E}(\hat{w}^{\text{var}} - w^*)^{\mathsf{T}} A (\hat{w}^{\text{var}} - w^*)^{\mathsf{T}} = \frac{4\alpha^2\sigma^2}{T^2} \sum_{i,t,s} \sum_{l=1}^{\min(t-1,s-1)} \mathbb{E} X_{l,i}^{\mathsf{T}} \left(\Gamma_{t,l+1}^{(i)}\right)^{\mathsf{T}} A \Gamma_{s,l+1}^{(i)} X_{l,i}$$

$$\leq \frac{8\alpha^2\sigma^2}{T^2} \sum_{i,s} \sum_{t=s}^{T/K-1} \sum_{l=1}^{s-1} \mathbb{E} X_{l,i}^{\mathsf{T}} \left(\Gamma_{t,l+1}^{(i)}\right)^{\mathsf{T}} A \Gamma_{s,l+1}^{(i)} X_{l,i}$$

$$\leq \frac{8\alpha^2\sigma^2}{T^2} \sum_{i,s} \sum_{l=1}^{s-1} \left[ \frac{\alpha}{4K^2 T^{r-2}} + \frac{1}{\alpha} \mathbb{E}\|\Gamma_{s,l+1}^{(i)} X_{l,i}\|^2 \right]$$

$$\leq \frac{2\alpha^3\sigma^2}{K^3 T^{r-2}} + \frac{8\alpha\sigma^2}{T^2} \sum_{i,s} \sum_{l=1}^{s-1} \left[ \mathbb{E}\|\Gamma_{s,l+1}^{(i)} X_{l,i}\|^2 \right]$$

$$= \frac{2\alpha^3\sigma^2}{K^3 T^{r-2}} + \frac{8\alpha\sigma^2}{T^2} \sum_{i} \sum_{l=1}^{T/K-1} \sum_{s=l+1}^{T/K} \left[ \mathbb{E}\|\Gamma_{s,l+1}^{(i)} a_{l,i}\|^2 \right]$$

$$\leq \frac{2\alpha^3\sigma^2}{K^3 T^{r-2}} + \frac{d(32\alpha^2 + 16\alpha^3)\sigma^2}{K^2 T^{r-1}} + \frac{8\alpha\sigma^2}{T^2} \sum_{i} \sum_{l=1}^{T/K-1} \sum_{s=l+1}^{\infty} \text{Tr}(\Lambda^{s-l-1}(A))$$

$$\leq \frac{2\alpha^3\sigma^2}{K^3 T^{r-2}} + \frac{d(32\alpha^2 + 16\alpha^3)\sigma^2}{K^2 T^{r-1}} + \frac{16d\sigma^2}{T}. \tag{59}$$

In the third step we have used Lemma 14. In the sixth step we have used Lemma 16. In the seventh step we have used Equation (58).

The above equation bounds the variance term. Theorem 9 now follows by combining the bias and variance bounds given above.

# E   Experience Replay Accelerates Bias Decay for Gaussian Autoregressive (AR) Dynamics

## E.1   Problem Setting

Suppose our sample vectors $X \in \mathbb{R}^d$ are generated from a Markov chain with the following dynamics:

$$X_1 = G_1$$

$$X_2 = \sqrt{1 - \epsilon^2} X_1 + \epsilon G_2$$

$$\dots$$

$$X_{t+1} = \sqrt{1 - \epsilon^2} X_t + \epsilon G_{t+1}$$

where $\epsilon$ is fixed and known, and each $G_j$ is independently sampled from $\frac{1}{\sqrt{d}}\mathcal{N}(0, I)$.

Each observation $Y_i = X_i^T w^* + \xi_i$, where $\xi_i$ are independently drawn random variables with mean $0$ and variance $\sigma^2$.

We use SGD to find $w$ that minimizes the loss

$$L(w) = \mathbb{E}[(X^T w - Y)^2] - \mathbb{E}[(X^T w^* - Y)^2]$$

for some $X \sim \mathcal{N}(0, \frac{1}{\sqrt{d}}I_d)$.

We first establish standard properties of the vectors $X_i$.

**Lemma 17.** *With probability* $1 - \beta$, $1 - \frac{c}{\sqrt{d}} \log(\frac{1}{\beta}) \leq \|X_j\|_2^2 \leq 1 + \frac{c}{\sqrt{d}} \log(\frac{1}{\beta})$, *for some constant* $c$.

*Proof.* Note that for each $X_j$, each component $k \in [1, d]$, is independently normally distributed with mean 0 and variance $\frac{1}{d}$. Then writing $\|X_j\|^2 = \sum\limits_{k=1}^{d} X_{jk}^2$ and using Bernstein's inequality for sub-exponential random variables , we will get the desired result. $\qquad \square$

**Lemma 18.** *With probability* $1 - \beta$, $-\frac{c}{\sqrt{d}} \log(\frac{1}{\beta}) \leq X_i^T G_j \leq \frac{c}{\sqrt{d}} \log(\frac{1}{\beta})$, *for some constant* $c$, *where* $X$, *and* $G_j$ *are defined as before, ie* $G_j \sim \frac{1}{\sqrt{d}}\mathcal{N}(0, I)$ *and* $j > i$ *(so that* $G_j$ *is independent of* $X_i$*).*

*Proof.* Note that $X_i \sim \mathcal{N}(0, \sigma^2 I)$, where $\sigma^2 = \frac{1}{d}$. Then, for any fixed $X_1$, it follows that $X_i^T G_j = \sum\limits_{\ell=1}^{d} X_{i\ell} G_{j\ell}$, where each random variable in the summation is independent. Since $G_j \sim \mathcal{N}(0, \frac{1}{d}I)$, the result follows by Bernstein's inequality for sub-exponential random variables. $\qquad \square$

**Lemma 19.** *The mixing time of the Gaussian AR chain is* $\Theta\left(\frac{1}{\epsilon^2} \log(d)\right)$.

*Proof.* The stationary distribution of the Gaussian AR chain is $\pi = \mathcal{N}(0, \frac{1}{d}I)$. Given $X_0$, we compare $KL(P^t(X_0), \pi)$. The standard formula for the KL divergence of two multivariate Gaussians $\mathcal{N}(\mu_1, \Sigma_1)$ and $\mathcal{N}(\mu_2, \Sigma_2)$ is:

$$KL(G_1, G_2) = \frac{1}{2}\left[\log\frac{|\Sigma_2|}{|\Sigma_1|} - d + Tr\left(\Sigma_2^{-1}\Sigma_1\right) + (\mu_2 - \mu_1)^T \Sigma_2^{-1}(\mu_2 - \mu_1)\right]$$

Note that $X_t = (1-\epsilon^2)^{\frac{t}{2}} X_0 + \epsilon \sum\limits_{j=0}^{t-1}(1-\epsilon^2)^{\frac{j}{2}} G_j$. Therefore, $P^t(X_0) \sim \mathcal{N}((1-\epsilon^2)^{\frac{t}{2}} X_0, \frac{1-(1-\epsilon^2)^t}{d}I)$
For the KL divergence to be $\leq \delta$ where $\delta$ is a fixed constant, we need:

$$\frac{1}{2}\left[\log C - d + d^2 \frac{1 - (1-\epsilon^2)^t}{d} + d \cdot (1-\epsilon^2)^t \left(1 + \sqrt{\frac{\log(\frac{1}{\beta})}{d}}\right)\right] \leq \delta$$

where $C$ is an appropriate constant. Eventually, we will get that we need $(1-\epsilon^2)^t \leq \frac{c}{\sqrt{d}}$, for some constant $c$. A direct application of Pinsker's inequality shows that $\tau_{\text{mix}} = \Theta\left(\frac{1}{\epsilon^2} \log(d)\right)$. $\qquad \square$

Suppose we have a continuous stream of samples $X_1, X_2, \ldots X_T$ from the Markov Chain. We split the $T$ samples into $\frac{T}{S}$ separate buffers of size $S$ in a sequential manner, ie $X_1, \ldots X_S$ belong to the first buffer. Let $S = B + u$, where $B$ is orders of magnitude larger than $u$. From within each buffer, we drop the first $u$ samples. Then starting from the first buffer, we perform $B$ steps of SGD, where for each iteration, we sample uniformly at random from within the $[u, B + u]$samples in the first buffer. Then perform the next $B$ steps of SGD by uniformly drawing samples from within the $[u, B + u]$ samples in the second buffer. We will choose $u$ so that the buffers are are approximately i.i.d..

We run SGD this way for the first $\frac{T}{2S}$ buffers to ensure that the bias of each iterate is small. Then for the last $\frac{T}{2S}$ buffers, we perform SGD in the same way, but we tail average over the last iterate produced using each buffer to give our final estimate $w$. We formally write Algorithm 4.

**Theorem 10** (SGD with Experience Replay for Gaussian AR Chain). *For any* $\epsilon \leq 0.21$, *if* $B \geq \frac{1}{\epsilon^7}$ *and* $d = \Omega(B^4 \log(\frac{1}{\beta}))$, *with probability at least* $1 - \beta$, *Algorithm 4 returns* $w$ *such that* $\mathbb{E}[\mathcal{L}(w)] \leq O\left(\exp\left(\frac{-T \log(d)}{\kappa \sqrt{\tau_{\text{mix}}}}\right) \|w_0 - w^*\|^2\right) + \tilde{O}\left(\frac{\sigma^2 d \sqrt{\tau_{\text{mix}}}}{T}\right) + \mathcal{L}(w^*)$. *Recall that* $\kappa = d$

*Proof.* $\mathbb{E}[\mathcal{L}(w)] = \mathbb{E}(\mathcal{L}(w^{\text{bias}})) + \mathbb{E}(\mathcal{L}(w^{\text{var}})) + \mathcal{L}(w^*)$. We analyze $\mathbb{E}(\mathcal{L}(w^{\text{bias}}))$ in Theorem 11 and analyze $\mathbb{E}(\mathcal{L}(w^{\text{bias}}))$ in Theorem 12. $\qquad \square$

## E.2 Bias Decay with Experience Replay

Standard analysis of SGD says that for bias decay, $w_{t+1}^{\mathrm{bias}} - w^* = (I - \eta \hat{X}_t \hat{X}_t^T)(w_t^{\mathrm{bias}} - w^*)$, where $(\hat{X}_t, \hat{y}_t)$ is the sample used in the $t$-th iteration of SGD.

**Theorem 11** (Bias Decay for SGD with Experience Replay for Gaussian AR Chain). *For any $\epsilon \le 0.21$, if $B \ge \frac{1}{\epsilon^7}$ and $d = \Omega(B^4 \log(\frac{1}{\beta}))$, with probability at least $1 - \beta$, Algorithm 4 produces $w$ such that $\mathbb{E}[\mathcal{L}(w^{\mathrm{bias}})] \le O\left( \exp\left( \frac{-T \log(d)}{\kappa \sqrt{\tau_{\mathrm{mix}}}} \right) \|w_0 - w^*\|^2 \right)$.*

*Proof of Theorem 11.* We first write $Tr\left( \mathbb{E}\left[ (w_{\frac{BT}{S}} - w^*)(w_{\frac{BT}{S}} - w^*)^T \right] \right)$ as:

$$Tr\left( \mathbb{E}\left[ \left( I - \eta \hat{X}_{\frac{BT}{S}} \hat{X}_{\frac{BT}{S}}^T \right) \dots \left( I - \eta \hat{X}_1 \hat{X}_1^T \right) (w_0 - w^*)(w_0 - w^*)^T \left( I - \eta \hat{X}_1 \hat{X}_1^T \right) \dots \left( I - \eta \hat{X}_{\frac{BT}{S}} \hat{X}_{\frac{BT}{S}}^T \right) \right] \right)$$

Since we assume $(w_0 - w^*)^T$ to be independent standard Gaussian, we are mostly interested in:

$$Tr\left( \mathbb{E}\left[ \left( I - \eta \hat{X}_1 \hat{X}_1^T \right) \dots \left( I - \eta \hat{X}_{\frac{BT}{S}} \hat{X}_{\frac{BT}{S}}^T \right) \left( I - \eta \hat{X}_{\frac{BT}{S}} \hat{X}_{\frac{BT}{S}}^T \right) \dots \left( I - \eta \hat{X}_1 \hat{X}_1^T \right) \right] \right)$$

This can be written as:

$$Tr\left( \mathbb{E}\left[ \mathbb{E}\left[ \left( I - \eta \hat{X}_1 \hat{X}_1^T \right) \dots \left( I - \eta \hat{X}_{\frac{BT}{S}} \hat{X}_{\frac{BT}{S}}^T \right) \left( I - \eta \hat{X}_{\frac{BT}{S}} \hat{X}_{\frac{BT}{S}}^T \right) \dots \left( I - \eta \hat{X}_1 \hat{X}_1^T \right) | X_1, \dots X_{\frac{(B-1)T}{S}} \right] \right] \right)$$

The quantity of interest is therefore,

$$\mathbb{E}\left[ \left( I - \eta \hat{X}_{\frac{(B-1)T}{S}+1} \hat{X}_{\frac{(B-1)T}{S}+1}^T \right) \dots \left( I - \eta \hat{X}_{\frac{BT}{S}} \hat{X}_{\frac{BT}{S}}^T \right) \left( I - \eta \hat{X}_{\frac{BT}{S}} \hat{X}_{\frac{BT}{S}}^T \right) \dots \left( I - \eta \hat{X}_{\frac{(B-1)T}{S}} \hat{X}_{\frac{(B-1)T}{S}}^T \right) | X_1, \dots X_{\frac{(B-1)T}{S}} \right],$$

which Lemma 21 says is $\preceq \left( 1 - \frac{1}{8} \cdot \frac{\epsilon B}{40 d \pi} \right) I$. Therefore, if $N$ is the number of buffers, it follows that the loss decays at a rate of $\exp\left( \frac{-NB\epsilon}{\kappa} \right) \|w_0 - w^*\|^2$, and since $T = NB$, we conclude that the rate is $\exp\left( \frac{-T \log(d)}{\kappa \sqrt{\tau_{\mathrm{mix}}}} \right) \|w_0 - w^*\|^2$. $\qquad\square$

We first solve the issue that the buffers are approximately iid by establishing a relationship between the contraction rate of the sampled vectors with the contraction rate of vectors sampled from buffers that are iid. The proof compares two parallel processes, one where the samples of the buffer follow Gaussian AR dynamics from an initial $X_0$, where this initial $X_0$ is the $Sj$-th sample from the Markov Chain for some buffer index $j$, and another process which follows Gaussian AR dynamics from an initial $\tilde{X}_0$ generated independently from $\frac{1}{\sqrt{d}} \mathcal{N}(0, I)$. We show that the expected contraction of the first processes can be bounded by the expected contraction of the second process plus a constant.

**Lemma 20.** *Suppose that $\hat{X}_1, \dots \hat{X}_j$ are vectors sampled from arbitrary buffers (ie, they are of the form $X_{u+s} = \left(1 - \epsilon^2\right)^{(u+s)/2} X_0 + \epsilon \sum_{t=1}^{u+s} \left(1 - \epsilon^2\right)^{(t-(u+s))/2} G_t$, for $X_0$ which is the first vector in the buffer). Sample a new random $\tilde{X}_0$ independently from $\frac{1}{\sqrt{d}} \mathcal{N}(0, I)$ and denote $\tilde{X}_{u+s} := \left(1 - \epsilon^2\right)^{(u+s)/2} \tilde{X}_0 + \epsilon \sum_{t=1}^{u+s} \left(1 - \epsilon^2\right)^{(t-(u+s))/2} G_t$. Then we have:*

$$\mathbb{E}\left[ \left( I - \eta \hat{X}_1 \hat{X}_1^T \right) \dots \left( I - \eta \hat{X}_j \hat{X}_j^T \right) \right] \preceq \mathbb{E}\left[ \left( I - \eta \hat{\tilde{X}}_1 \hat{\tilde{X}}_1^T \right) \dots \left( I - \eta \hat{\tilde{X}}_j \hat{\tilde{X}}_j^T \right) \right] + c \cdot j^2 (1 - \epsilon^2)^{\frac{u}{2}},$$

*where $c$ is a constant, and $\hat{\tilde{X}}_1$ denotes the same sample index as that of $\hat{X}_1$, except that the initial vector was sampled independently from $\frac{1}{\sqrt{d}} \mathcal{N}(0, I)$.*

*Proof.* We see that in general $X_{u+s} - \tilde{X}_{u+s} = \left( 1 - \epsilon^2 \right)^{(u+s)/2} \left( X_0 - \tilde{X}_0 \right)$. With probability more than $1 - \beta$, we have $\left\| X_0 - \tilde{X}_0 \right\| \le \sqrt{2} + \sqrt{\frac{\log \frac{1}{\delta}}{d}}$ and so $\left\| X_{u+s} - \tilde{X}_{u+s} \right\| \le \left( 1 - \epsilon^2 \right)^{(u+s)/2} \cdot \left( \sqrt{2} + \sqrt{\frac{\log \frac{1}{\delta}}{d}} \right) \le \left( 1 - \epsilon^2 \right)^{u/2} \cdot \left( \sqrt{2} + \sqrt{\frac{\log \frac{1}{\delta}}{d}} \right)$

Therefore,

$$\mathbb{E}\left[\left(I - \eta\hat{X}_1\hat{X}_1^T\right)\ldots\left(I - \eta\hat{X}_j\hat{X}_j^T\right)\right]$$

$$= \mathbb{E}\left[\left(I - \eta\hat{X}_1\hat{X}_1^T + \eta\hat{\tilde{X}}_1\hat{\tilde{X}}_1^T - \eta\hat{\tilde{X}}_1\hat{\tilde{X}}_1^T\right)\ldots\left(I - \eta\hat{X}_j\hat{X}_j^T + \eta\hat{\tilde{X}}_j\hat{\tilde{X}}_j^T - \eta\hat{\tilde{X}}_j\hat{\tilde{X}}_j^T\right)\right]$$

$$\preceq \mathbb{E}\left[\left(I - \eta\hat{\tilde{X}}_1\hat{\tilde{X}}_1^T\right)\ldots\left(I - \eta\hat{\tilde{X}}_j\hat{\tilde{X}}_j^T\right)\right] + \sum_{s=1}^{j}\binom{j}{s}\left(4\left(1-\epsilon^2\right)^{u/2}\cdot\left(\sqrt{2}+\sqrt{\frac{\log\frac{1}{\delta}}{d}}\right)\right)^s\cdot I$$

$$\preceq \mathbb{E}\left[\left(I - \eta\hat{\tilde{X}}_1\hat{\tilde{X}}_1^T\right)\ldots\left(I - \eta\hat{\tilde{X}}_j\hat{\tilde{X}}_j^T\right)\right] + j\cdot\max_{s=1}^{j}(40j\cdot(1-\epsilon^2)^{u/2})^s\cdot I$$

Therefore, for sufficiently large $u$, the lemma is proved. $\qquad\square$

Lemma 21 establishes the per buffer contraction rate, using 20. The rest of the proofs in this section is devoted to establishing the contraction of the process where the vectors are sampled from iid buffers.

**Lemma 21.** *Let $X_0$ be the first vector in the buffer. If $u > \frac{2}{\epsilon^2}\log\frac{300000\pi dB}{\epsilon}$, and $\|X_0\| \leq 1 + \sqrt{\frac{\log\frac{1}{\delta}}{d}}$, then*

$$\mathbb{E}\left[\left(I - \eta\hat{X}_{u+B}\hat{X}_{u+B}^T\right)\ldots\left(I - \eta\hat{X}_{u+1}\hat{X}_{u+1}^T\right)\left(I - \eta\hat{X}_{u+1}\hat{X}_{u+1}^T\right)\ldots\left(I - \eta\hat{X}_{u+B}\hat{X}_{u+B}^T\right)|X_0\right]$$

$$\preceq \left(1 - \frac{1}{8}\cdot\frac{\epsilon B}{40d\pi}\right)I.$$

*Proof.*

$$\mathbb{E}\left[\left(I - \eta\hat{X}_{u+B}\hat{X}_{u+B}^T\right)\ldots\left(I - \eta\hat{X}_{u+1}\hat{X}_{u+1}^T\right)\left(I - \eta\hat{X}_{u+1}\hat{X}_{u+1}^T\right)\ldots\left(I - \eta\hat{X}_{u+B}\hat{X}_{u+B}^T\right)|X_0\right]$$

$$\preceq \mathbb{E}\left[\left(I - \eta\hat{\tilde{X}}_{u+B}\hat{\tilde{X}}_{u+B}^T\right)\ldots\left(I - \eta\hat{\tilde{X}}_{u+1}\hat{\tilde{X}}_{u+1}^T\right)\left(I - \eta\hat{\tilde{X}}_{u+1}\hat{\tilde{X}}_{u+1}^T\right)\ldots\left(I - \eta\hat{\tilde{X}}_{u+B}\hat{\tilde{X}}_{u+B}^T\right)|X_0\right]$$

$$+\sum_{s=1}^{2B}\binom{2B}{s}\left(4\left(1-\epsilon^2\right)^{u/2}\cdot\left(\sqrt{2}+\sqrt{\frac{\log\frac{1}{\delta}}{d}}\right)\right)^s\cdot I$$

$$= \mathbb{E}\left[\left(I - \eta\hat{\tilde{X}}_{u+B}\hat{\tilde{X}}_{u+B}^T\right)\ldots\left(I - \eta\hat{\tilde{X}}_{u+1}\hat{\tilde{X}}_{u+1}^T\right)\left(I - \eta\hat{\tilde{X}}_{u+1}\hat{\tilde{X}}_{u+1}^T\right)\ldots\left(I - \eta\hat{\tilde{X}}_{u+B}\hat{\tilde{X}}_{u+B}^T\right)\right]$$

$$+\left(\sum_{s=1}^{2B}\binom{2B}{s}\left(4\left(1-\epsilon^2\right)^{u/2}\cdot\left(\sqrt{2}+\sqrt{\frac{\log\frac{1}{\delta}}{d}}\right)\right)^s\right)\cdot I\left(\text{ since }\hat{\tilde{X}}_{u+s}\text{ are all independent of }X_0\right)$$

$$\preceq \left(1 - \frac{1}{4}\cdot\frac{\epsilon B}{40d\pi}\right)I + 2B\cdot\max_{s=1}^{2B}(40B\cdot(1-\epsilon^2)^{u/2})^s\cdot I\text{ ( from Lemma 22 )}$$

$$\preceq \left(1 - \frac{1}{4}\cdot\frac{\epsilon B}{40d\pi}\right)I + 80B^2\cdot(1-\epsilon^2)^{u/2}\cdot I$$

$$\preceq \left(1 - \frac{1}{4}\cdot\frac{\epsilon B}{40d\pi}\right)I + \frac{1}{8}\cdot\frac{\epsilon B}{40d\pi}\cdot I\text{ ( from hypothesis on }u)$$

$$= \left(1 - \frac{1}{8}\cdot\frac{\epsilon B}{40d\pi}\right)I.$$

This finishes the proof. $\qquad\square$

We now define $H$ as $\frac{1}{B}\sum_{j=u}^{S}\tilde{X}_j\tilde{X}_j^T$, where $\tilde{X}_u, \tilde{X}_{u+1},\ldots\tilde{X}_S$ are the vectors that we are sampling from in the parallel process (where $\tilde{X}_0$ is sampled i.i.d. from $\mathcal{N}(0,\frac{1}{d})$). For the sake of convenience, for the rest of this section we also say $\hat{X}_i$ is the sampled vector at the $i$-th iteration, where the samples are taken from the set of $\tilde{X}$s coming from the parallel process.

**Lemma 22.** *Suppose $\frac{\epsilon B}{40\pi}$ of the eigenvalues of $H$ are larger than or equal to $\frac{1}{B}$. Then*

$$\mathbb{E}\left[(w_B - w^*)(w_B - w^*)^T\right] \preceq \left[\frac{3}{4} \cdot \frac{\epsilon B}{40 d\pi} + \left(1 - \frac{\epsilon B}{40 d\pi}\right)\right] I$$

*Proof.* After iterating through 1 buffer, we have:

$$\mathbb{E}\left[(w_B - w^*)(w_B - w^*)^T\right]$$

$$\mathbb{E}\left[\mathbb{E}\left[\left(I - \eta \hat{X}_B \hat{X}_B^T\right) \dots \left(I - \eta \hat{X}_1 \hat{X}_1^T\right)\left(I - \eta \hat{X}_1 \hat{X}_1^T\right) \dots \left(I - \eta \hat{X}_B \hat{X}_B^T\right) | \tilde{X}_1, \dots \tilde{X}_B\right]\right]$$

Suppose that by Lemma 17, we have that each $\|X_j\|_2^2 \le 2$ with high probability. Then when $\eta = \frac{1}{2}$, $\mathbb{E}\left[\left(I - \eta \hat{X}_1 \hat{X}_1^T\right)\left(I - \eta \hat{X}_1 \hat{X}_1^T\right)\right] \preceq I - \eta H$ in the PSD sense.

So now we can write:

$$\mathbb{E}\left[\left(I - \eta \hat{X}_B \hat{X}_B^T\right) \dots \left(I - \eta \hat{X}_1 \hat{X}_1^T\right)\left(I - \eta \hat{X}_1 \hat{X}_1^T\right) \dots \left(I - \eta \hat{X}_B \hat{X}_B^T\right) | \tilde{X}_1, \dots \tilde{X}_B\right]$$

$$\preceq \mathbb{E}\left[\left(I - \eta \hat{X}_B \hat{X}_B^T\right) \dots \left(I - \eta \hat{X}_2 \hat{X}_2^T\right)(I - \eta H)\left(I - \eta \hat{X}_2 \hat{X}_2^T\right) \dots \left(I - \eta \hat{X}_B \hat{X}_B^T\right) | \tilde{X}_1, \dots \tilde{X}_B\right]$$

$$= \mathbb{E}\left[\left[\prod_{i=2}^{B}\left(I - \eta \hat{X}_i \hat{X}_i^T\right)\right]\left[\prod_{i=2}^{B}\left(I - \eta \hat{X}_i \hat{X}_i^T\right)\right]^T | \tilde{X}_1, \dots \tilde{X}_B\right]$$

$$- \eta \mathbb{E}\left[\left[\prod_{i=2}^{B}\left(I - \eta \hat{X}_i \hat{X}_i^T\right)\right] H \left[\prod_{i=2}^{B}\left(I - \eta \hat{X}_i \hat{X}_i^T\right)\right]^T | \tilde{X}_1, \dots \tilde{X}_B\right]$$

$$\preceq \mathbb{E}\left[\left[\prod_{i=2}^{B}\left(I - \eta \hat{X}_i \hat{X}_i^T\right)\right]\left[\prod_{i=2}^{B}\left(I - \eta \hat{X}_i \hat{X}_i^T\right)\right]^T | \tilde{X}_1, \dots \tilde{X}_B\right]$$

$$- \eta \mathbb{E}\left[\left[\prod_{i=3}^{B}\left(I - \eta \hat{X}_i \hat{X}_i^T\right)\right](I - \eta H) H (I - \eta H)\left[\prod_{i=3}^{B}\left(I - \eta \hat{X}_i \hat{X}_i^T\right)\right]^T | \tilde{X}_1, \dots \tilde{X}_B\right]$$

where the last inequality comes from the fact that
$$\mathbb{E}\left[\left(I - \eta \hat{X}_i \hat{X}_i^T\right) S \left(I - \eta \hat{X}_i \hat{X}_i^T\right) | \tilde{X}_1, \dots \tilde{X}_B\right] \succeq (I - \eta H) S (I - \eta H) \text{ for } S \succeq 0.$$

Recursing on this inequality gives us that

$$\mathbb{E}\left[\left(I - \eta \hat{X}_B \hat{X}_B^T\right) \dots \left(I - \eta \hat{X}_1 \hat{X}_1^T\right)\left(I - \eta \hat{X}_1 \hat{X}_1^T\right) \dots \left(I - \eta \hat{X}_B \hat{X}_B^T\right) | \tilde{X}_1, \dots \tilde{X}_B\right]$$

$$\preceq \mathbb{E}\left[I - \eta \sum_{k=0}^{n-1} (I - \eta H)^k H (I - \eta H)^k | \tilde{X}_1, \dots \tilde{X}_B\right]$$

Suppose that $\lambda$ is an eigenvalue of $H$. Then using the formulas for geometric series, it follows that $\frac{1-\eta\lambda}{2-\eta\lambda} + \frac{(1-\eta\sigma)^{2B}}{2-\eta\lambda}$ is an eigenvalue of $I - \eta \sum_{k=0}^{n-1} (I - \eta H)^k H (I - \eta H)^k$.

Suppose that $\frac{\epsilon B}{40\pi}$ of the eigenvalues of $H$ are larger than or equal to $\frac{1}{B}$. For those eigenvalues $\frac{1-\eta\lambda}{2-\eta\lambda} + \frac{(1-\eta\sigma)^{2B}}{2-\eta\lambda} \le \frac{1}{2} + (1 - \eta\lambda)^{2B} \le \frac{1}{2} + (1 - \frac{1}{2B})^{2B} \le \frac{3}{4}$, where we use $\eta = \frac{1}{2}$ without loss of generality, and the fact that $\left(1 - \frac{1}{2B}\right)^{2B} \le \frac{1}{4}$.

Therefore,

$$\mathbb{E}\left[\mathbb{E}\left[\left(I - \eta\hat{X}_B\hat{X}_B^T\right)\ldots\left(I - \eta\hat{X}_1\hat{X}_1^T\right)\left(I - \eta\hat{X}_1\hat{X}_1^T\right)\ldots\left(I - \eta\hat{X}_B\hat{X}_B^T\right)|\tilde{X}_1,\ldots\tilde{X}_B\right]\right]$$
$$\preceq \left[\frac{3}{4}\cdot\frac{\epsilon B}{40d\pi} + \left(1 - \frac{\epsilon B}{40d\pi}\right)\right] I$$

□

**Lemma 23.** $\frac{\epsilon B}{40\pi}$ *of the eigenvalues of $H$ are larger than or equal to $\frac{1}{B}$ when $d \geq B^4 C \log(\frac{1}{\beta})$ for some constant $C$, $\epsilon < 0.21$ and $B = \frac{1}{\epsilon^7}$.*

*Proof.* $H = \frac{1}{B}\sum_{j=1}^{B} X_j X_j^T = \frac{1}{B}XX^T$, where the $j$-th column of $X$ is $X_j$. The non-zero eigenvalues of $H$ are equivalent to the non-zero eigenvalues of the gram matrix $M = \frac{1}{B}X^TX$. We can characterize each entry of $M$. For $j \geq i$,

$$X_j^T X_i = \left((1-\epsilon^2)^{\frac{j-i}{2}} X_i + \epsilon \sum_{k=i+1}^{j} (1-\epsilon^2)^{\frac{j-k}{2}} G_k\right)^T X_i$$

$$= (1-\epsilon^2)^{\frac{j-i}{2}}\|X_i\|^2 + \epsilon \sum_{k=i+1}^{j} (1-\epsilon^2)^{\frac{j-k}{2}} G_k^T X_i$$

Define Toeplitz matrix $Z$ with the following Toeplitz structure, $Z_{ij} = \frac{1}{B}(1-\epsilon^2)^{\frac{|i-j|}{2}}$ for $1 \leq i, j \leq B$. Then we can write $M = Z + E$. Lemma 24 establishes that $\frac{\epsilon B}{40\pi}$ of the eigenvalues of $Z$ are larger than or equal to $\frac{2}{B}$. By Weyl's inequality, the corresponding eigenvalues in $M$ can be perturbed by at most $\|E\|_F$, which we bound below to be within $\frac{1}{B}$. Therefore $\frac{\epsilon B}{40\pi}$ of the eigenvalues of $H$ are larger than or equal to $\frac{1}{B}$.

We conclude the proof with the analysis of the Frobenius norm of $E$.

Note that $\epsilon \sum_{k=i+1}^{j} (1-\epsilon^2)^{\frac{j-k}{2}} G_k \sim \mathcal{N}(0, \sigma^2)$, where $\sigma^2 \leq \frac{1}{d}$.

By Lemmas 17 and 18, we have that for $j \geq i$,

$$|E[i][j]| \leq \frac{c}{\sqrt{d}}\log\left(\frac{1}{\beta}\right)\left((1-\epsilon^2)^{\frac{j-i}{2}} + 1\right)$$

$$= \frac{2c}{\sqrt{d}}\log\left(\frac{1}{\beta}\right)$$

So then the Frobenius norm of $E$, $\|E\|_F \leq \sqrt{B^2\cdot\frac{c}{d}\log\left(\frac{1}{\beta}\right)} \leq B\cdot\frac{c}{\sqrt{d}}\log\left(\frac{1}{\beta}\right)$ for some constant $c$. Therefore, $d \geq B^4 C$ suffices for $\|E\|_F \leq \frac{1}{B}$, where $C$ is some constant. □

**Lemma 24.** $\frac{\epsilon B}{40\pi}$ *of the eigenvalues of $Z$ are larger than or equal to $\frac{2}{B}$ when $\epsilon < 0.21$ and $B = \frac{1}{\epsilon^7}$.*

*Proof.* To study the eigenvalues of $Z$, we first study the eigenvalues of the circulant matrix $C$, where the first row of $C$ has the following entries:

If $B$ is even: $C[1][j] = Z[1][j]$ if $1 \leq j \leq \frac{B}{2}$, $C[1][\frac{B}{2}+1] = 0$, and $C[1][\frac{B}{2}+j] = Z[1][\frac{B}{2}-j+2]$ for $2 \leq j \leq \frac{B}{2}$

If $B$ is odd: $C[1][j] = Z[1][j]$ if $1 \leq j \leq \frac{B+1}{2}$, $C[1][\frac{B+1}{2}+j] = Z[1][\frac{B+1}{2}-j+1]$ for $1 \leq j \leq \frac{B-1}{2}$.

The circulent matrices have the following eigenstructure. For simplicity, let $c_j = C[1][j]$. Then $\lambda_j = c_1 + c_2 w_j + c_3 w_j^2 + \ldots c_B w_j^{B-1}$, where $w_j$ is the $j$-th root of unity.

We first claim that the eigenvalues of the circulent matrix closely approximate the eigenvalues of the Topelitz matrix for sufficiently high $B$.

We first write $Z = C + P$, where $P$ is a perturbation matrix. Let $\lambda_1(C) \geq \cdots \geq \lambda_B(C)$ be the eigenvalues of $C$ in descending order. We establish in Lemma 25 that $\lambda_{\frac{\epsilon B}{20\pi}}(C) \geq \frac{9}{B}$. Moreover, in Lemma 26 we analyze $P$ and establish that $\lambda_{B+1-\frac{\epsilon B}{40\pi}}(P) \geq \frac{-7}{B}$. Therefore, by the generalized Weyl's theorem, then it follows that $\lambda_1(M) \ldots \lambda_{\frac{\epsilon B}{40\pi}}(M) \geq \frac{2}{B}$.

$\square$

**Lemma 25.** *For $\epsilon < 0.21$, at least $\frac{\epsilon B}{20\pi}$ of the eigenvalues of $C$ are greater than or equal to $\frac{9}{B}$.*

*Proof.* We first characterize all the eigenvalues $\lambda_j$ of $C$, and then we show that for odd $j$, $j \leq \frac{\epsilon B}{10\pi}$, $\lambda_j \geq \frac{9}{B}$ when $\epsilon < 0.21$. We now characterize $\lambda_j$. Using the formula for the eigenvalues of a circulent matrix, it follows that $\lambda_j = c_1 + c_2 w_j + c_3 w_j^2 + \ldots c_B w_j^{B-1}$, where $w_j$ is the $j$-th root of unity, ie $w_j = \cos\left(\frac{2\pi j}{B}\right) + i \sin\left(\frac{2\pi j}{B}\right)$. For simplicity and without loss of generality, suppose $B$ is odd, so that $\frac{B-1}{2}$ is an integer. Using the symmetry of our circulent matrix as well as the symmetry of powers of the roots of unity, we have that:

$$\lambda_j = \frac{1}{B}\left(1 + \sum_{k=1}^{\frac{B-1}{2}}(1-\epsilon^2)^{\frac{k}{2}} w_j^k + \sum_{\ell=\frac{B-1}{2}+1}^{B-1}(1-\epsilon^2)^{\frac{B-\ell}{2}} w_j^\ell\right)$$

Note that the $(1-\epsilon^2)$ coefficients in the two different summations are equal when $\ell = B - k$. Then we can rewrite as:

$$\lambda_j = \frac{1}{B}\left(1 + \sum_{k=1}^{\frac{B-1}{2}}(1-\epsilon^2)^{\frac{k}{2}}(w_j^k + w_j^{B-k})\right)$$

We can further write

$$w_j^k + w_j^{B-k} = \cos\left(\frac{2\pi j \cdot k}{B}\right) + i\sin\left(\frac{2\pi j \cdot k}{B}\right) + \cos\left(\frac{2\pi j(B-k)}{B}\right) + i\sin\left(\frac{2\pi j(B-k)}{B}\right)$$

$$= \cos\left(\frac{2\pi j \cdot k}{B}\right) + i\sin\left(\frac{2\pi j \cdot k}{B}\right) + \cos\left(\frac{2\pi j(B)}{B} - \frac{2\pi jk}{B}\right) + i\sin\left(\frac{2\pi j(B)}{B} - \frac{2\pi jk}{B}\right)$$

$$= 2\cos\left(\frac{2\pi j \cdot k}{B}\right)$$

Therefore, we have:

$$\lambda_j = \frac{2}{B}\left(\sum_{k=0}^{\frac{B-1}{2}}(1-\epsilon^2)^{\frac{k}{2}}\cos\left(\frac{2\pi k \cdot j}{B}\right)\right) - \frac{1}{B}$$

$$= \frac{2}{B}Re\left(\sum_{k=0}^{\frac{B-1}{2}}(1-\epsilon^2)^{\frac{k}{2}} w_j^k\right) - \frac{1}{B}$$

$$= \frac{2}{B}Re\left(\frac{1 - \left(\sqrt{1-\epsilon^2}\right)^{\frac{B+1}{2}} w_j^{\frac{B+1}{2}}}{1 - \sqrt{1-\epsilon^2}w_j}\right) - \frac{1}{B}$$

$$= \frac{2}{B}Re\left(\frac{\left(1 - \sqrt{1-\epsilon^2}^{\frac{B+1}{2}}(\cos(\pi j + \frac{\pi j}{B}) + i\sin(\pi j + \frac{\pi j}{B}))\right)\left(1 - \sqrt{1-\epsilon^2}\cos(\frac{2\pi j}{B}) + i\sqrt{1-\epsilon^2}\sin(\frac{2\pi j}{B})\right)}{2 - \epsilon^2 - 2\sqrt{1-\epsilon^2}\cos\left(\frac{2\pi j}{B}\right)}\right) - \frac{1}{B}$$

$$= \frac{2}{B}\left(\frac{\left(1 - \sqrt{1-\epsilon^2}^{\frac{B+1}{2}}\cos(\pi j + \frac{\pi j}{B})\right)\left(1 - \sqrt{1-\epsilon^2}\cos(\frac{2\pi j}{B})\right) + \sqrt{1-\epsilon^2}^{\frac{B+1}{2}+1}\sin(\frac{2\pi j}{B})\sin(\pi j + \frac{\pi j}{B})}{2 - \epsilon^2 - 2\sqrt{1-\epsilon^2}\cos\left(\frac{2\pi j}{B}\right)}\right) - \frac{1}{B}$$

When $j$ is odd, then for sufficiently small $j$ and sufficiently large $B$, we can say that:

$$\left(\frac{\left(1 - \sqrt{1-\epsilon^2}^{\frac{B+1}{2}}\cos(\pi j + \frac{\pi j}{B})\right)\left(1 - \sqrt{1-\epsilon^2}\cos(\frac{2\pi j}{B})\right) + \sqrt{1-\epsilon^2}^{\frac{B+1}{2}+1}\sin(\frac{2\pi j}{B})\sin(\pi j + \frac{\pi j}{B})}{2 - \epsilon^2 - 2\sqrt{1-\epsilon^2}\cos\left(\frac{2\pi j}{B}\right)}\right)$$

$$\geq \frac{1}{2} \frac{\left(1 - \sqrt{1-\epsilon^2}\cos\left(\frac{2\pi j}{B}\right)\right)}{2 - \epsilon^2 - 2\sqrt{1-\epsilon^2}\cos\left(\frac{2\pi j}{B}\right)} \geq \frac{1}{2} \frac{\left(1 - \sqrt{1-\epsilon^2}\right)}{2 - \epsilon^2 - 2\sqrt{1-\epsilon^2}\cos\left(\frac{2\pi j}{B}\right)} \geq \frac{1}{2} \frac{\left(1 - \sqrt{1-\epsilon^2}\right)}{2 - \epsilon^2 - 2\sqrt{1-\epsilon^2}\cos\left(\frac{\epsilon}{5}\right)}$$

Standard computation shows that the last term is $\geq 5$ when $\epsilon < 0.21$ so that $\lambda_j \geq \frac{9}{B}$.

$\square$

**Lemma 26.** *Let $\lambda_1(P) \geq \ldots \geq \lambda_B(P)$ be the eigenvalues of $P$ in descending order. Suppose that $\epsilon < 0.21$ and $B = \frac{1}{\epsilon^7}$. Then $\lambda_{B+1-\frac{\epsilon B}{40\pi}} \geq -\frac{7}{B}$.*

*Proof.* $P$ can be shown to have the following block form:

For even $B$,

$$P = \begin{pmatrix} 0 & A \\ A^T & 0 \end{pmatrix}$$

where $A$ is an $\frac{B}{2}$ square upper triangular matrix with $\frac{1}{B}\left(1-\epsilon^2\right)^{\frac{B}{4}}$ along the diagonal.

For odd $B$,

$$P = \begin{pmatrix} 0 & A \\ 0 & 0 \\ A^T & 0 \end{pmatrix}$$

where $A$ is an $\frac{B-1}{2}$ square upper triangular matrix with $\frac{1}{B}\left[(1-\epsilon^2)^{\frac{B+1}{4}} - (1-\epsilon^2)^{\frac{B-1}{4}}\right]$ along the diagonal. The eigenvalues of $P$ come in positive-negative pairs, so that $\lambda^2_{B+1-\frac{\epsilon B}{40\pi}} = \lambda^2_{\frac{\epsilon B}{40\pi}}$. Notice that $\sum_{i=1}^{B} \lambda_i^2 = \|P\|_F^2$, where $\|\cdot\|_F^2$ denotes the Frobenius norm. We will bound $\lambda^2_{\frac{\epsilon B}{40\pi}}$ using the Frobenius norm. Suppose for loss of generality that $B$ is odd. Then it follows that the Frobenius norm, $\|P\|_F^2 = 2 \cdot \|A\|_F^2$. So we can focus on $\|A\|_F^2$. Note that in this case, the circulent matrix $C$ has entries $C[1][\frac{B+1}{2} + j] = Z[1][\frac{B+1}{2} - j + 1] = \frac{1}{B}(1-\epsilon^2)^{\frac{B+1}{2}-j}{2}$ for $1 \leq j \leq \frac{B-1}{2}$, whereas the original Toeplitz matrix has entries $Z[1][\frac{B+1}{2} + j] = \frac{1}{B}(1-\epsilon^2)^{\frac{B+1}{2}+j-1}{2}$.

$$\|A\|_F^2 = \sum_{\ell=1}^{\frac{B-1}{2}} \sum_{j=1}^{\ell} \frac{1}{B^2}\left((1-\epsilon^2)^{\frac{\frac{B+1}{2}+j-1}{2}} - (1-\epsilon^2)^{\frac{\frac{B+1}{2}-j}{2}}\right)^2$$

$$= \frac{1}{B^2}(1-\epsilon^2)^{\frac{B+1}{2}} \sum_{\ell=1}^{\frac{B-1}{2}} \sum_{j=1}^{\ell}\left((1-\epsilon^2)^{\frac{j-1}{2}} - (1-\epsilon^2)^{\frac{-j}{2}}\right)^2$$

$$= \frac{1}{B^2}(1-\epsilon^2)^{\frac{B+1}{2}} \sum_{\ell=1}^{\frac{B-1}{2}} \sum_{j=1}^{\ell}\left((1-\epsilon^2)^{j-1} - 2(1-\epsilon^2)^{\frac{-1}{2}} + (1-\epsilon^2)^{-j}\right)$$

$$= \frac{1}{B^2}(1-\epsilon^2)^{\frac{B+1}{2}} \sum_{\ell=1}^{\frac{B-1}{2}} \sum_{j=1}^{\ell}-2(1-\epsilon^2)^{\frac{-1}{2}} + \frac{1}{B^2}(1-\epsilon^2)^{\frac{B+1}{2}} \sum_{\ell=1}^{\frac{B-1}{2}} \sum_{j=1}^{\ell}\left((1-\epsilon^2)^{j-1} + (1-\epsilon^2)^{-j}\right)$$

$$\leq \frac{1}{B^2}(1-\epsilon^2)^{\frac{B+1}{2}} \sum_{\ell=1}^{\frac{B-1}{2}} \sum_{j=1}^{\ell}\left((1-\epsilon^2)^{j-1} + (1-\epsilon^2)^{-j}\right) \tag{60}$$

Notice that

$$\sum_{j=1}^{\ell}(1-\epsilon^2)^{j-1} = \frac{1 - (1-\epsilon^2)^\ell}{1 - (1-\epsilon^2)} = \frac{1 - (1-\epsilon^2)^\ell}{\epsilon^2},$$

$$\sum_{j=1}^{\ell}(1-\epsilon^2)^{-j} = \frac{1}{1-\epsilon^2} + \left(\frac{1}{1-\epsilon^2}\right)^2 + \cdots \left(\frac{1}{1-\epsilon^2}\right)^{\ell}$$

$$= \frac{1}{1-\epsilon^2}\left(1 + \cdots \left(\frac{1}{1-\epsilon^2}\right)^{\ell-1}\right)$$

$$= \frac{1}{1-\epsilon^2}\left(\frac{1 - \left(\frac{1}{1-\epsilon^2}\right)^{\ell}}{1 - \frac{1}{1-\epsilon^2}}\right)$$

$$= \frac{1}{\epsilon^2}\left(\left(\frac{1}{1-\epsilon^2}\right)^{\ell} - 1\right)$$

Therefore, following line 60, we have:

$$\|A\|_F^2 \leq \frac{1}{B^2}(1-\epsilon^2)^{\frac{B+1}{2}} \cdot \frac{1}{\epsilon^2}\sum_{\ell=1}^{\frac{B-1}{2}}\left(\left(\frac{1}{1-\epsilon^2}\right)^{\ell} - (1-\epsilon^2)^{\ell}\right) \tag{61}$$

Notice that

$$\sum_{\ell=1}^{\frac{B-1}{2}}\left(\frac{1}{1-\epsilon^2}\right)^{\ell} = \frac{1}{1-\epsilon^2}\left(\frac{1 - \left(\frac{1}{1-\epsilon^2}\right)^{\frac{B-1}{2}}}{1 - \left(\frac{1}{1-\epsilon^2}\right)}\right) = \frac{\left(\frac{1}{1-\epsilon^2}\right)^{\frac{B-1}{2}} - 1}{\epsilon^2},$$

$$\sum_{\ell=1}^{\frac{B-1}{2}}(1-\epsilon^2)^{\ell} = (1-\epsilon^2)\left(\frac{1 - (1-\epsilon^2)^{\frac{B-1}{2}}}{1 - (1-\epsilon^2)}\right) = \frac{(1-\epsilon^2) - (1-\epsilon^2)^{\frac{B-1}{2}+1}}{\epsilon^2},$$

Therefore,

$$\|A\|_F^2 \leq \frac{1}{B^2}(1-\epsilon^2)^{\frac{B+1}{2}} \cdot \frac{1}{\epsilon^2}\sum_{\ell=1}^{\frac{B-1}{2}}\left(\left(\frac{1}{1-\epsilon^2}\right)^{\ell} - (1-\epsilon^2)^{\ell}\right)$$

$$= \frac{1}{B^2\epsilon^2}(1-\epsilon^2)^{\frac{B+1}{2}}\left(\frac{(1-\epsilon^2)^{-\left(\frac{B-1}{2}\right)} - 2 + (1-\epsilon^2)^{\frac{B-1}{2}+1}}{\epsilon^2} + 1\right)$$

$$= \frac{1}{B^2\epsilon^4}(1-\epsilon^2) - \frac{2(1-\epsilon^2)^{\frac{B+1}{2}}}{B^2\epsilon^4} + \frac{(1-\epsilon^2)^{B+1}}{B^2\epsilon^4} + \frac{1}{B^2\epsilon^2}(1-\epsilon^2)^{\frac{B+1}{2}}$$

$$\leq \frac{1}{B^2\epsilon^4}(1-\epsilon^2)$$

So we conclude that the Frobenius norm of $P$, satisfies:

$$\|P\|_F^2 \leq \frac{2}{B^2\epsilon^4}(1-\epsilon^2)$$

Therefore. $\lambda_{\frac{\epsilon B}{40\pi}}^2 \leq \frac{\frac{2}{B^2\epsilon^4}(1-\epsilon^2)}{\frac{\epsilon B}{40\pi}} = \frac{1}{B^2}\frac{80\pi}{B\epsilon^5}(1-\epsilon^2) = \frac{1}{B^2}\left(\epsilon^2 80\pi(1-\epsilon^2)\right)$. For our choice of $\epsilon, B$, we know that $\lambda_{\frac{\epsilon B}{40\pi}} \leq \frac{7}{B}$, Therefore, $\lambda_{B+1-\frac{\epsilon B}{40\pi}} \geq \frac{-7}{B}$. $\qquad\square$

### E.3 Variance Decay with Experience Replay

To analyze the variance, we start with $w_0^{\text{var}} = w^*$. The dynamics of SGD say that:

$$w_0^{\text{var}} - w^* = 0$$

$$w_1^{\text{var}} - w^* = \eta\hat{\xi}_1^{(1)}\hat{X}_1^{(1)}$$

$$w_{t+1}^{\text{var}} - w^* = \left(I - \eta \hat{X}_{t+1} \hat{X}_{t+1}^T\right)(w_t^{\text{var}} - w^*) + \eta \hat{\xi}_{t+1} \hat{X}_{t+1}$$

We let the superscript $(i)$ denote the $i$-th buffer index. Let $H^{(i)} = \frac{1}{B} \sum_{j=1}^{B} X_j^{(i)} X_j^{(i)T}$, where $X_j^{(i)}$ are the vectors that comprise the sampling pool from buffer $i$.

We produce our final $w$ by tail averaging over the last iterate of SGD from within each buffer $i$. Let $w_{iB}$ denote the last SGD iterate using buffer $i$. Then

$$w = \frac{1}{N} \sum_{i=1}^{N} w_{iB} , \tag{62}$$

where $N$ is the number of buffers.

Clearly,

$$\mathbb{E}[(w^{\text{var}} - w^*)(w^{\text{var}} - w^*)^T] = \frac{1}{N^2} \mathbb{E}\left[\sum_{i=1}^{N} w_{iB}^{\text{var}} \sum_{i=1}^{N} w_{iB}^{\text{var}T}\right]$$

**Theorem 12** (Variance Decay for SGD with Experience Replay for Gaussian AR Chain). *For any $\epsilon \leq 0.21$, if $B \geq \frac{1}{\epsilon^7}$ and $d = \Omega(B^4 \log(\frac{1}{\beta}))$, with probability at least $1 - \beta$, Algorithm 4 returns $w$ such that $\mathcal{L}(w^{\text{var}}) \leq \tilde{O}\left(\frac{\sigma^2 d \sqrt{\tau_{\text{mix}}}}{T}\right)$.*

*Proof.*

$$
\begin{aligned}
\mathbb{E}[(w^{\text{var}} - w^*)(w^{\text{var}} - w^*)^T] &= \mathbb{E}\left[\frac{1}{N} \sum_{k=1}^{N} (w_{kB}^{\text{var}} - w^*) \frac{1}{N} \sum_{k=1}^{N} (w_{kB}^{\text{var}} - w^*)^T\right] \\
&= \mathbb{E}\left[\frac{1}{N^2} \sum_{k=1}^{N} \sum_{\ell=1}^{N} (w_{kB}^{\text{var}} - w^*)(w_{\ell B}^{\text{var}} - w^*)^T\right] \\
&= \frac{1}{N^2} \mathbb{E}\left[\sum_{k>\ell} (w_{kB}^{\text{var}} - w^*)(w_{\ell B}^{\text{var}} - w^*)^T\right] + \frac{1}{N^2} \mathbb{E}\left[\sum_{k<\ell} (w_{kB}^{\text{var}} - w^*)(w_{\ell B}^{\text{var}} - w^*)^T\right] \\
&\quad + \frac{1}{N^2} \mathbb{E}\left[\sum_{k=1}^{N} (w_{kB}^{\text{var}} - w^*)(w_{kB}^{\text{var}} - w^*)^T\right] \\
&\preceq \frac{3\sigma^2}{N^2} \mathbb{E}\left[\sum_{\ell=1}^{N} \sum_{k=\ell+1}^{N} \prod_{j=\ell+1}^{k} \prod_{i=1}^{B} \left(I - \hat{X}_i^{(j)} \hat{X}_i^{(j)T}\right)\right] \\
&\quad + \frac{3\sigma^2}{N^2} \mathbb{E}\left[\left(\prod_{j=\ell+1}^{k} \prod_{i=1}^{B} \left(I - \hat{X}_i^{(j)} \hat{X}_i^{(j)T}\right)\right)^T\right] + \frac{3\sigma^2}{N^2} [N \cdot I] ,
\end{aligned}
$$

where the last line follows as a consequence of Lemma 27.

Now we can focus on $\mathbb{E}\left[\prod_{j=\ell+1}^{k} \prod_{i=1}^{B} \left(I - \hat{X}_i^{(j)} \hat{X}_i^{(j)T}\right)\right]$, which by spherical symmetry is equal to $c \cdot I$ for constant $c$.

Following Lemma 20, let $\hat{\tilde{X}}_i^{(j)}$ be a sample from $\tilde{X}_{u+s}^{(j)} := (1 - \epsilon^2)^{(u+s)/2} \tilde{X}_0^{(j)} + \epsilon \sum_{t=1}^{u+s} (1 - \epsilon^2)^{(t-(u+s))/2} G_t$, where $\tilde{X}_0^{(j)}$ is sampled independently from $\mathcal{N}(0, \frac{1}{\sqrt{d}} I_d)$.

$$\mathbb{E}\left[\prod_{j=\ell+1}^{k}\prod_{i=1}^{B}\left(I-\hat{X}_i^{(j)}\hat{X}_i^{(j)T}\right)\right] = \mathbb{E}\left[\prod_{j=\ell+1}^{k}\prod_{i=1}^{B}\left(I-\hat{\hat{X}}_i^{(j)}\hat{\hat{X}}_i^{(j)T}+\hat{\hat{X}}_i^{(j)}\hat{\hat{X}}_i^{(j)T}-\hat{X}_i^{(j)}\hat{X}_i^{(j)T}\right)\right]$$

$$\preceq \mathbb{E}\left[\prod_{j=\ell+1}^{k}\prod_{i=1}^{B}\left(I-\hat{\hat{X}}_i^{(j)}\hat{\hat{X}}_i^{(j)T}\right)\right] + c\cdot(NB)^2(1-\epsilon^2)^{u/2}\cdot I$$

where $c$ is an appropriate constant.

By Lemma 23, $\mathbb{E}\left[\sum_{\ell=1}^{N}\sum_{k=\ell+1}^{N}\prod_{j=\ell+1}^{k}\prod_{i=1}^{B}\left(I-\hat{\hat{X}}_i^{(j)}\hat{\hat{X}}_i^{(j)T}\right)\right] \preceq \sum_{\ell=1}^{N}\sum_{k=\ell+1}^{N}\left[1-\frac{\epsilon B}{160\pi d}\right]^{k-\ell}\cdot I$.

Therefore, $\mathbb{E}[(w^{\mathrm{var}}-w^*)(w^{\mathrm{var}}-w^*)^T] \preceq \frac{6\sigma^2}{N^2}\sum_{\ell=1}^{N}\sum_{k=\ell}^{N}\left[1-\frac{\epsilon B}{160\pi d}\right]^{k-\ell} + 3\sigma^2(NB)^2(1-\epsilon^2)^{u/2}\cdot I$.

$$\sum_{\ell=1}^{N}\sum_{k=\ell}^{N}\left[1-\frac{\epsilon B}{160\pi d}\right]^{k-\ell} = \sum_{\ell=1}^{N}\sum_{i=0}^{N-\ell}\left[1-\frac{\epsilon B}{160\pi d}\right]^{i}$$

$$= \sum_{\ell=1}^{N}\frac{160\pi d}{\epsilon B} - \frac{160\pi d}{\epsilon B}\sum_{\ell=1}^{N}\left[1-\frac{\epsilon B}{160\pi d}\right]^{N-\ell+1}$$

$$\leq N\cdot\frac{160\pi d}{\epsilon B}$$

Putting everything together, we have that $\mathbb{E}[(w^{\mathrm{var}}-w^*)(w^{\mathrm{var}}-w^*)^T] \preceq \frac{6\sigma^2}{N}\cdot\frac{160\pi d}{\epsilon B}\cdot I + 3\sigma^2(NB)^2(1-\epsilon^2)^{u/2}\cdot I$.

Therefore, when $u = \frac{2}{\epsilon^2}\log\frac{300000\pi d^2\sigma^6}{\epsilon^2\delta}$, it follows that $\mathcal{L}(w^{\mathrm{var}}) \leq O(\frac{\sigma^2 d}{\epsilon T})$. □

**Lemma 27.** $\mathbb{E}[(w_t^{\mathrm{var}}-w^*)(w_t^{\mathrm{var}}-w^*)^T|X_1^{(1)},\dots X_S^{(T/S)}] \preceq 3\sigma^2 I$ for all $t$.

*Proof.* Proof by induction.

In the first iterate of the first buffer, we have:

$$(w_1^{\mathrm{var}}-w^*)(w_1^{\mathrm{var}}-w^*)^T = \eta^2(\hat{\xi}_1^{(1)})^2\hat{X}_1^{(1)}\hat{X}_1^{(1)T}$$

Since each $\|X_j\|_2^2 \leq 2$ with high probability, and when $\eta = \frac{1}{2}$, we have:

$$\mathbb{E}\left[(w_1^{\mathrm{var}}-w^*)(w_1^{\mathrm{var}}-w^*)^T|X_1,\dots X_B\right] = \eta^2\sigma^2 H^{(1)} \preceq \sigma^2 I$$

For the second iterate, we have:

$$w_2^{\mathrm{var}}-w^* = \left(I-\eta\hat{X}_2^{(1)}\hat{X}_2^{(1)T}\right)\hat{\xi}_1^{(1)}\hat{X}_1^{(1)} + \eta\hat{\xi}_2^{(1)}\hat{X}_2^{(1)}$$

$$\mathbb{E}\left[(w_2^{\mathrm{var}}-w^*)(w_2^{\mathrm{var}}-w^*)^T|X_1,\dots X_B\right] = \mathbb{E}[\left(I-\eta\hat{X}_2^{(1)}\hat{X}_2^{(1)T}\right)\eta^2(\hat{\xi}_1^{(1)})^2\hat{X}_1^{(1)}\hat{X}_1^{(1)T}\left(I-\eta\hat{X}_2^{(1)}\hat{X}_2^{(1)T}\right)$$

$$+ \left(I-\eta\hat{X}_2^{(1)}\hat{X}_2^{(1)T}\right)\eta\hat{\xi}_1^{(1)}\hat{\xi}_2^{(1)}\hat{X}_1^{(1)}\hat{X}_2^{(1)T}$$

$$+ \eta\hat{\xi}_1^{(1)}\hat{\xi}_2^{(1)}\hat{X}_1^{(1)}\hat{X}_2^{(1)T}\left(I-\eta\hat{X}_2^{(1)}\hat{X}_2^{(1)T}\right)$$

$$+ \eta^2(\hat{\xi}_2^{(1)})^2\hat{X}_2^{(1)}\hat{X}_2^{(1)T}|X_1\dots X_B]$$

$$\preceq \sigma^2(I-\eta H^{(1)}+\eta H^{(1)}+\eta H^{(1)}+\eta^2 H^{(1)})$$

$$\preceq 3\sigma^2 I$$

□

Suppose that in the first buffer, for $k \leq B$,
$$\mathbb{E}\left[(w_{k-1}^{\text{var}} - w^*)(w_{k-1}^{\text{var}} - w^*)^T | X_1, \ldots X_B\right] \preceq 3\sigma^2 I$$

Then we look at
$$\mathbb{E}\left[(w_k^{\text{var}} - w^*)(w_k^{\text{var}} - w^*)^T | X_1, \ldots X_B\right] = \mathbb{E}[\left(I - \eta \hat{X}_k^{(1)} \hat{X}_k^{(1)T}\right)(w_{k-1}^{\text{var}} - w^*)(w_{k-1}^{\text{var}} - w^*)^T \left(I - \eta \hat{X}_k^{(1)} \hat{X}_k^{(1)T}\right)$$
$$+ \left(I - \eta \hat{X}_k^{(1)} \hat{X}_k^{(1)T}\right)(w_{k-1}^{\text{var}} - w^*)\eta \hat{\xi}_k^{(1)} \hat{X}_k^{(1)T}$$
$$+ \eta \hat{\xi}_k^{(1)} \hat{X}_k^{(1)}(w_{k-1}^{\text{var}} - w^*)^T \left(I - \eta \hat{X}_k^{(1)} \hat{X}_k^{(1)T}\right)$$
$$+ \eta^2 (\hat{\xi}_k^{(1)})^2 \hat{X}_k^{(1)} \hat{X}_k^{(1)T} | X_1 \ldots X_B]$$

Notice that $w_{k-1}^{\text{var}} - w^* = \sum_{j=1}^{k-1} \prod_{i=j+1}^{k-1} \left(I - \eta \hat{X}_i^{(1)} \hat{X}_i^{(1)T}\right) \eta \hat{\xi}_j^{(1)} \hat{X}_j^{(1)}$

We first focus on the cross term:
$$\mathbb{E}\left[\left(I - \eta \hat{X}_k^{(1)} \hat{X}_k^{(1)T}\right)(w_{k-1}^{\text{var}} - w^*)\eta \hat{\xi}_k^{(1)} \hat{X}_k^{(1)T} | X_1, \ldots X_B\right]$$
$$= \mathbb{E}\left[\left(I - \eta \hat{X}_k^{(1)} \hat{X}_k^{(1)T}\right) \sum_{j=1}^{k-1} \prod_{i=j+1}^{k-1} \left(I - \eta \hat{X}_i^{(1)} \hat{X}_i^{(1)T}\right) \eta \hat{\xi}_j^{(1)} \hat{X}_j^{(1)} \left(\eta \hat{\xi}_k^{(1)} \hat{X}_k^{(1)T}\right) | X_1, \ldots X_B\right]$$

Notice that by the independence of the noise, only those terms where $\hat{X}_j = \hat{X}_k$ will be non-zero (and this event happens with probability $\frac{1}{B}$ for each $j$). Moreover, note that
$$\mathbb{E}\left[\eta \hat{X}_k^{(1)} \hat{X}_k^{(1)T} \sum_{j=1}^{k-1} \prod_{i=j+1}^{k-1} \left(I - \eta \hat{X}_i^{(1)} \hat{X}_i^{(1)T}\right) \eta \hat{\xi}_k^{(1)} \hat{X}_k^{(1)} \left(\eta \hat{\xi}_k^{(1)} \hat{X}_k^{(1)T}\right) | X_1, \ldots X_B\right]$$
$$= \mathbb{E}\left[\eta \hat{X}_k^{(1)} \hat{X}_k^{(1)T} \sum_{j=1}^{k-1} \left(I - \eta H^{(1)}\right)^{k-j-2} \eta \hat{\xi}_k^{(1)} \hat{X}_k^{(1)} \left(\eta \hat{\xi}_k^{(1)} \hat{X}_k^{(1)T}\right) | X_1, \ldots X_B\right]$$
$$\succeq 0$$

Therefore, we have:
$$\mathbb{E}\left[\left(I - \eta \hat{X}_k^{(1)} \hat{X}_k^{(1)T}\right)(w_{k-1}^{\text{var}} - w^*)\eta \hat{\xi}_k^{(1)} \hat{X}_k^{(1)T} | X_1, \ldots X_B\right]$$
$$\preceq \frac{\sigma^2}{B}\left[\sum_{j=1}^{k-1} \prod_{i=j+1}^{k-1} \left(I - \eta H^{(1)}\right) \eta H^{(1)}\right]$$
$$\preceq \eta \sigma^2 \frac{k-1}{B} H^{(1)} \preceq \eta \sigma^2 H^{(1)}$$

Therefore,
$$\mathbb{E}\left[(w_k^{\text{var}} - w^*)(w_k^{\text{var}} - w^*)^T | X_1, \ldots X_B\right] \preceq 3\sigma^2(I - \eta H^{(1)}) + 2\eta \sigma^2 H^{(1)} + \eta^2 \sigma^2 H^{(1)}$$
$$\preceq 3\sigma^2 I$$

Therefore, in the first buffer, $\mathbb{E}\left[(w_k^{\text{var}} - w^*)(w_k^{\text{var}} - w^*)^T | X_1, \ldots X_B\right] \preceq 3\sigma^2 I$ for all $k \leq B$.

For the first iterate using the second buffer, because the cross terms are 0 by independent noise, it is easy to show that
$$\mathbb{E}\left[(w_{B+1}^{\text{var}} - w^*)(w_{B+1}^{\text{var}} - w^*)^T | X_1^{(2)}, \ldots X_B^{(2)}\right] = \mathbb{E}[\left(I - \eta \hat{X}_1^{(2)} \hat{X}_1^{(2)T}\right)(w_B^{\text{var}} - w^*)(w_B^{\text{var}} - w^*)^T \left(I - \eta \hat{X}_1^{(2)} \hat{X}_1^{(2)T}\right)$$
$$+ \eta^2 (\hat{\xi}_1^{(2)})^2 \hat{X}_1^{(2)} \hat{X}_1^{(2)T} | X_1^{(2)}, \ldots X_B^{(2)}]$$
$$\preceq 3\sigma^2 I$$

For subsequent iterates in the second buffer,

We write out $w_{B+k-1}^{\text{var}} - w^* = \prod_{j=1}^{k} \left( I - \eta \hat{X}_j^{(2)} \hat{X}_j^{(2)T} \right) (w_B^{\text{var}} - w^*) +$

$\sum_{j=1}^{k-1} \prod_{i=j+1}^{k-1} \left( I - \eta \hat{X}_i^{(2)} \hat{X}_i^{(2)T} \right) \eta \hat{\xi}_j^{(2)} \hat{X}_j^{(2)}.$

Therefore, the cross term

$$\mathbb{E}\left[ \left( I - \eta \hat{X}_{B+k}^{(2)} \hat{X}_{B+k}^{(2)T} \right) (w_{B+k-1}^{\text{var}} - w^*) \eta \hat{\xi}_{B+k}^{(2)} \hat{X}_{B+k}^{(2)} | X_1^{(2)}, \ldots X_B^{(2)} \right]$$

$$= \mathbb{E}\left[ \left( I - \eta \hat{X}_{B+k}^{(2)} \hat{X}_{B+k}^{(2)T} \right) \sum_{j=1}^{k-1} \prod_{i=j+1}^{k-1} \left( I - \eta \hat{X}_i^{(2)} \hat{X}_i^{(2)T} \right) \eta \hat{\xi}_j^{(2)} \hat{X}_j^{(2)} \eta \hat{\xi}_{B+k}^{(2)} \hat{X}_{B+k}^{(2)} | X_1^{(2)}, \ldots X_B^{(2)} \right]$$

$$\preceq \eta \sigma^2 H^{(2)}$$

Therefore, by induction, we have that $\mathbb{E}[(w_t^{\text{var}} - w^*)(w_t^{\text{var}} - w^*)^T | X_1^{(1)}, \ldots X_S^{(T/S)}] \preceq 3\sigma^2 I$ for all $t$.

### E.4 Lower Bound for SGD with Constant Step Size

*Proof of Theorem 6.* We know that $w_{t+1} - w^* = (I - \eta X_t X_t^T)(w_t - w^*)$. We define:

$$\alpha_t = X_t^T(w_t - w^*)$$

$$\gamma_t = \|w_t - w^*\|$$

Then we have:

$$\alpha_{t+1} = X_{t+1}^T(w_{t+1} - w^*)$$

$$= \left( \sqrt{1-\epsilon^2} X_t + \epsilon G_{t+1} \right)^T \left( I - \eta X_t X_t^T \right) (w_t - w^*)$$

$$= \sqrt{1-\epsilon^2}\alpha_t + \epsilon G_{t+1}^T(w_t - w^*) - \eta\sqrt{1-\epsilon^2}\alpha_t \|X_t\|^2 - \eta\epsilon G_{t+1}^T X_t X_t^T(w_t - w^*)$$

Suppose that $\frac{c \log(\frac{1}{\beta})}{\sqrt{d}} \leq 8$, then by Lemmas 17 and 18, we have:

$$\mathbb{E}\left[\alpha_{t+1}^2\right] \leq (1-\epsilon^2)\mathbb{E}[\alpha_t^2] + \frac{\epsilon^2}{d}\mathbb{E}[\gamma_t^2] + 7\eta^2(1-\epsilon^2)\mathbb{E}[\alpha_t^2] + 7\frac{\eta^2\epsilon^2}{d}\mathbb{E}[\alpha_t^2]$$

$$+ 14\eta(1-\epsilon^2)\mathbb{E}[\alpha_t^2] - \frac{2\eta\epsilon^2}{d}\mathbb{E}[\alpha_t^2]$$

$$\leq \left( (1-\epsilon^2)(1 + 7\eta^2 + 14\eta) + \epsilon^2\left(\frac{7\eta^2}{d} - \frac{2\eta}{d}\right) \right)\mathbb{E}[\alpha_t^2] + \frac{\epsilon^2}{d}\mathbb{E}[\gamma_t^2]$$

$$= \left( 1 - \left[\epsilon^2 - (1-\epsilon^2)(7\eta^2 + 14\eta) - \frac{\epsilon^2}{d}\left(7\eta^2 - 2\eta\right)\right] \right)\mathbb{E}[\alpha_t^2] + \frac{\epsilon^2}{d}\mathbb{E}[\gamma_t^2]$$

Now we turn to $\gamma_{t+1}$. We have:

$$\gamma_{t+1}^2 = \|w_{t+1} - w^*\|^2 = (w_t - w^*)^T(I - \eta X_t X_t^T)(I - \eta X_t X_t^T)(w_t - w^*)$$

$$= (w_t - w^*)^T(I - (2\eta - \eta^2\|X_t\|^2)X_t X_t^T)(w_t - w^*)$$

$$= \gamma_t^2 - (2\eta - \eta^2\|X_t\|^2)\alpha_t^2$$

Therefore, we can say that $\mathbb{E}[\gamma_{t+1}^2] \geq \mathbb{E}[\gamma_t^2] - (2\eta - \eta^2(-7))\mathbb{E}[\alpha_t^2]$.

When $\epsilon^2 > 0.5$ and $\eta < 0.05$, it follows that

- $\mathbb{E}[\gamma_{t+1}^2] \leq \mathbb{E}[\gamma_t^2]$

- $\mathbb{E}\left[\alpha_{t+1}^2\right] \leq (1-\varsigma)\mathbb{E}\left[\alpha_t^2\right] + \frac{\epsilon^2}{d}\mathbb{E}[\gamma_t^2]$, where $0 < \varsigma < 1$

- Moreover, $\varsigma > 7\eta^2 + 2\eta$, so $\mathbb{E}[\gamma_{t+1}^2] \geq \mathbb{E}[\gamma_t^2] - \varsigma\mathbb{E}[\alpha_t^2]$.

Unwrapping, the recursion, we can say that

$$
\begin{aligned}
\mathbb{E}[\alpha_{t+1}^2] &\leq (1-\varsigma)\mathbb{E}[\alpha_t^2] + \frac{\epsilon^2}{d}\mathbb{E}[\gamma_1^2] \\
&\leq (1-\varsigma)\left((1-\varsigma)\mathbb{E}[\alpha_{t-1}^2] + \frac{\epsilon^2}{d}\mathbb{E}[\gamma_1^2]\right) + \frac{\epsilon^2}{d}\mathbb{E}[\gamma_1^2] \\
&\leq (1-\varsigma)^t\mathbb{E}[\alpha_1^2] + \frac{\epsilon^2}{d}\left(\sum_{j=0}^{t-1}(1-\varsigma)^j\right)\mathbb{E}[\gamma_1^2] \\
&\leq (1-\varsigma)^t\mathbb{E}[\alpha_1^2] + \frac{\epsilon^2}{d\varsigma}\mathbb{E}[\gamma_1^2]
\end{aligned}
$$

Note that $\mathbb{E}[\alpha_1^2] = \mathbb{E}\left[(X_1^T(w_1 - w^*))^2\right] = \frac{\mathbb{E}[\gamma_1^2]}{d}$ Therefore we can say that $\mathbb{E}[\alpha_{t+1}^2] \leq (1-\varsigma)^t\frac{\mathbb{E}[\gamma_1^2]}{d} + \frac{\epsilon^2}{d\varsigma}\mathbb{E}[\gamma_1^2]$

Now we unwrap the recursion for $\mathbb{E}[\gamma_{t+1}^2] \geq \mathbb{E}[\gamma_t^2] - \varsigma\mathbb{E}[\alpha_t^2]$. We have:

$$
\begin{aligned}
\mathbb{E}[\gamma_{t+1}^2] &\geq \mathbb{E}[\gamma_t^2] - \varsigma\mathbb{E}[\alpha_t^2] \\
&\geq \mathbb{E}[\gamma_{t-1}^2] - \varsigma\mathbb{E}[\alpha_{t-1}^2] - \varsigma\mathbb{E}[\alpha_t^2] \\
&\geq \mathbb{E}[\gamma_1^2] - \varsigma\sum_{j=1}^t\mathbb{E}[\alpha_j^2] \\
&\geq \mathbb{E}[\gamma_1^2] - \varsigma\sum_{j=1}^t\left((1-\varsigma)^{j-1}\mathbb{E}[\alpha_1^2] + \frac{\epsilon^2}{d\varsigma}\mathbb{E}[\gamma_1^2]\right) \\
&= \mathbb{E}[\gamma_1^2] - \varsigma\left(\frac{t\epsilon^2}{d\varsigma}\right)\mathbb{E}[\gamma_1^2] - \varsigma \cdot \frac{1 - (1-\varsigma)^t}{\varsigma} \cdot \mathbb{E}[\alpha_1^2] \\
&\geq \mathbb{E}[\gamma_1^2] - \varsigma\left(\frac{t\epsilon^2}{d\varsigma}\right)\mathbb{E}[\gamma_1^2] - \frac{1}{d}\mathbb{E}[\gamma_1^2]
\end{aligned}
$$

In order for $\frac{t\epsilon^2}{d} > \frac{1}{2}$, we need $t \geq \frac{d}{2\epsilon^2}$. Therefore the number of samples required is $T = \Omega\left(\frac{d}{\epsilon^2}\right)$. $\quad\square$