[Reviews · NeurIPS 2020]

Review 1

Summary and Contributions: This paper studies the problem of online least squares with dependent Markov data. The paper provides information-theoretical lower bounds for bias and variance. Regarding algorithms, the paper shows that SGD is sub-optimal but SGD-DD and parallel SGD are optimal. Furthermore, the paper shows that for a special case, the Gaussian Autoregressive model, SGD does not have a better rate than SGD-DD, and SGD with experience replay is better.

Strengths: The lower bounds and upper bounds show a very interesting separation between Markov data and normal i.i.d. data. I believe these results are significant enough to be accepted. === After rebuttal, the authors have addressed my concerns. I will maintain my sorces.

Weaknesses: The assumptions on the bounds need a better explanation.

Correctness: Yes. I believe so.

Clarity: Yes.

Relation to Prior Work: Yes.

Reproducibility: Yes

Additional Feedback: * I believe the following papers about PCA are related to the study of this paper. It might be good to compare them with your results. [http://auai.org/uai2019/proceedings/papers/299.pdf] is using SGD-DD [http://papers.nips.cc/paper/7609-dimensionality-reduction-for-stationary-time-series-via-stochastic-nonconvex-optimization.pdf] is directly using SGD while achieving some good rates. * In Table-1, why there is no LB for SGD, and no upper bound for SGD with Gaussian AR data. * Section 2.1, why not discuss carefully about the motivations of these assumptions? * Text after (3), what is the difference between w1 and w0? * Theorem 5: did you try to analyze SGD with decreasing step size? SGD with decreasing step size may perform well in terms of bias and does not waste data. * Figure 1: Although the theorems show that SGD-DD should be no worse than SGD, but simulation shows that SGD is better. Any explanation?


Review 2

Summary and Contributions: In this paper, the authors study a very interesting problem: the fundamental limits of SGD for the least square problem under Markovian data. The main contribution of this paper is to provide a good understanding on the performance of SGD under Markovian data through a sequence of lower and upper bounds for different noise settings. The dependence on the mixing time is also verified in the prior work. The authors also theoretically explain an important implementation in reinforcement learning, experience replay, which is missing in the literature.

Strengths: The best part of this paper is to use a simple setting ( least square problem) to nicely explain the limits of SGD when the data is Markovian. To the best of my knowledge, the results in this paper are new and complement the existing literature. I have tried to check the analysis in the appendix and it seems to be correct, althouth I may skip some details. I read the authors' feedback to my comments as well as other reviewers' comments. Overall, I think this is a good theoretical paper and would benefit for the community. I keep my decision and recommend to accept the paper.

Weaknesses: The paper is presented in a way that is not very clear to me on my first reading, in particular, the noise model. Perhaps, the authors may want to emphasize more on the noise model and Markovian data.

Correctness: The results seem to be correct. I tried to check the appendix but I may skip some details there.

Clarity: The paper is written nicely but can be improved about the explanation of the noise model and Markovian data.

Relation to Prior Work: Yes, although couple of references are missing. The authors may want to consider the following related work. 1. On Markov Chain Gradient Descent - Sun et. al. 2. Performance of Q-learning with Linear Function Approximation: Stability and Finite-Time Analysis - Chen et. al. 3. Convergence Rates of Accelerated Markov Gradient Descent with Applications in Reinforcement Learning - Doan et. al.

Reproducibility: Yes

Additional Feedback:


Review 3

Summary and Contributions: This paper studies the behavior of SGD under Markovian noise. The authors show the lower bound and the optimality of SGD-DD. They also show the possibility to improve the convergence of SGD by using experience-replay for a specific setting.

Strengths: The results on the lower bound and the optimality of SGD-DD appear to be known and hence are not particularly novel. The result on SGD with experience-replay might inspire some interest with limitation.

Weaknesses: 1. the lower bound and the optimality of SGD-DD is not new. 2. The analysis of SGD with experience-replay applies to a very specific setting, i.e., Gaussian AR Chain, and thus appears to be narrow.

Correctness: The reviewer did not check the proof due to the limited review time. This paper focuses on the analysis of algorithms, but not the invention of new methods, the empirical evidence does not seem to be essential.

Clarity: The paper is easy to follow,

Relation to Prior Work: Seem to be correct.

Reproducibility: Yes

Additional Feedback:


Review 4

Summary and Contributions: The authors theoretically study the problem of least squares regression where it is assumed that the data is generated from a Markov chain that has reached stationary. In this setting, the authors first establish information theoretic lower bounds for the minimax excess risk. It is shown that the convergence rate suffer by a factor \tau_mix that indicates the mixing time showing that the problem of Markovian data is intrinsically harder. It is also established that the lower bounds are tight by showing that for different noise settings SGD with data drop and Parallel SGD achieves the rate up to lograthmic factor. It is also shown that for both the noise settings, vanilla SGD with constant step size in sub-optimal. This is shown by constructing an example where updating at each step leads to a constant bias. Finally, for the setting where data is generated from a known Gaussian AR Markov chain, authors give a lower bound on the data (and steps) required to achieve \delta error. In this setting authors also study SGD with experience replay (ER). The main idea of SGD with ER It is shown empirically with a simulation study that SGD with experience replay performs better than vanilla SGD and SGD with Data Drop.

Strengths: In this paper, the authors study the problem of optimization where the data points have Markovian dependence and take a first step in this direction by studying the simple problem of least squares regression. In this setting (to the best of my knowledge), authors establish novel lower bounds on the excess risk.

Weaknesses: Main weakness of the paper is that the experimental evaluation is only done on toy data. It would have been useful to see the experimental results for non-iid data when the data is generated from a Markov chain that is more complicated than a simple Gaussian AR chain. I understand that the theoretical results were provided in this setting but it would have been useful to see how SGD with experience replay compared on a more complicated example.

Correctness: From only briefly skimming the appendix for the proofs, the results seem to be correct, but this is more of a judgement call than a thorough analysis.

Clarity: The paper is well written.

Relation to Prior Work: There is some prior work which has not been discussed. In particular Authors should discuss the relationship of their work to Wolfer, Geoffrey, and Aryeh Kontorovich. "Minimax learning of ergodic markov chains." In Algorithmic Learning Theory, pp. 904-930. 2019. This seems very related to the present work.

Reproducibility: Yes

Additional Feedback: =========== Post Rebuttal Comment ============ After looking at other reviews and author's feedback, I still believe this is a good submission and will keep my score at 8.

[Author Response · NeurIPS 2020]

We thank the reviewers for their valuable and constructive feedback. We will incorporate the suggestions in the next
revision. Our responses below.

**Reviewer 2**:

• **Related works**: Both of these works seem to use SGD-DD. The second paper in particular considers Gaussian Vector
Autoregressive (GVAR) model for the Markov chain and our proof techniques and results on experience replay could
perhaps help in this setting as well.

• **Bounds for SGD in Table 1**: (a) Lower bounds – The information theoretic lower bounds for bias in the independent
setting (Theorem 1) and variance in the agnostic setting (Theorem 2) also apply to SGD. (b) Upper bounds in
AR chain – Since we want to demonstrate the benefits of SGD-ER over SGD, we focus only on proving that the
upper-bounds for SGD-ER are better than the *lower bounds* for SGD.

• **Section 2.1, modeling assumptions**: The agnostic and independent noise settings are standard models to account
for inexact observations in real data and in nature and have been studied in literature e.g., see [7] and references there
in. The AR chain is a popular chain used for forecasting and time series prediction problems, and has nice structural
properties that benefit from experience replay. We will include a detailed discussion regarding this in the manuscript.

• **Text in and around equation (3)**: This is a typo. $w_1$ should be $w_0$, which is the initial iterate.

• **Theorem 5**: We restricted our attention to constant step size SGD since this is used widely in practice for least
squares regression and is optimal for i.i.d data. Analysis of SGD with decreasing step size would be a great direction
for future work, but even for iid data, decreasing step size leads to sub-optimal rates for the bias term.

• **Figure 1**: Our theorems show that SGD-DD is no worse than SGD in the worst case, i.e, for certain Markov chains.
For the AR chain, it seems that SGD performs better than SGD-DD due to the special structure of the AR process.
There is also a $\log(d)$ difference between the lower bound for SGD (specialized to the AR chain) and upper bound
for SGD-DD, as seen in Table 1.

**Reviewer 3**: We will include a detailed explanation of the noise model by explicitly constructing the underlying
probability space. We will also include the suggested references in related work.

**Reviewer 4**:

• **Lower bound and optimality of SGD-DD**: We respectfully disagree with the reviewer's claim that these results are
already known. We request the reviewer to please share related references that he/she thinks are relevant.

• **Gaussian AR chain is very specific and is of limited interest**: (a) As demonstrated by our lower bound results,
experience replay cannot in general improve upon SGD-DD in the worst-case. Improvements can be obtained only
for some well-structured Markov chains and we show that the AR chain is one such. (b) Gaussian AR chain is a
popular chain used for forecasting and time series prediction.

**Reviewer 5**: Our experimental results (a) validate the tightness of our theoretical bounds and (b) show that SGD-ER
outperforms SGD and SGD-DD even on small/moderate dimensional problems. Experience replay is widely used in
practice for reinforcement learning with great results, so we expect good outcomes from SGD-ER in other regression
settings as well.

[Meta-Review · NeurIPS 2020]

The paper makes strong theoretical contributions towards the understanding of SGD under Markovian data for the simple least square problems. After discussion, reviews were consistent about the novelty of the lower bound and optimality of SGD-DD. The analysis of SGD with experience-replay, albeit for a very narrow setting, is considered interesting. In the final version, please take the reviewer comments into account. In particular, the following should be addressed: - Detailed discussion and comparison to prior results of information-theoretic lower bounds under Markovian noise for general convex problems established in [Duchi et al., 2012] - Clear explanation of the noise model - Additional numerical experiments on a more complicated example